# GAMEGEN-𝕏: INTERACTIVE OPEN-WORLD GAME VIDEO GENERATION

**Haoxuan Che**[1*]**, Xuanhua He**[2,3*]**, Quande Liu**[4#]**, Cheng Jin**[1]**, Hao Chen**[1#]

[1]The Hong Kong University of Science and Technology
[2]University of Science and Technology of China
[3]Hefei Institute of Physical Science, Chinese Academy of Sciences
[4]The Chinese University of Hong Kong
`{hche, cjinag, jhc}@cse.ust.hk`
`hexuanhua@mail.ustc.edu.cn`
`qdliu0226@gmail.com`

## ABSTRACT

We introduce *GameGen-𝕏*, the first diffusion transformer model specifically designed for both generating and interactively controlling open-world game videos. This model facilitates high-quality, open-domain generation by approximating various game elements, such as innovative characters, dynamic environments, complex actions, and diverse events. Additionally, it provides interactive controllability, predicting and altering future content based on the current clip, thus allowing for gameplay simulation. To realize this vision, we first collected and built an *Open-World Video Game Dataset* (OGameData) from scratch. It is the first and largest dataset for open-world game video generation and control, which comprises over *one million* diverse gameplay video clips with informative captions. GameGen-𝕏 undergoes a two-stage training process, consisting of pre-training and instruction tuning. Firstly, the model was pre-trained via text-to-video generation and video continuation, enabling long-sequence open-domain game video generation with improved fidelity and coherence. Further, to achieve interactive controllability, we designed InstructNet to incorporate game-related multi-modal control signal experts. This allows the model to adjust latent representations based on user inputs, advancing the integration of character interaction and scene content control in video generation. During instruction tuning, only the InstructNet is updated while the pre-trained foundation model is frozen, enabling the integration of interactive controllability without loss of diversity and quality of generated content. GameGen-𝕏 contributes to advancements in open-world game design using generative models. It demonstrates the potential of generative models to serve as auxiliary tools to traditional rendering techniques, demonstrating the potential for merging creative generation with interactive capabilities. The project will be available at `https://github.com/GameGen-X/GameGen-X`.

## 1 INTRODUCTION

Generative models (Croitoru et al. (2023); Ramesh et al. (2022); Tim Brooks & Ramesh (2024); Rombach et al. (2022b)) have made remarkable progress in generating images or videos conditioned on multi-modal inputs such as text, images, and videos. These advancements have benefited content creation in design, advertising, animation, and film by reducing costs and effort. Inspired by the success of generative models in these creative fields, it is natural to explore their application in the modern game industry. This exploration is particularly important because developing open-world video game prototypes is a resource-intensive and costly endeavor, requiring substantial investment in concept design, asset creation, programming, and preliminary testing (Anastasia (2023)). Even early development stages of games still involved months of intensive work by small teams to build functional prototypes showcasing the game's potential (Wikipedia (2023)).

---

[*]Equal contribution; [#]Co-corresponding authors.

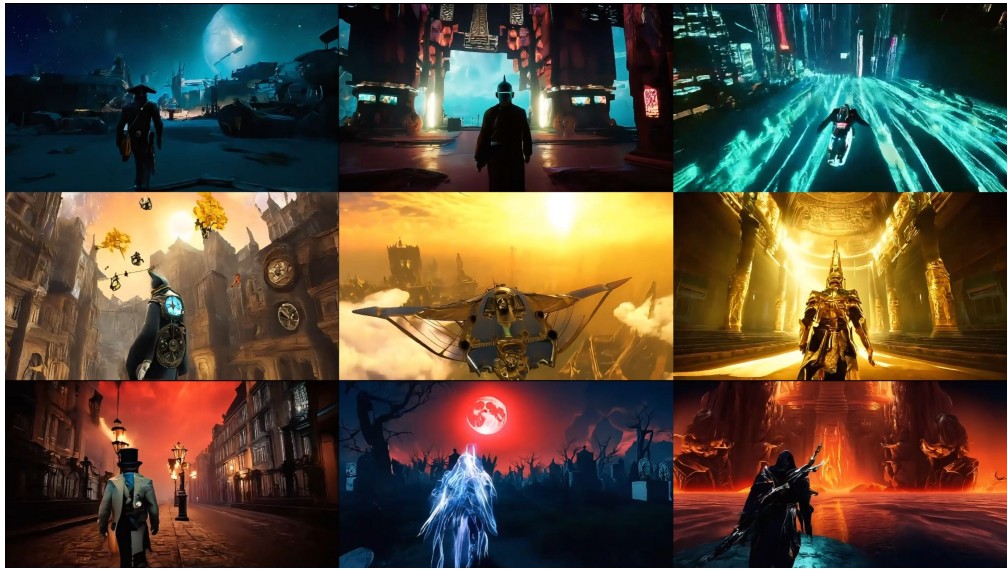

Figure 1: GameGen-𝕏 can generate novel open-world video games and enable interactive control to simulate game playing. *Best view with Acrobat Reader and click the image to play the interactive control demo videos.*

Several pioneering works, such as World Model (Ha & Schmidhuber (2018)), GameGAN (Kim et al. (2020)), R2PLAY (Jin et al. (2024)), Genie (Bruce et al. (2024)), and GameNGen (Valevski et al. (2024)), have explored the potential of neural models to simulate or play video games. They have primarily focused on 2D games like "Pac-Man", "Super Mario", and early 3D games such as "DOOM (1993)". Impressively, they demonstrated the feasibility of simulating interactive game environments. However, the generation of novel, complex open-world game content remains an open problem. A key difficulty lies in generating novel and coherent next-gen game content. Open-world games feature intricate environments, dynamic events, diverse characters, and complex actions that are far more challenging to generate (Eberly (2006)). Further, ensuring interactive controllability, where the generated content responds meaningfully to user inputs, remains a formidable challenge. Addressing these challenges is crucial for advancing the use of generative models in game content design and development. Moreover, successfully simulating and generating these games would also be meaningful for generative models, as they strive for highly realistic environments and interactions, which in turn may approach real-world simulation (Zhu et al. (2024)).

In this work, we provide an initial answer to the question: *Can a diffusion model generate and control high-quality, complex open-world video game content?* Specifically, we introduce *GameGen-𝕏*, the first diffusion transformer model capable of both generating and simulating open-world video games with interactive control. GameGen-𝕏 sets a new benchmark by excelling at generating diverse and creative game content, including dynamic environments, varied characters, engaging events, and complex actions. Moreover, GameGen-𝕏 enables interactive control within generative models, allowing users to influence the generated content and unifying character interaction and scene content control for the first time. It initially generates a video clip to set up the environment and characters. Subsequently, it produces video clips that dynamically respond to user inputs by leveraging the current video clip and multimodal user control signals. This process can be seen as simulating a game-like experience where both the environment and characters evolve dynamically.

GameGen-𝕏 undergoes a two-stage training: foundation model pre-training and instruction tuning. In the first stage, the foundation model is pre-trained on OGameData using text-to-video generation and video continuation tasks. This enables the model to learn a broad range of open-world game dynamics and generate high-quality game content. In the second stage, InstructNet is designed to enable multi-modal interactive controllability. The foundation model is frozen, and InstructNet is trained to map user inputs—such as structured text instructions for game environment dynamics and keyboard controls for character movements and actions—onto the generated game content. This allows GameGen-𝕏 to generate coherent and controllable video clips that evolve based on the player's inputs, simulating an interactive gaming experience. To facilitate this development, we constructed

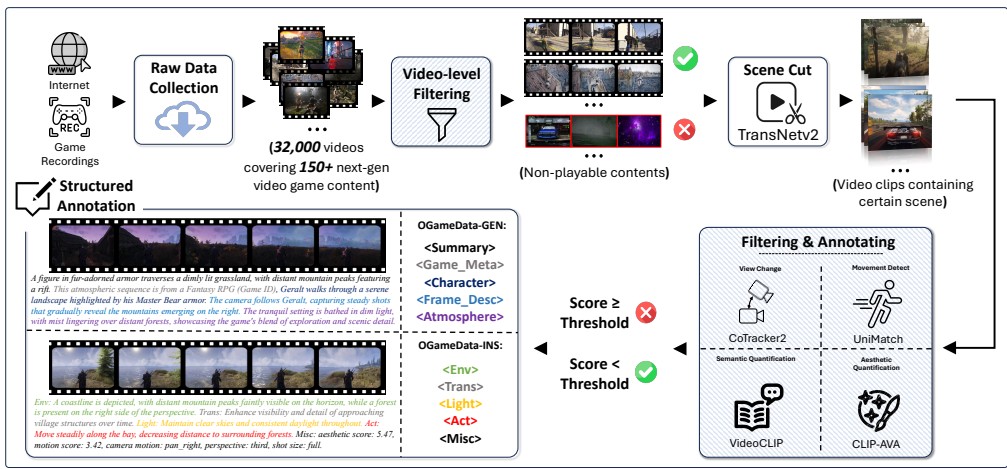

Figure 2: The OGameData collection and processing pipeline with human-in-the-loop.

*Open-World Video Game Dataset* (OGameData), the first large-scale dataset for game video generation and control. This dataset contained videos from over 150 next-generation games and was built by using a human-in-the-loop proprietary data pipeline that involves scoring, filtering, sorting, and structural captioning. OGameData contains one million video clips from two subsets including OGameData-GEN and OGameData-INS, providing the foundation for training generative models capable of producing realistic game content and achieving interactive control, respectively.

In summary, GameGen-𝕏 offers a novel approach for interactive open-world game video generation, where complex game content is generated and controlled interactively. While challenges for practical application remain, it explores the potential for generative models to complement traditional game design methods by offering a new approach to scalable content generation. Our main contributions are summarized as follows: **1)** We developed OGameData, a large-scale dataset curated for open-world game video generation and interactive control, containing one million video-text pairs. It is collected from 150+ next-gen games, and empowered by vision language models. **2)** We introduced GameGen-𝕏, the first generative model for open-world video game content, combining a foundation model with the InstructNet. GameGen-𝕏 utilizes a two-stage training strategy, with the foundation model and InstructNet trained separately to ensure stable, high-quality content generation and control. InstructNet provides multi-modal interactive controllability, allowing players to influence the continuation of generated content, simulating gameplay. **3)** We conducted extensive experiments comparing our model's generative and interactive control abilities to other open-source and commercial models. Results show that our approach excels in high-quality game content generation and offers superior control over the environment and character. Experimental results indicate that our approach achieves strong performance in game content generation and provides enhanced control over the environment and character compared to other models.

## 2 OGAMEDATA: LARGE-SCALE FINE-GRAINED GAME DOMAIN DATASET

**OGameData** is the first dataset designed for open-world game video generation and interactive control. As shown in Table 1, OGameData excels in fine-grained annotations, offering a structural caption with high text density for video clips per minute. It is meticulously designed for game video by offering game-specific knowledge and incorporating elements such as game names, player perspectives, and character details. It comprises two parts: the generation dataset (OGameData-GEN) and the instruction dataset (OGameData-INS). The resulting OGameData-GEN is tailored for training the generative foundation model, while OGameData-INS is optimized for instruction tuning and interactive control tasks. The details and analysis are in Appendix B.

### 2.1 DATASET CONSTRUCTION PIPELINE

As illustrated in Figure 2, we developed a robust data processing pipeline encompassing collection, cleaning, segmentation, filtering, and structured caption annotation. This process integrates

both AI and human expertise, as automated techniques alone are insufficient due to domain-specific intricacies present in various games.

**Data Collection and Filtering.** We gathered video from the Internet, local game engines, and existing dataset (Chen et al. (2024); Ju et al. (2024)), which contain more than 150 next-gen games and game engine direct outputs. These data specifically focus on gameplay footage that minimizes UI elements. Despite the rigorous collection, some low-quality videos were included, and these videos lacked essential metadata like game name, genre, and player perspective. Low-quality videos were manually filtered out, with human experts ensuring the integrity of the metadata, such as game genre and player perspective. To prepare videos for clip segmentation, we used PyScene and TransNetV2 (Souček & Lokoč (2020)) to detect scene changes, discarding clips shorter than 4 seconds and splitting longer clips into 16-second segments. To filter and annotate clips, we sequentially employed models: CLIP-AVA (Schuhmann (2023)) for aesthetic scoring, UniMatch (Xu et al. (2023)) for motion filtering, VideoCLIP (Xu et al. (2021)) for content similarity, and CoTrackerV2 (Karaev et al. (2023)) for camera motion.

**Structured Text Captioning.** The OGameData supports the training of two key functionalities: text-to-video generation and interactive control. These tasks require distinct captioning strategies. For OGameData-GEN, detailed captions are crafted to describe the game metadata, scene context, and key characters, ensuring comprehensive textual descriptions for the generative model foundation training. In contrast, OGameData-INS focuses on describing the changes in game scenes for interactive generation, using concise instruction-based captions that highlight differences between initial and subsequent frames. This structured captioning approach enables precise and fine-grained generation and control, allowing the model to modify specific elements while preserving the scene.

Table 1: Comparison of OGameData and previous large-scale text-video paired datasets.

| Dataset | Domain | Text-video pairs | Caption density | Captioner | Resolution | Purpose | Total video len. |
|---|---|---|---|---|---|---|---|
| ActivityNet (Caba Heilbron et al. (2015)) | Action | 85K | 23 words/min | Manual | - | Understanding | 849h |
| DiDeMo (Anne Hendricks et al. (2017)) | Flickr | 45k | 70 words/min | Manual | - | Temporal localization | 87h |
| YouCook2 (Zhou et al. (2018)) | Cooking | 32k | 26 words/min | Manual | - | Understanding | 176h |
| How2 (Sanabria et al. (2018)) | Instruct | 191k | 207 words/min | Manual | - | Understanding | 308h |
| MiraData (Ju et al. (2024)) | Open | 330k | 264 words/min | GPT-4V | 720P | Generation | 16000h |
| OGameData (Ours) | **Game** | **1000k** | **607 words/min** | **GPT-4o** | **720P-4k** | **Generation & Control** | 4000h |

## 2.2 DATASET SUMMARY

As depicted in Table 1, OGameData comprises 1 million high-resolution video clips, derived from sources spanning minutes to hours. Compared to other domain-specific datasets (Caba Heilbron et al. (2015); Zhou et al. (2018); Sanabria et al. (2018); Anne Hendricks et al. (2017)), OGameData stands out for its scale, diversity, and richness of text-video pairs. Unlike the latest open-domain generation dataset Miradata (Ju et al. (2024)), OGameData focuses on the game domain and provides more fine-grained annotations, featuring extensive captions per unit of time. This is crucial for training interactive controllability. This dataset features several key characteristics: OGameData features highly fine-grained text and boasts a large number of trainable video-text pairs, enhancing text-video alignment in model training. Additionally, it comprises two subsets—generation and control—supporting both types of training tasks. The dataset's high quality is ensured by meticulous curation from over 10 human experts. Each video clip is accompanied by captions generated using vision language models, maintaining clarity and coherence and ensuring the dataset remains free of UI and visual artifacts. Critical to its design, OGameData is tailored specifically for the gaming domain. It effectively excludes non-gameplay scenes, incorporating a diverse array of game styles while preserving authentic in-game camera perspectives. This specialization ensures the dataset represents real gaming experiences, maintaining high domain-specific relevance.

## 3 GAMEGEN-𝕏

**GameGen-𝕏** is a novel generative diffusion model that learns to generate open-world game videos and interactively control the environments and characters in them. The overall framework is illustrated in Fig 3. In section 3.1, we introduce the problem formulation. In section 3.2, we discuss the design and training of the foundation model, which facilitates both initial game content generation and video continuation. In section 3.3, we delve into the design of InstructNet and explain the process of instruction tuning, which enables clip-level interactive control over generated content.

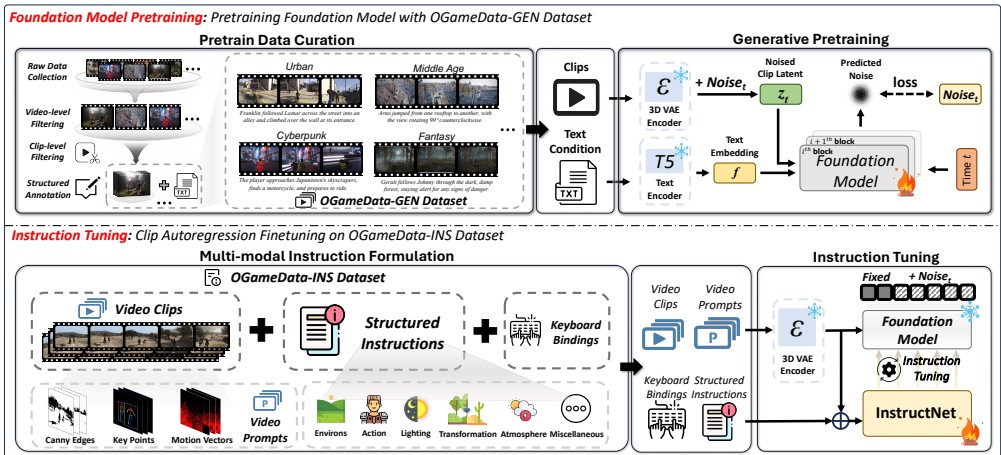

Figure 3: An overview of our two-stage training framework. In the first stage, we train the foundation model via OGameData-GEN. In the second stage, InstructNet is trained via OGameData-INS.

## 3.1 GAME VIDEO GENERATION AND INTERACTION

The primary objective of GameGen-$\mathbb{X}$ is to generate dynamic game content where both the virtual environment and characters are synthesized from textual descriptions, and users can further influence the generated content through interactive controls. Given a textual description $T$ that specifies the initial game scene—including characters, environments, and corresponding actions and events—we aim to generate a video sequence $\mathbb{V} = \{V_t\}_{t=1}^{N}$ that brings this scene to life. We model the conditional distribution: $p(V_{1:N} \mid T, C_{1:N})$, where $C_{1:N}$ represents the sequence of multi-modal control inputs provided by the user over time. These control inputs allow users to manipulate character movements and scene dynamics, simulating an interactive gaming experience.

Our approach integrates two main components: 1) **Foundation Model**: It generates an initial video clip based on $T$, capturing the specified game elements including characters and environments. 2) **InstructNet**: It enables the controllable continuation of the video clip by incorporating user-provided control inputs. By unifying text-to-video generation with interactive controllable video continuation, our approach synthesizes game-like video sequences where the content evolves in response to user interactions. Users can influence the generated video at each generation step by providing control inputs, allowing for manipulation of the narrative and visual aspects of the content.

## 3.2 FOUNDATION MODEL TRAINING FOR GENERATION

**Video Clip Compression.** To address the redundancy in temporal and spatial information (Lab & etc. (2024)), we introduce a 3D Spatio-Temporal Variational Autoencoder (3D-VAE) to compress video clips into latent representations. This compression enables efficient training on high-resolution videos with longer frame sequences. Let $V \in \mathbb{R}^{F \times C \times H \times W}$ denote a video clip, where $F$ is the number of frames, $H$ and $W$ are the height and width of each frame, and $C$ is the number of channels. The encoder $E$ compresses $V$ into a latent representation $z = E(V) \in \mathbb{R}^{F' \times C' \times H' \times W'}$, where $F' = F/s_f$, $H' = H/s_h$, $W' = W/s_w$, and $C'$ is the number of latent channels. Here, $s_t$, $s_h$, and $s_w$ are the temporal and spatial downsampling factors. Specifically, 3D-VAE first performs the spatial downsampling to obtain frame-level latent features. Further, it conducts temporal compression to capture temporal dependencies and reduce redundancy over frame effectively, inspired by Yu et al. (2023a). By processing the video clip through the 3D-VAE, we can obtain a latent tensor $z$ of spatial-temporally informative and reduced dimensions. Such $z$ can support long video and high-resolution model training, which meets the requirements of game content generation.

**Masked Spatial-Temporal Diffusion Transformer.** GameGen-$\mathbb{X}$ introduces a Masked Spatial-Temporal Diffusion Transformer (MSDiT). Specifically, MSDiT combines spatial attention, temporal attention, and cross-attention mechanisms (Vaswani (2017)) to effectively generate game videos guided by text prompts. For each time step $t$, the model processes latent features $z_t$ that capture frame details. Spatial attention enhances intra-frame relationships by applying self-attention over

spatial dimensions $(H', W')$. Temporal attention ensures coherence across frames by operating over the time dimension $F'$, capturing inter-frame dependencies. Cross-attention integrates guidance of external text features $f$ obtained via T5 (Raffel et al. (2020a)), aligning video generation with the semantic information from text prompts. As shown in Fig. 4, we adopt the design of stacking paired spatial and temporal blocks, where each block is equipped with cross-attention and one of spatial or temporal attention. Such design allows the model to capture spatial details, temporal dynamics, and textual guidance simultaneously, enabling GameGen-$\mathbb{X}$ to generate high-fidelity, temporally consistent videos that are closely aligned with the provided text prompts.

Additionally, we introduce a masking mechanism that excludes certain frames from noise addition and denoising during diffusion processing. A masking function $M(i)$ over frame indices $i \in I$ is defined as: $M(i) = 1$ if $i > x$, and $M(i) = 0$ if $i \leq x$, where $x$ is the number of context frames provided for video continuation. The noisy latent representation at time step $t$ is computed as: $\tilde{z}_t = (1 - M(I)) \odot z + M(I) \odot \epsilon_t$, where $\epsilon_t \sim \mathcal{N}(0, \mathbf{I})$ is Gaussian noise of the same dimensions as $z$, and $\odot$ denotes element-wise multiplication. Such a masking strategy provides the support of training both text-to-video and video continuation into one foundation model.

**Unified Video Generation and Continuation.** By integrating the text-to-video diffusion training logic with the masking mechanism, GameGen-$\mathbb{X}$ effectively handles both video generation and continuation tasks within a unified framework. This strategy aims to enhance the simulation experience by enabling temporal continuity, catering to an extended and immersive gameplay experience. Specifically, for *text-to-video generation*, where no initial frames are provided, we set $x = 0$, and the masking function becomes $M(i) = 1$ for all frames $i$. The model learns the conditional distribution $p(V \mid T)$, where $T$ is the text prompt. The diffusion process is applied to all frames, and the model generates video content solely based on the text prompt. For *video continuation*, initial frames $v_{1:x}$ are provided as context. The masking mechanism ensures that these frames remain unchanged during the diffusion process, as $M(i) = 0$ for $i \leq x$. The model focuses on generating the subsequent frames $v_{x+1:N}$ by learning the conditional distribution $p(v_{x+1:N} \mid v_{1:x}, T)$. This allows the model to produce video continuations that are consistent with both the preceding context and the text prompt. Additionally, during the diffusion training (Song et al. (2020a;b); Ho et al. (2020); Rombach et al. (2022a)), we incorporated the bucket training (Zheng et al. (2024b), classifier-free diffusion guidance (Ho & Salimans (2021)) and rectified flow (Liu et al. (2023b)) for better generation performance. Overall, this unified training approach enhances the ability to generate complex, contextually relevant open-world game videos while ensuring smooth transitions and continuations.

## 3.3 Instruction Tuning for Interactive Control

**InstructNet Design.** To enable interactive controllability in video generation, we propose *InstructNet*, designed to guide the foundation model's predictions based on user inputs, allowing for control of the generated content. The core concept is that the generation capability is provided by the foundation model, with InstructNet subtly adjusting the predicted content using user input signals. Given the high requirement for visual continuity in-game content, our approach aims to minimize abrupt changes, ensuring a seamless experience. Specifically, the primary purpose of InstructNet is to modify future predictions based on instructions. When no user input signal is given, the video extends naturally. Therefore, we keep the parameters of the pre-trained foundation model frozen, which preserves its inherent generation and continuation abilities. Meanwhile, the additional trainable InstructNet is introduced to handle control signals. As shown in Fig. 4, InstructNet modifies the generation process by incorporating control signals via the operation fusion expert layer and instruction fusion expert layer. This component comprises $N$ InstructNet blocks, each utilizing a specialized Operation Fusion Expert Layer and an Instruct Fusion Expert Layer to integrate different conditions. The output features are injected into the foundation model to fuse the original latent, modulating the latent representations based on user inputs and effectively aligning the output with user intent. This enables users to influence character movements and scene dynamics. InstructNet is primarily trained through video continuation to simulate the control and feedback mechanism in gameplay. Further, Gaussian noise is subtly added to initial frames to mitigate error accumulation.

**Multi-modal Experts.** Our approach leverages multi-modal experts to handle diverse controls, which is crucial for several reasons. Intuitively, each structured text, keyboard binding, and video prompt—uniquely impacts the video prediction process, requiring specialized handling to fully capture their distinct characteristics. By employing multi-modal experts, we can effectively integrate

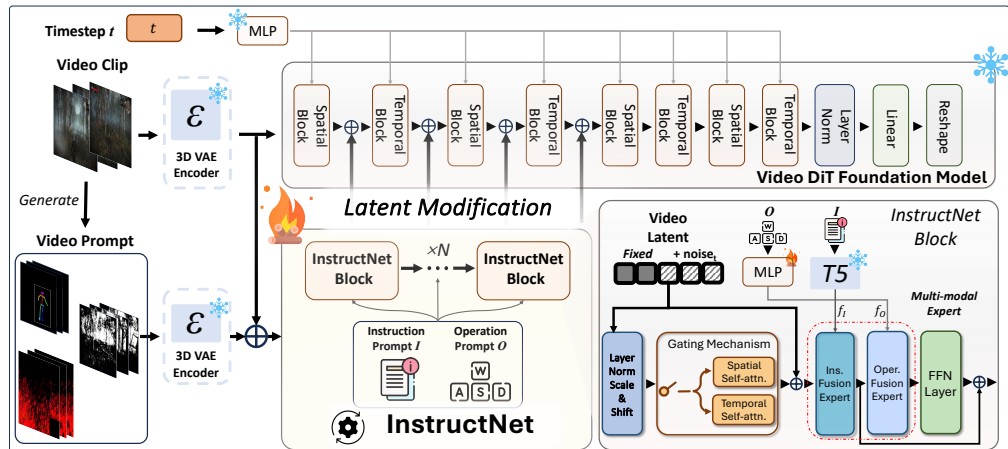

Figure 4: The architecture of GameGen-$\mathbb{X}$, including the foundation model and InstructNet. It enables the latent modification based on user inputs, mainly instruction and operation prompts, allowing for interactive control over the video generation process.

these varied inputs, ensuring that each control signal is well utilized. Let $f_I$ and $f_O$ be structured instruction embedding and keyboard input embedding, respectively. $f_O$ is used to modulate the latent features via operation fusion expert as follows: $\hat{z} = \gamma(f_O) \odot \frac{z-\mu}{\sigma} + \beta(f_O)$, where $\mu$ and $\sigma$ are the mean and standard deviation of $z$, $\gamma(f_O)$ and $\beta(f_O)$ are scale and shift parameters predicted by a neural network conditioned on $c$, where $c$ includes both structured text instructions and keyboard inputs. , and $\odot$ denotes element-wise multiplication. The keyboard signal primarily influences video motion direction and character control, exerting minimal impact on scene content. Consequently, a lightweight feature scaling and shifting approach is sufficient to effectively process this information. The instruction text is primarily responsible for controlling complex scene elements such as the environment and lighting. To incorporate this text information into InstructNet, we utilize an instruction fusion expert, which integrates $f_I$ into the model through cross-attention. Video prompts $V_p$, such as canny edges, motion vectors, or pose sequences, are introduced to provide auxiliary information. These prompts are processed through the 3D-VAE encoder to extract features $e_p$, which are then incorporated into the InstructNet via addition with $z$. It's worth clarifying that, during the inference, these video prompts are not necessary, except to execute the complex action generation or video editing.

**Interactive Control.** Interactive control is achieved through an autoregressive generation process. Based on a sequence of past frames $v_{1:x}$, the model generates future frames $v_{x+1:N}$ while adhering to control signals. The overall objective is to model the conditional distribution: $p(v_{x+1:N} \mid v_{1:x}, c)$. During generation, the foundation model predicts future latent representations, and InstructNet modifies these predictions according to the control signals. Thus, users can influence the video's progression by providing structured text commands or keyboard inputs, enabling a high degree of controllability in the open-world game environment. Furthermore, the incorporation of video prompts $V_p$ provides additional guidance, making it possible to edit or stylize videos quickly, which is particularly useful for specific motion patterns.

## 4 EXPERIMENTS

### 4.1 QUANTITATIVE RESULTS

**Metrics.** To comprehensively evaluate the performance of GameGen-$\mathbb{X}$, we utilize a suite of metrics that capture various aspects of video generation quality and interactive control, following Huang et al. (2024b) and Yang et al. (2024). These metrics include Fréchet Inception Distance (FID), Fréchet Video Distance (FVD), Text-Video Alignment (TVA), User Preference (UP), Motion Smoothness (MS), Dynamic Degrees (DD), Subject Consistency (SC), and Imaging Quality (IQ). It's worth noting that the TVA and UP are subjective scores that indicate whether the generation meets the requirements of humans, following Yang et al. (2024). By employing this comprehensive set of metrics, we aim to thoroughly evaluate model capabilities in generating high-quality, realistic,

and interactively controllable video game content. Readers can find experimental settings and metric introductions in Appendix D.2.

Table 2: Generation Performance Evaluation (* denotes key metric for generation ability)

| Method | Resolution | Frames | FID* ↓ | FVD* ↓ | TVA* ↑ | UP* ↑ | MS ↑ | DD ↑ | SC ↑ | IQ ↑ |
|---|---|---|---|---|---|---|---|---|---|---|
| Mira (Zhang et al. (2023)) | 480p | 60 | 360.9 | 2254.2 | 0.27 | 0.25 | 0.98 | 0.62 | 0.94 | 0.63 |
| OpenSora-Plan1.2 (Lab & etc. (2024)) | 720p | 102 | 407.0 | 1940.9 | 0.38 | 0.43 | 0.99 | 0.42 | 0.92 | 0.39 |
| CogVideoX-5B (Yang et al. (2024)) | 480p | 49 | 316.9 | 1310.2 | 0.49 | 0.37 | 0.99 | **0.94** | 0.92 | **0.53** |
| OpenSora1.2 (Zheng et al. (2024b)) | 720p | 102 | 318.1 | 1016.3 | 0.50 | 0.37 | 0.98 | 0.90 | 0.87 | 0.52 |
| GameGen-𝕏 (Ours) | 720p | 102 | **252.1** | **759.8** | **0.87** | **0.82** | 0.99 | 0.80 | 0.94 | 0.50 |

Table 3: Control Performance Evaluation (* denotes key metric for control ability)

| Method | Resolution | Frames | SR-C* ↑ | SR-E* ↑ | UP ↑ | MS ↑ | DD ↑ | SC ↑ | IQ ↑ |
|---|---|---|---|---|---|---|---|---|---|
| OpenSora-Plan1.2 (Lab & etc. (2024)) | 720p | 102 | 26.6% | 31.7% | 0.46 | 0.99 | 0.72 | **0.90** | 0.51 |
| CogVideoX-5B (Yang et al. (2024)) | 480p | 49 | 23.0% | 30.3% | 0.45 | 0.98 | 0.63 | 0.85 | **0.55** |
| OpenSora1.2 (Zheng et al. (2024b)) | 720p | 102 | 21.6% | 14.2% | 0.17 | 0.99 | **0.97** | 0.84 | 0.45 |
| GameGen-𝕏 (Ours) | 720p | 102 | **63.0%** | **56.8%** | **0.71** | 0.99 | 0.88 | 0.88 | 0.44 |

**Generation and Control Ability Evaluation.** As shown in Table 2, we compared GameGen-𝕏 against four well-known open-source models, i.e., Mira (Zhang et al. (2023)), OpenSora-Plan1.2 (Lab & etc. (2024)), OpenSora1.2 (Zheng et al. (2024b)) and CogVideoX-5B (Yang et al. (2024)) to evaluate its generation capabilities. Notably, both Mira and OpenSora1.2 explicitly mention training on game data, while the other two models, although not specifically designed for this purpose, can still fulfill certain generation needs within similar contexts. Our evaluation showed that GameGen-𝕏 performed well on metrics such as FID, FVD, TVA, MS, and SC. It implies GameGen-𝕏's strengths in generating high-quality and coherent video game content while maintaining competitive visual and technical quality. Further, we investigated the control ability of these models, except Mira, which does not support video continuation, as shown in Table 3. We used conditioned video clips and dense prompts to evaluate the model generation response. For GameGen-𝕏, we employed instruct prompts to generate video clips. Beyond the aforementioned metrics, we introduced the Success Rate (SR) to measure how often the models respond accurately to control signals. This is evaluated by both human experts and PLLaVA (Xu et al. (2024)). The SR metric is divided into two parts: SR for Character Actions (SR-C), which assesses the model's responsiveness to character movements, and SR for Environment Events (SR-E), which evaluates the model's handling of changes in weather, lighting, and objects. As demonstrated, GameGen-𝕏 exhibits superior control ability compared to other models, highlighting its effectiveness in generating contextually appropriate and interactive game content. Since IQ metrics favor models trained on natural scene datasets, such models score higher. In generation performance, CogVideo's 8fps videos and OpenSora 1.2's frequent scene changes result in higher DD.

Table 4: Ablation Study for Generation Ability

| Method | FID ↓ | FVD ↓ | TVA ↑ | UP ↑ | MS ↑ | SC ↑ |
|---|---|---|---|---|---|---|
| w/ MiraData | 303.7 | 1423.6 | 0.70 | 0.48 | 0.99 | 0.94 |
| w/ Short Caption | 303.8 | **1167.7** | 0.53 | 0.49 | 0.99 | 0.94 |
| w/ Progression | 294.2 | 1169.8 | 0.68 | 0.53 | 0.99 | 0.93 |
| Baseline | **289.5** | 1181.3 | **0.83** | **0.67** | 0.99 | **0.95** |

Table 5: Ablation Study for Control Ability.

| Method | SR-C ↑ | SR-E ↑ | UP ↑ | MS ↑ | SC ↑ |
|---|---|---|---|---|---|
| w/o Instruct Caption | 31.6% | 20.0% | 0.34 | 0.99 | 0.87 |
| w/o Decomposition | 32.7% | 23.3% | 0.41 | 0.99 | 0.88 |
| w/o InstructNet | 12.3% | 17.5% | 0.16 | 0.98 | 0.86 |
| Baseline | **45.6%** | **45.0%** | **0.50** | 0.99 | **0.90** |

**Ablation Study.** As shown in Table 4, we investigated the influence of various data strategies, including leveraging MiraData (Ju et al. (2024)), short captions (Chen et al. (2024)), and progression training (Lab & etc. (2024)). The results indicated that our data strategy outperforms the others, particularly in terms of semantic consistency, distribution alignment, and user preference. The visual quality metrics are comparable across all strategies. This consistency implies that visual quality metrics may be less sensitive to these strategies or that they might be limited in evaluating game domain generation. Further, as shown in Table 5, we explored the effects of our design on interactive control ability through ablation studies. This experiment involved evaluating the impact of removing key components such as InstructNet, Instruct Captions, or the decomposition process. The results demonstrate that the absence of InstructNet significantly reduces the SR and UP, highlighting its crucial role in user-preference interactive controllability. Similarly, the removal of Instruct Captions and the decomposition process also negatively impacts control metrics, although to a lesser extent. These findings underscore the importance of each component in enhancing the model's ability to generate and control game content interactively.

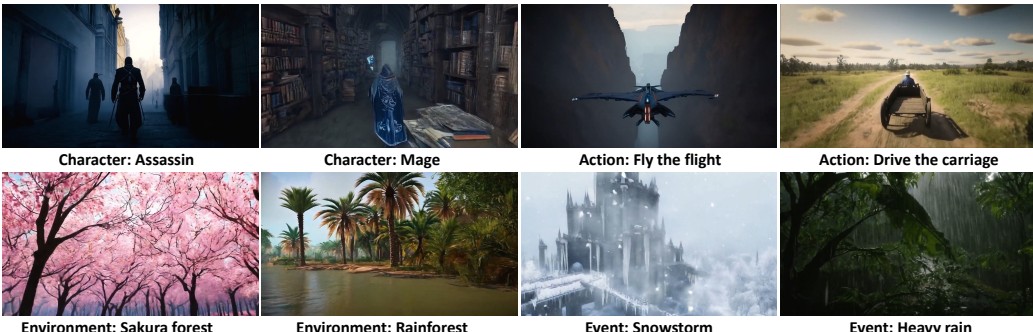

Figure 5: The generation showcases of characters, environments, actions, and events.

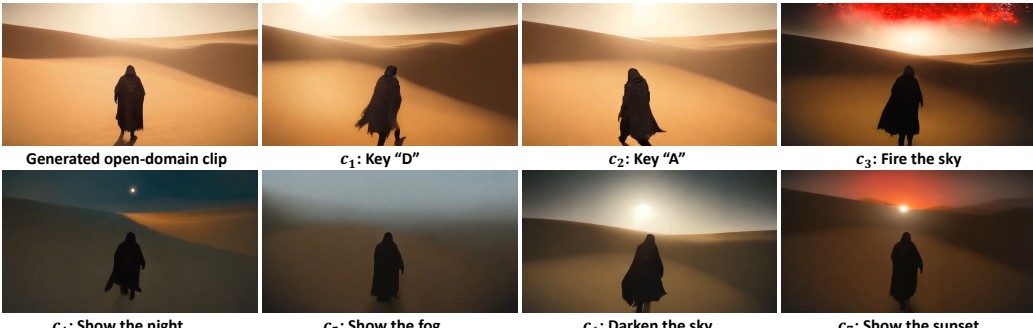

Figure 6: The qualitative results of different control signals, given the same open-domain clip.

## 4.2 QUALITATIVE RESULTS

**Generation Functionality.** Fig. 5 illustrates the basic generation capabilities of our model in generating a variety of characters, environments, actions, and events. The examples show that the model can create characters such as assassins and mages, simulate environments such as Sakura forests and rainforests, execute complex actions like flying and driving, and reproduce environmental events like snowstorms and heavy rain. This demonstrates the model's ability to generate and control diverse scenarios, highlighting its potential application in generating open-world game videos.

**Interactive Control Ability.** As shown in Fig. 6, our model demonstrates the capability to control both environmental events and character actions based on textual instructions and keyboard inputs. In the example provided, the model effectively manipulates various aspects of the scene, such as lighting conditions and atmospheric effects, highlighting its ability to simulate different times of day and weather conditions. Additionally, the character's movements, primarily involving navigation through the environment, are precisely controlled through input keyboard signals. This interactive control mechanism enables the simulation of a dynamic gameplay experience. By adjusting environmental factors like lighting and atmosphere, the model provides a realistic and immersive setting. Simultaneously, the ability to manage character movements ensures that the generated content responds intuitively to user interactions. Through these capabilities, our model showcases its potential to enhance the realism and engagement of open-world video game simulations.

**Open-domain Generation, Gameplay Simulation and Further Analysis.** As shown in Fig. , we presented initial qualitative experiment results, where GameGen-$\mathbb{X}$ generates novel domain game video clips and interactively controls them, which can be seen as a game simulation. Further, we compared GameGen-$\mathbb{X}$ with other open-source models in the open-domain generation ability as shown in Fig. 7. All the open-source models can generate some game-like content, implying their training involves corresponding game source data. As expected, the GameGen-$\mathbb{X}$ can better meet the game content requirements in character details, visual environments, and camera logic, owing to the strict dataset collection and building of OGameData. Further, we compared GameGen-$\mathbb{X}$ with other commercial products including Kling, Pika, Runway, Luma, and Tongyi, as shown in Fig. 8. In the left part, i.e., the initially generated video clip, only Pika, Kling1.5, and GameGen-$\mathbb{X}$ correctly followed the text description. Other models either failed to display the character or depicted them entering the cave instead of exiting. In the right part, both GameGen-$\mathbb{X}$ and Kling1.5 successfully guided the character out of the cave. GameGen-$\mathbb{X}$ achieved high-quality control response as well as

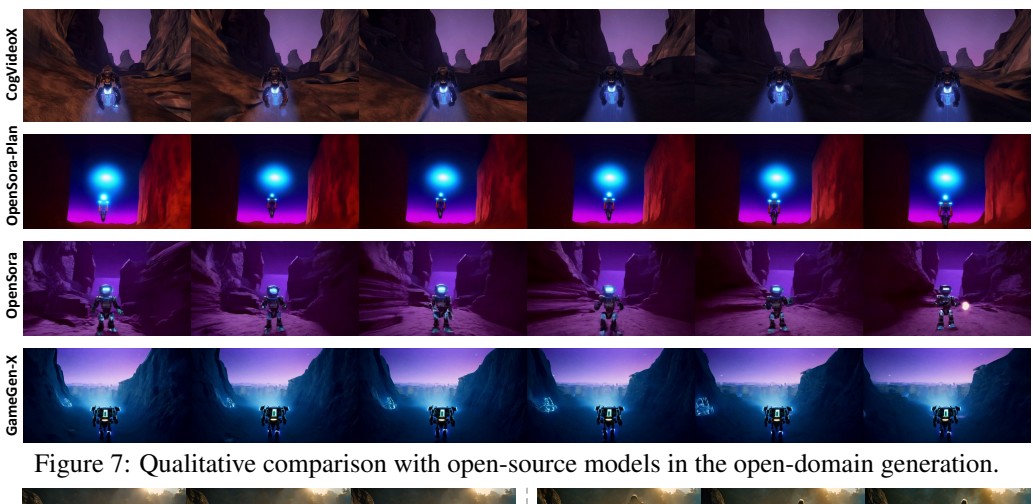

Figure 7: Qualitative comparison with open-source models in the open-domain generation.

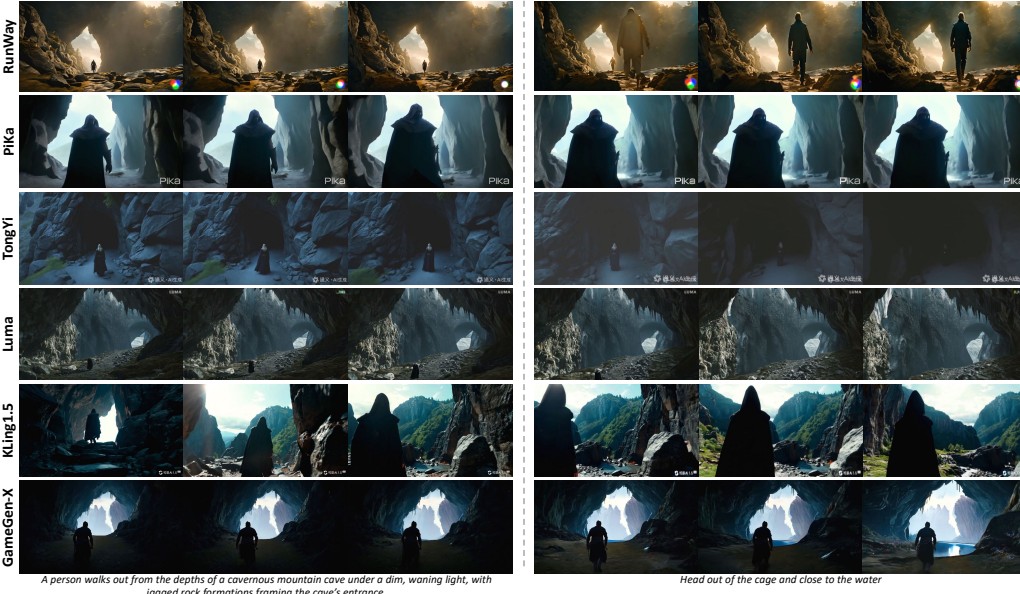

*A person walks out from the depths of a cavernous mountain cave under a dim, waning light, with jagged rock formations framing the cave's entrance.*

*Head out of the cage and close to the water*

Figure 8: Qualitative comparison with commercial models in the interactive control ability.

maintaining a consistent camera logic, obeying the game-like experience at the same time. This is owing to the design of a holistic training framework and InstructNet.

# 5    CONCLUSION

We have presented *GameGen-X*, the first diffusion transformer model with multi-modal interactive control capabilities, specifically designed for generating open-world game videos. By simulating key elements such as dynamic environments, complex characters, and interactive gameplay, GameGen-X advances the state of generative models by demonstrating their potential in both generating and controlling game content. The development of the *OGameData* provided a crucial foundation for our model's training, enabling it to capture the diverse and intricate nature of open-world games. Through a two-stage training process, GameGen-X integrates content generation with interactive control, improving coherence and user adaptability in simulated gameplay. Beyond its technical contributions, GameGen-X presents new possibilities for game content design. It suggests a shift towards more automated, data-driven methods that could help streamline early-stage game content creation. By leveraging models to create immersive worlds and interactive gameplay, we may move closer to a future where game engines are more attuned to creative, user-guided experiences. While challenges remain (Appendix E), GameGen-X represents an early step toward a novel paradigm in game design. It lays the groundwork for future research into the role of generative models in creating interactive digital worlds.

**Acknowledgments.** This work was supported by the Hong Kong Innovation and Technology Commission (Project No. MHP/002/22 and ITCPD/17-9). Additionally, we extend our sincere gratitude for the valuable discussions, comments, and help provided by Dr. Guangyi Liu, Mr. Wei Lin and Mr. Jingran Su (listed in alphabetical order). We also appreciate the HKUST SuperPOD for computation support.

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

# A RELATED WORKS

## A.1 VIDEO DIFFUSION MODELS

The advent of diffusion models, particularly latent diffusion models, has significantly advanced image generation, inspiring researchers to extend their applicability to video generation (Liu et al. (2023a; 2024)). This field can be broadly categorized into two approaches: image-to-video and text-to-video generation. The former involves transforming a static image into a dynamic video, while the latter generates videos based solely on textual descriptions, without any input images. Pioneering methods in this domain include AnimateDiff (Guo et al. (2023)), Dynamicrafter (Xing et al. (2023)), Modelscope (Wang et al. (2023a)), AnimateAnything (Dai et al. (2023)), and Stable Video Diffusion (Rombach et al. (2022b)). These techniques typically leverage pre-trained text-to-image models, integrating them with various temporal mixing layers to handle the temporal dimension inherent in video data. However, the traditional U-Net based framework encounters scalability issues, limiting its ability to produce high-quality videos. The success of transformers in the natural language processing community and their scalability has prompted researchers to adapt this architecture for diffusion models, resulting in the development of DiTs (Peebles & Xie (2023). Subsequent work, such as Sora (Tim Brooks & Ramesh (2024)), has demonstrated the powerful capabilities of DiTs in video generation tasks. Open-source implementations like Latte (Ma et al. (2024)), Opensora (Zheng et al. (2024b)), and Opensora-Plan (Lab & etc. (2024)) have further validated the superior performance of DiT-based models over traditional U-Net structures in both text-to-video and image-to-video generation. Despite these advancements, the exploration of gaming video generation and its interactive controllability remains under-explored.

## A.2 GAME SIMULATION AND INTERACTION

Several pioneering works have attempted to train models for game simulation with action inputs. For example, UniSim (Yang et al. (2023)) and Pandora (Xiang et al. (2024)) built a diverse dataset of real-world and simulated videos and could predict a continuation video given a previous video segment and an action prompt via a supervised learning paradigm, while PVG (Menapace et al. (2021)) and Genie (Bruce et al. (2024)) focused on unsupervised learning of actions from videos. Similar to our work, GameGAN (Kim et al. (2020)), GameNGen (Valevski et al. (2024)) and DIA-MOND (Alonso et al. (2024)) focused on the playable simulation of early games such as Atari and DOOM, and demonstrates its combination with a gaming agent for interaction (Zheng et al. (2024a)). Recently, Oasis (Decart (2024)) simulated Minecraft at a real-time level, including both the footage and game system via the diffusion model while GameFactory (Yu et al. (2025)) brought the game domain manipulation to real-world prior. However, they didn't explore the potential of generative models in simulating the complex environments of next-generation games. Instead, GameGen-$\mathbb{X}$ can create intricate environments, dynamic events, diverse characters, and complex actions with a high degree of realism and variety. Additionally, GameGen-$\mathbb{X}$ allows the model to generate subsequent frames based on the current video segment and player-provided multi-modal control signals. This approach ensures that the generated content is not only visually compelling but also contextually appropriate and responsive to player actions, bridging the gap between simple game simulations and the sophisticated requirements of next-generation open-world games.

# B DATASET

## B.1 DATA AVAILABILITY STATEMENT AND CLARIFICATION

We are committed to maintaining transparency and compliance in our data collection and sharing methods. Please note the following:

- **Publicly Available Data**: The data utilized in our studies is publicly available. We do not use any exclusive or private data sources.

- **Data Sharing Policy:** Our data sharing policy aligns with precedents set by prior works, such as InternVid (Wang et al. (2023c)), Panda-70M (Chen et al. (2024)), and Miradata (Ju et al. (2024)). Rather than providing the original raw data, we only supply the YouTube video IDs necessary for downloading the respective content.

- **Usage Rights:** The data released is intended exclusively for research purposes. Any potential commercial usage is not sanctioned under this agreement.

- **Compliance with YouTube Policies:** Our data collection and release practices strictly adhere to YouTube's data privacy policies and fair of use policies. We ensure that no user data or privacy rights are violated during the process.

- **Data License:** The dataset is made available under the Creative Commons Attribution 4.0 International License (CC BY 4.0).

Moreover, the OGameData dataset is only available for informational purposes only. The copyright remains with the original owners of the video. All videos of the OGameData datasets are obtained from the Internet which is not the property of our institutions. Our institution is not responsible for the content or the meaning of these videos. Related to the future open-sourcing version, the researchers should agree not to reproduce, duplicate, copy, sell, trade, resell, or exploit for any commercial purposes, any portion of the videos, and any portion of derived data, and not to further copy, publish, or distribute any portion of the OGameData dataset.

## B.2 CONSTRUCTION DETAILS

**Data Collection.** Following Ju et al. (2024) and Chen et al. (2024), we selected online video websites and local game engines as one of our primary video sources. Prior research predominantly focused on collecting game cutscenes and gameplay videos containing UI elements. Such videos are not ideal for training a game video generation model due to the presence of UI elements and non-playable content. In contrast, our method adheres to the following principles: 1) We exclusively collect videos showcasing playable content, as our goal is to generate actual gameplay videos rather than cutscenes or CG animations. 2) We ensure that the videos are high-quality and devoid of any UI elements. To achieve this, we only include high-quality games released post-2015 and capture some game footage directly from game engines to enhance diversity. Following the Internet data collection stage, we collected 32,000 videos from YouTube, which cover more than 150 next-generation video games. Additionally, we recorded the gameplay videos locally, to collect the keyboard control signals. We purchased games on the Steam platform to conduct our instruction data collection. To accurately simulate the in-game lighting and weather effects, we parsed the game's console functions and configured the weather and lighting change events to occur randomly every 5-10 seconds. To emulate player input, we developed a virtual keyboard that randomly controls the character's movements within the game scenes. Our data collection spanned multiple distinct game areas, resulting in nearly 100 hours of recorded data. The program meticulously logged the output signals from the virtual keyboard, and we utilized Game Bar to capture the corresponding gameplay footage. This setup allowed us to synchronize the keyboard signals with frame-level data, ensuring precise alignment between the input actions and the visual output.

**Video-level Selection and Annotation.** Despite our rigorous data collection process, some low-quality videos inevitably collected into our dataset. Additionally, the collected videos lack essential metadata such as game name, genre, and player perspective. This metadata is challenging to annotate using AI alone. Therefore, we employed human game experts to filter and annotate the videos. In this stage, human experts manually review each video, removing those with UI elements or non-playable content. For the remaining usable videos, they annotate critical metadata, including game name, genre (e.g., ACT, FPS, RPG), and player perspective (First-person, Third-person). After this filtering and annotation phase, we curated a dataset of 15,000 high-quality videos complete with game metadata.

**Scene Detection and Segmentation.** The collected videos, ranging from several minutes to hours, are unsuitable for model training due to their extended duration and numerous scene changes. We employed TransNetV2 (Souček & Lokoč (2020)) and PyScene for scene segmentation, which can adaptively identify scene change timestamps within videos. Upon obtaining these timestamps, we discard video clips shorter than 4 seconds, considering them too brief. For clips longer than 16 seconds, we divide them into multiple 16-second segments, discarding any remainder shorter than 4 seconds. Following this scene segmentation stage, we obtained around 1,000,000 video clips, each containing 4-16 seconds of content at 24 frames per second.

**Clips-level Filtering and Annotation.** Some clips contain game menus, maps, black screens, low-quality scenes, or nearly static scenes, necessitating further data cleaning. Given the vast number

of clips, manual inspection is impractical. Instead, we sequentially employed an aesthetic scoring model, a flow scoring model, the video CLIP model, and a camera motion model for filtering and annotation. First, we used the CLIP-AVA model (Schuhmann (2023)) to score each clip aesthetically. We then randomly sampled 100 clips to manually determine a threshold, filtering out clips with aesthetic scores below this threshold. Next, we applied the UniMatch model (Xu et al. (2023)) to filter out clips with either excessive or minimal motion. To address redundancy, we used the video-CLIP (Xu et al. (2021)) model to calculate content similarity within clips from the same game, removing overly similar clips. Finally, we utilized CoTrackerV2 (Karaev et al. (2023)) to annotate clips with camera motion information, such as "pan-left" or "zoom-in."

**Structural Caption.** We propose a Structural captioning approach for generating captions for OGameData-GEN and OGameData-INS. To achieve this, we uniformly sample 8 frames from each video and stack them into a single image. Using this image as a representation of the video's content, we designed two specific prompts to instruct GPT-4o to generate captions. For OGameData-GEN, we have GPT-4o describe the video across five dimensions: Summary of the video, Game Meta information, Character details, Frame Descriptions, and Game Atmosphere. This Structural information enables the model to learn mappings between text and visual information during training and allows us to independently modify one dimension's information while keeping the others unchanged during the inference stage. For OGameData-INS, we decompose the video changes into five perspectives, with each perspective described in a short sentence. The Environment Basic dimension describes the fundamental environment information, while the Transition dimension captures changes in the environment. The Light and Act dimensions describe the lighting conditions and character actions, respectively. Lastly, the MISC dimension includes meta-information about the video, such as keyboard operations or camera motion. This Structural captioning approach allows the model to focus entirely on content changes, thereby enhancing control over the generated video. By enabling independent modification of specific dimensions during inference, we achieve fine-grained generation and control, ensuring the model effectively captures both static and dynamic aspects of the game world.

**Prompt Design.** In our collection of 32,000 videos, we identified two distinct categories. The first category comprises free-camera videos, which primarily focus on showcasing environmental and scenic elements. The second category consists of gameplay videos, characterized by the player's perspective during gameplay, including both first-person and third-person views. We believe that free-camera videos can help the model better align with engine-specific features, such as textures and physical properties, while gameplay videos can directly guide the model's behavior. To leverage these two types of videos effectively, we designed different sets of prompts for each category. Each set includes a summary prompt and a dense prompt. The core purpose of the summary prompt is to succinctly describe all the scene elements in a single sentence, whereas the dense prompt provides structural, fine-grained guidance. Additionally, to achieve interactive controllability, we designed structural instruction prompts. These prompts describe the differences between the initial frame and subsequent frames across various dimensions, simulating how instructions can guide the generation of follow-up video content.

```
1   prompt_summry = '''You are ChatGPT, a large language model trained by OpenAI, based on the GPT-4
        architecture.
2       Knowledge cutoff: 2023-10.
3       Current date: 2024-05-15.
4       Image input capabilities: Enabled.
5       Personality: v2.
6       # Character
7       You are a video game environment captioning assistant that generates concise descriptions of game
            environment.
8       # Skills
9       - Analyzing a sequence of 8 images that represent a game environment
10      - Identifying key environmental features and atmospheric elements
11      - Generating a brief, coherent caption that captures the main elements of the game world
12      # Constraints
13      - The caption should be no more than 20 words long
14      - The caption must describe the main environmental features visible
15      - The caption must include the overall atmosphere or mood of the setting
16      - Use present tense to describe the environment
17      # Input: [8 sequential frames of the game environment, arranged in 2 rows of 4 images each]
18      # Output: [A concise, English caption describing the main features and atmosphere of the game
            environment]
19      # Example: A misty forest surrounds ancient ruins, with towering trees and crumbling stone structures
            creating a mysterious atmosphere.'''
```

Listing 1: Summary prompt for free-camera videos

```
1  prompt_summry = '''You are ChatGPT, a large language model trained by OpenAI, based on the GPT-4
       architecture.
2      Knowledge cutoff: 2023-10.
3      Current date: 2024-05-15.
4      Image input capabilities: Enabled.
5      Personality: v2.
6      # Character
7      You are a highly skilled video game environment captioning AI assistant. Your task is to generate a
           detailed, dense caption for a game environment based on 8 sequential frames provided as input.
           The caption should comprehensively describe the key elements of the game world and setting.
8      # Skills
9      - Identifying the style and genre of the video game
10     - Recognizing and describing the main environmental features and landscapes
11     - Detailing the atmosphere, lighting, and overall mood of the setting
12     - Noting key architectural elements, structures, or natural formations
13     - Describing any notable weather effects or environmental conditions
14     - Synthesizing the 8 frames into a cohesive description of the game world
15     - Using vivid and precise language to paint a detailed picture for the reader
16     # Constraints
17     - The input will be a single image containing 8 frames of the game environment, arranged in two rows
           of 4 frames each
18     - The output should be a single, dense caption of 2-4 sentences covering the entire environment shown
19     # Background
20     - This video is from GAME ID.
21     # The caption must mention:
22     - The main environmental features that are the focus of the frames
23     - The overall style or genre of the game world (e.g. fantasy, sci-fi, post-apocalyptic)
24     - Key details about the landscape, vegetation, and terrain
25     - Any notable structures, ruins, or settlements visible
26     - The general atmosphere, time of day, and weather conditions
27     - Use concise yet descriptive language to capture the essential elements
28     - The change of environment in these frames
29     - Avoid speculating about areas not represented in the 8 frames'''
```

Listing 2: Dense prompt for free-camera videos

```
1  prompt_summry = ''' You are ChatGPT, a large language model trained by OpenAI, based on the GPT-4
       architecture.
2      Knowledge cutoff: 2023-10.
3      Current date: 2024-05-15.
4      Image input capabilities: Enabled.
5      Personality: v2.
6      # Character
7      You are a video captioning assistant that generates concise descriptions of short video clips.
8      # Skills
9      Analyzing a sequence of 8 images that represent a short video clip
10     If it is a third-person view, identify key characters and their actions, else, identify key objects
           and environments.
11     Generating a brief, coherent caption that captures the main elements of the video
12     # Constraints
13     - The caption should be no more than 20 words long
14     - If it is a third-person view, the caption must include the main character(s) and their action(s)
15     - The caption must describe the environment shown in the video
16     - Use present tense to describe the actions
17     - If there are multiple distinct actions, focus on the most prominent one
18     # Input: [8 sequential frames of the video, arranged in 2 rows of 4 images each]
19     # Output: [A concise, English caption describing the main character(s) and action(s) in the video]
20     # Example: There is a person walking on a path surrounded by trees and ruins of an ancient city.'''
```

Listing 3: Summary prompt for gameplay videos

```
1  prompt_summry = '''You are ChatGPT, a large language model trained by OpenAI, based on the GPT-4
       architecture.
2      Knowledge cutoff: 2023-10.
3      Current date: 2024-05-15.
4      Image input capabilities: Enabled.
5      Personality: v2.
6      # Character
7      You are a highly skilled video captioning AI assistant. Your task is to generate a detailed, dense
           caption for a short video clip based on 8 sequential frames provided as input. The caption
           should comprehensively describe the key elements of the video.
8      # Skills
9      - Identifying the style and genre of the video game footage
10     - Recognizing and naming the main object or character in focus
11     - Describing the background environment and setting
```

```
12          - Noting key camera angles, movements, and shot types
13          - Synthesizing the 8 frames into a cohesive description of the video action
14          - Using vivid and precise language to paint a detailed picture for the reader
15          # Constraints
16          - The input will be a single image containing 8 frames of the video, arranged in two rows of 4 frames
                each, in sequential order
17          - The output should be a single, dense caption of 2-6 sentences covering the entire 8-frame video
18          - The caption should be no more than 200 words long
19          # Background
20          - This video is from GAME ID.
21          ## The caption must mention:
22          - The main object or character that is the focus of the video
23          - If it is a third-person view, include the name of the main character, the appearance, clothing, and
                anything related to the character generation guidance.
24          - The game style or genre (e.g. first-person/third-person, shooter, open-world, racing, etc.)
25          - Key details about the background environment and setting
26          - Any notable camera angles, movements, or shot types
27          - Use concise yet descriptive language to capture the essential elements
28          - Avoid speculating about parts of the video not represented in the 8 frames'''
```

Listing 4: Dense prompt for gameplay videos

```
1   prompt_summry = '''You are ChatGPT, a large language model trained by OpenAI, based on the GPT-4
        architecture.
2       Knowledge cutoff: 2023-10.
3       Current date: 2024-05-15.
4       Image input capabilities: Enabled.
5       Personality: v2.
6       # Character
7       You are a highly skilled AI assistant specializing in detecting and describing changes in video
            sequences. Your task is to analyze 8 sequential frames from a video and generate a concise
            Structural caption focusing on the action and changes that occur after the first frame.  This
            Structural caption will be used to train a video generation model to create controllable video
            sequences based on textual commands.
8       # Skills
9       - Carefully observing the first frame to establish a baseline, comparing subsequent content to the
            first frame
10      - Please describe the input video in the following 4 dimensions, providing a single, concise
            instructional sentence for each:
11      1. Environmental Basics: Describe what the whole scene looks like.
12      2. Main Character: Direct the protagonist's actions and movements.
13      3. Environmental Changes: Command how the scene should change over time.
14      4. Sky/Lighting: Instruct on how to adjust sky conditions and lighting effects.
15      # Constraints
16      - The input will be a single image containing 8 frames of the video, arranged in two rows of 4 frames
            each, in sequential order
17      - Focus solely on changes that occur after the first frame
18      - Do not describe elements that remain constant throughout the sequence
19      - Use clear, precise language to describe the changes
20      - Frame each dimension as a clear, actionable instruction.
21      - Keep each instruction to one sentence only, each sentence should be concise and no more than 15
            words.
22      - Use imperative language suitable for directing a video generation model.
23      - If information for a particular dimension is not available, provide a general instruction like
            'Maintain current state' for that dimension.
24      - Do not include numbers or bullet points before each sentence in the output.
25      - Please use simple words.
26      # Instructions
27      - Examine the first frame carefully as a baseline
28      - Analyze the subsequent content as a continuous video sequence
29      - Avoid using terms like "frame," "image," or "figure" in your description
30      - Describe the sequence as if it were a continuous video, not separate frames
31      # Background
32      - This video is from GAME ID. Focus on describing the changes in action, environment, or character
            positioning rather than identifying specific game elements.
33      # Output
34      - Your output should be a list, with each number corresponding to the dimension as listed above. For
            example:
35      Environmental Basics: [Your Instruction for Environmental Basics].
36      Main Character: [Your Instruction for Main Character].
37      Environmental Changes: [Your Instruction for Environmental Changes].
38      Sky/Lighting: [Your Instruction for Sky/Lighting].
39
40      Please process the input and provide the Structural, instructional output:'''
```

Listing 5: Instruction prompt for interactive control

## B.3 DATASET SHOWCASES

We provide a visualization of the video clips along with their corresponding captions. We sampled four cases from the OGameData-GEN dataset and the OGameData-INS dataset, respectively. Both

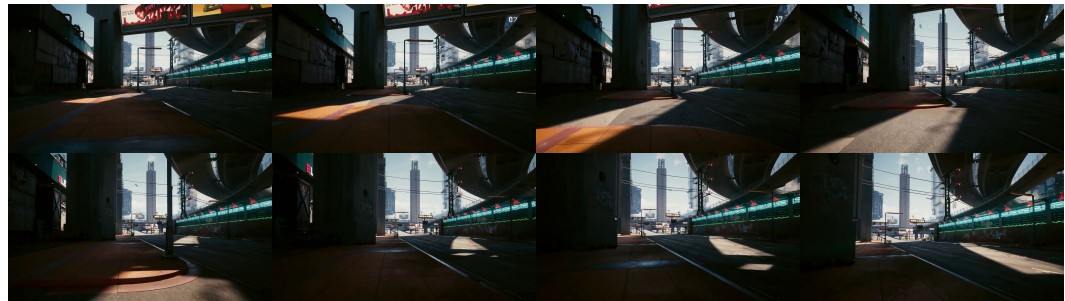

Figure 9: A sample from OGameData-GEN. Caption: An empty futuristic city street is seen under a large overpass with neon lights and tall buildings. In this sequence from Cyberpunk 2077, the scene unfolds in a sprawling urban environment marked by towering skyscrapers and elevated highways. The video clip showcases a first-person perspective that gradually moves forward along an empty street framed by futuristic neon-lit buildings on the right and industrial structures on the left. The atmospheric lighting casts dramatic shadows across the pavement, enhancing the gritty cyberpunk aesthetic of Night City. As the camera progresses smoothly towards a distant structure adorned with holographic advertisements, it captures key details like overhead cables and a monorail track above, highlighting both verticality and depth in this open-world dystopian setting devoid of any characters or vehicles at this moment. The scene emphasizes a gritty, dystopian cyberpunk atmosphere, characterized by neon-lit buildings, dramatic shadows, and a sense of desolate futurism devoid of characters or vehicles.'

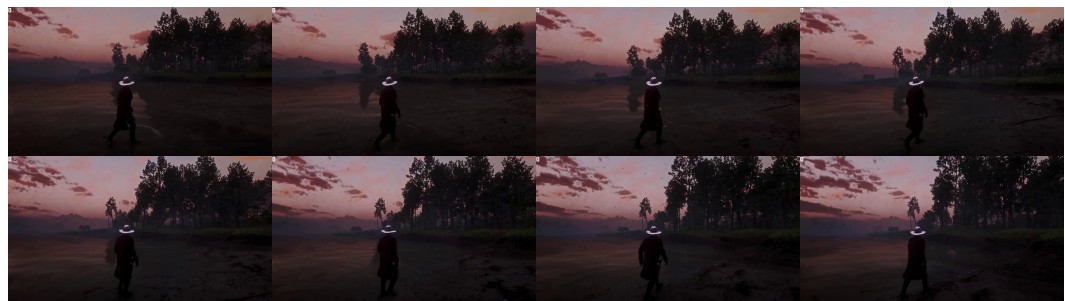

Figure 10: A sample from OGameData-GEN. Caption: A person in a white hat walks along a forested riverbank at sunset. In the dim twilight of a picturesque, wooded lakeshore in Red Dead Redemption 2, Arthur Morgan, dressed in his iconic red coat and wide-brimmed white hat, strides purposefully along the water's edge. The eight sequential frames capture him closely from behind at an over-the-shoulder camera angle as he walks towards the dense tree line under a dramatic evening sky tinged with pink and purple hues. Each step takes place against a tranquil backdrop featuring rippling water reflecting dying sunlight and silhouetted trees that deepen the serene yet subtly ominous atmosphere typical of this open-world action-adventure game. Dust particles float visibly through the air as Arthur's movement stirs up small puffs from the soil beneath his boots, adding to the immersive realism of this richly detailed environment. The scene captures a tranquil yet subtly ominous atmosphere.

types of captions are Structural, offering multidimensional annotations of the videos. This Structural approach ensures a comprehensive and nuanced representation of the video content.

**Structural Captions in OGameData-GEN.** It is evident from Fig. 9 and Fig. 10 that the captions in OGameData-GEN densely capture the overall information and intricate details of the videos, following the sequential set of 'Summary', 'Game Meta Information', 'Character Information', 'Frame Description', and 'Atmosphere'.

**Structural Instructions in OGameData-INS.** In contrast, the instructions in OGameData-INS, which are instruction-oriented and often use imperative sentences, effectively capture the changes in subsequent frames relative to the initial frame, as shown in Fig. 11 and Fig. 12. It has five decou-

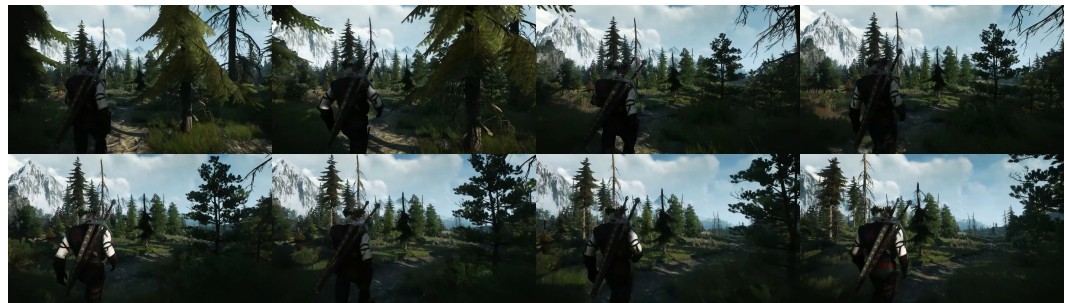

Figure 11: A sample from OGameData-INS. Caption: Environmental Basics: Maintain the dense forest scenery with mountains in the distant background. Main Character: Move forward along the path while maintaining a steady pace. Environmental Changes: Gradually clear some trees and bushes to reveal more of the landscape ahead. Sky/Lighting: Keep consistent daylight conditions with scattered clouds. aesthetic score: 5.02, motion score: 27.37, camera motion: Undetermined. camera size: full shot.

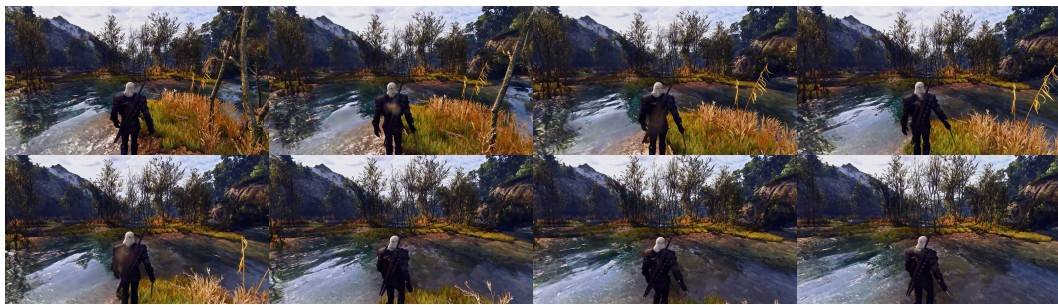

Figure 12: A sample from OGameData-INS. Caption: Environmental Basics: Show a scenic outdoor environment with trees, grass, and a clear water body in the foreground. Main Character: Move the protagonist slowly forward towards the right along the water's edge. Environmental Changes: Maintain current state without significant changes to background elements. Sky/Lighting: Keep sky conditions bright and lighting consistent throughout. aesthetic score: 5.36, motion score: 9.37, camera motion: zoom in. camera size: full shot".

pled dimensions following the sequential of 'Environment Basic', 'Character Action', 'Environment Change', 'Lighting and Sky', and 'Misc'.

## B.4 QUANTITATIVE ANALYSIS

To demonstrate the intricacies of our proposed dataset, we conducted a comprehensive analysis encompassing several key aspects. Specifically, we examined the distribution of game types, game genres, player viewpoints, motion scores, aesthetic scores, caption lengths, and caption feature distributions. Our analysis spans both the OGameData-GEN dataset and the OGameData-INS dataset, providing detailed insights into their respective characteristics.

**Game-related Data Analysis.** Our dataset encompasses a diverse collection of 150 games, with a primary focus on next-generation open-world titles. For the OGameData-GEN dataset, as depicted in Fig. 13, player perspectives are evenly distributed between first-person and third-person viewpoints. Furthermore, it includes a wide array of game genres, including RPG, Action, Simulation, and FPS, thereby showcasing the richness and variety of the dataset. In contrast, the OGameData-INS dataset, as shown in Fig. 14, is composed of five meticulously selected high-quality open-world games, each characterized by detailed and dynamic character motion. Approximately half of the videos feature the main character walking forward (zooming in), while others depict lateral movements such as moving right or left. These motion patterns enable us to effectively train an instructive network. To ensure the model's attention remains on the main character, we exclusively selected third-person perspectives.

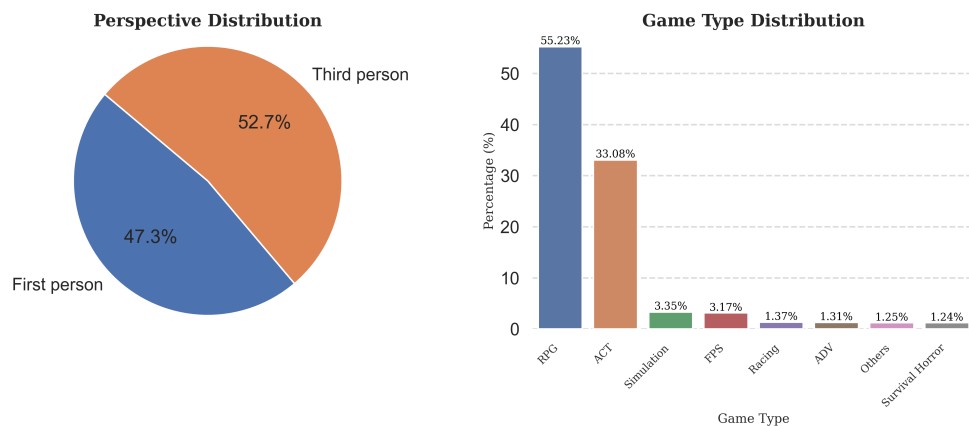

Figure 13: Statistical analysis of the OGameData-GEN dataset. The left pie chart illustrates the distribution of player perspectives, with 52.7% of the games featuring a third-person perspective and 47.3% featuring a first-person perspective. The right bar chart presents the distribution of game types, demonstrating a predominance of RPG (55.23%) and ACT (33.08%) genres, followed by Simulation (3.35%) and FPS (3.17%), among others.

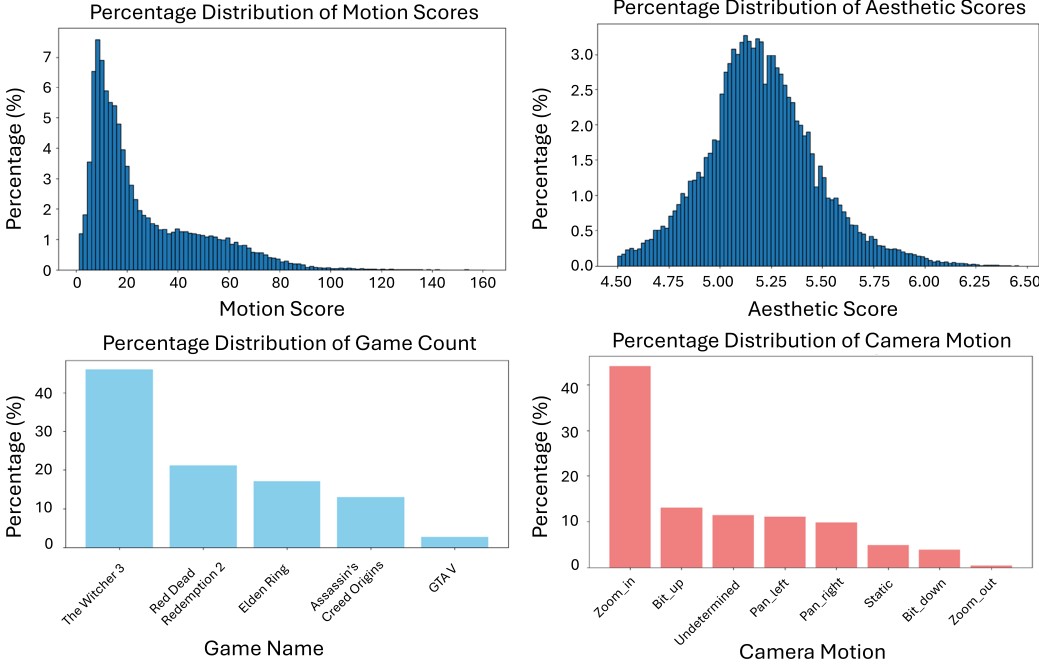

Figure 14: Comprehensive analysis of the OGameData-INS dataset. The top-left histogram shows the distribution of motion scores, with most scores ranging from 0 to 100. The top-right histogram illustrates the distribution of aesthetic scores, following a Gaussian distribution with the majority of scores between 4.5 and 6.5. The bottom-left bar chart presents the game count statistics, highlighting the most frequently occurring games. The bottom-right bar chart displays the camera motion statistics, with a significant portion of the clips featuring zoom-in motions, followed by various other camera movements.

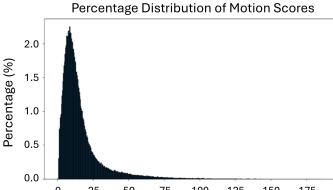 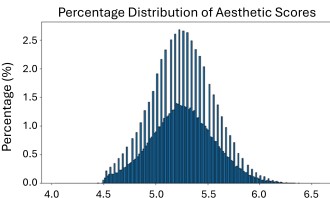 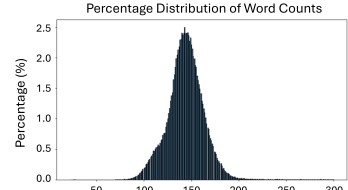

Figure 15: Clip-related data analysis for the OGameData-GEN dataset. The left histogram shows the distribution of motion scores, with most scores ranging from 0 to 75. The middle histogram displays the distribution of aesthetic scores, following a Gaussian distribution with the majority of scores between 4.5 and 6. The right histogram illustrates the distribution of word counts in captions, predominantly ranging between 100 and 200 words. This detailed analysis highlights the rich and varied nature of the clips and their annotations, providing comprehensive information for model training.

**Clips-related Data Analysis.** Apart from the game-related data analysis, we also conducted clip-related data analysis, encompassing metrics such as motion score, aesthetic score, and caption distribution. This analysis provides clear insights into the quality of our proposed dataset. For the OGameData-GEN dataset, as illustrated in Fig. 15, most motion scores range from 0 to 75, while the aesthetic scores follow a Gaussian distribution, with the majority of scores falling between 4.5 and 6. Furthermore, this dataset features dense captions, with most captions containing between 100 to 200 words, providing the model with comprehensive game-related information. For the OGameData-INS dataset, as shown in Fig. 14, the aesthetic and motion scores are consistent with those of the OGameData-GEN dataset. However, the captions in OGameData-INS are significantly shorter, enabling the model to focus more on the instructional content itself. This design choice ensures that the model prioritizes the instructional elements, thereby enhancing its effectiveness in understanding and executing tasks based on the provided instructions.

## C IMPLEMENTATION AND DESIGN DETAILS

### C.1 TRAINING STRATEGY

We adopted a two-phase training strategy to build our model. In the first phase, our goal was to train a foundation model capable of both video continuation and generation. To achieve this, we allocated 75% of the training probability to text-to-video generation tasks and 25% to video extension tasks. This approach allowed the model to develop strong generative abilities while also building a solid foundation for video extension.

To enhance the model's ability to handle diverse scenarios, we implemented a bucket-based sampling strategy. Videos were sampled across a range of resolutions (480p, 512×512, 720p, and 1024×1024) and durations (from single frames to 480 frames at 24 fps), as shown in Table 6. For example, 1024×1024 videos with 102 frames had an 8.00% sampling probability, while 480p videos with 408 frames were sampled with an 18.00% probability. This approach ensured the model was exposed to both short and long videos with different resolutions, preparing it for a wide variety of tasks. For longer videos, we extracted random segments for training. All videos were resized and center-cropped to meet resolution requirements before being processed through a 3D VAE, which compressed spatial dimensions by 8× and temporal dimensions by 4×, reducing computational costs significantly.

We employed several techniques to optimize training and improve output quality. Rectified flow (Liu et al. (2023b)) was used to accelerate training and enhance generation accuracy. The Adam optimizer with a fixed learning rate of 5e-4 was applied for 20 epochs. Additionally, we followed common practices in diffusion models by randomly dropping text inputs with a 25% probability to strengthen the model's generative capabilities Ho & Salimans (2021).

After completing the first training phase, we froze the base model and shifted our focus to training an additional branch, InstructNet, in the second phase. This phase concentrated entirely on the video extension task, with a 100% probability assigned to this task. Unlike the first phase, we abandoned

Table 6: Video Sampling Probabilities by Resolution and Frame Count

| Resolution | Number of Frames | Sampling Probability (%) |
|---|---|---|
| 1024×1024 | 102 | 8.00 |
| 1024×1024 | 51 | 1.80 |
| 1024×1024 | 1 | 2.00 |
| 480p | 204 | 6.48 |
| 480p | 408 | 18.00 |
| 480p | 89 | 6.48 |
| 720p | 102 | 54.00 |
| 512×512 | 51 | 3.24 |

the bucket-based sampling strategy and instead used videos with a fixed resolution of 720p and a duration of 4 seconds. To enhance control over the video extension process, we introduced additional conditions through InstructNet. In 20% of the samples, no control conditions were applied, allowing the model to generate results freely. For the remaining 80% of the samples, control conditions are included with the following probabilities: 30% of the time, both text and keyboard signals are provided as control; 30% of the time, only text is provided; and for another 30%, both text and a video prompt are used as control. In the remaining 10% of cases, all three control conditions—text, keyboard signals, and video prompts—are applied simultaneously. When video prompts are incorporated, we sample from a set of different prompt types with equal probability, including canny-edge videos, motion vector videos, and pose sequence videos. In both phases of training, during video extension tasks, we retain the first frame of latent as a reference for the model.

## C.2 MODEL ARCHITECTURE

Regarding the model architecture, our framework comprises four primary components: a 3D VAE for video compression, a T5 model for text encoding, the base model, and InstructNet.

**3D VAE.** We extended the 2D VAE architecture from Stable Diffusion Stability AI (2024) by incorporating additional temporal layers to compress temporal information. Multiple layers of Causal 3D CNN Yu et al. (2023b) were implemented to compress inter-frame information. T he VAE decoder maintains architectural symmetry with the encoder. Our 3D VAE effectively compresses videos in both spatial and temporal dimensions, specifically reducing spatial dimensions by a factor of 8 and temporal dimensions by a factor of 4.

**Text Encoder.** We employed the T5 model Raffel et al. (2020b) with a maximum sequence length of 300 tokens to accommodate our long-form textual inputs.

**Masked Spatial-Temporal Diffusion Transformer.** Our MSDiT is composed of stacked Spatial Transformer Blocks and Temporal Transformer Blocks, along with an initial embedding layer and a final layer that reorganizes the serialized tokens back into 2D features. Overall, our MSDiT consists of 28 layers, with each layer containing both a spatial and temporal transformer block, in addition to the embedding and final layers. Starting with the embedding layer, this layer first compresses the input features further, specifically performing a 2x downsampling along the height and width dimensions to transform the spatial features into tokens suitable for transformer processing. The resulting latent representation $z$, is augmented with various meta-information such as the video's aspect ratio, frame count, timesteps, and frames per second (fps). These metadata are projected into the same channel dimension as the latent feature via MLP layers and directly added to $z$, resulting in $z'$. Next, $z'$ is processed through the stack of Spatial Transformer Blocks and Temporal Transformer Blocks, after which it is decoded back into spatial features. Throughout this process, the latent channel dimension is set to 1152. For the transformer blocks, we use 16 attention heads and apply several techniques such as query-key normalization (QK norm) (Henry et al. (2020)) and rotary position embeddings (RoPE) (Su et al. (2024)) to enhance the model's performance. Additionally, we leverage masking techniques to enable the model to support both text-to-video generation and video extension tasks. Specifically, we unmask the frames that the model should condition on during video extension tasks. In the forward pass of the base model, unmasked frames are assigned a timestep value of 0, while the remaining frames retain their original timesteps. The pseudo-codes

of our feature processing pipeline and the Masked Temporal Transformer block are shown in the following.

```
class BaseModel:
    initialize(config):
        # Step 1: Set base configurations
        set pred_sigma, in_channels, out_channels, and model depth
    based on config
        initialize hidden size and positional embedding parameters

        # Step 2: Define embedding layers
        create patch embedder for input
        create timestep embedder for temporal information
        create caption embedder for auxiliary input
        create positional embeddings for spatial and temporal contexts

        # Step 3: Define processing blocks
        create spatial blocks for frame-level operations
        create temporal blocks for sequence-level operations

        # Step 4: Define final output layer
        initialize the final transformation layer to reconstruct
    output

    function forward(x, timestep, y, mask=None, x_mask=None, fps=None,
     height=None, width=None):
        # Step 1: Prepare inputs
        preprocess x, timestep, and y for model input

        # Step 2: Compute positional embeddings
        derive positional embeddings based on input size and dynamic
    dimensions

        # Step 3: Compute timestep and auxiliary embeddings
        encode timestep information
        encode auxiliary input (e.g., captions) if provided

        # Step 4: Embed input video
        apply spatial and temporal embeddings to video input

        # Step 5: Process through spatial and temporal blocks
        for each spatial and temporal block pair:
            apply spatial block to refine frame-level features
            apply temporal block to model dependencies across frames

        # Step 6: Finalize output
        transform processed features to reconstruct the output

        return final output
```

```
class TemporalTransformerBlock:
    initialize(hidden_size, num_heads):
        set hidden_size
        create TemporalAttention with hidden_size and num_heads
        create LayerNorm with hidden_size

    function t_mask_select(x_mask, x, masked_x, T, S):
        reshape x to [B, T, S, C]
        reshape masked_x to [B, T, S, C]
        apply mask: where x_mask is True, keep values from x;
    otherwise, use masked_x
```

```
11        reshape result back to [B, T * S, C]
12        return result
13
14    function forward(x, x_mask=None, T=None, S=None):
15        set x_m to x (modulated input)
16
17        if x_mask is not None:
18            create masked version of x with zeros
19            replace x with masked_x using t_mask_select
20
21        apply attention to x_m
22
23        if x_mask is not None:
24            reapply mask to output using t_mask_select
25
26        add residual connection (x + x_m)
27        apply layer normalization
28        return final output
```

**InstructNet** Our InstructNet consists of 28 InstructNet Blocks, alternating between Spatial and Temporal Attention mechanisms, with each type accounting for half of the total blocks. The attention mechanisms and dimensionality in InstructNet Blocks maintain consistency with the base model. The InstructNet Block incorporates textual instruction information through an Instruction Fusion Expert utilizing cross-attention, while keyboard operations are integrated via an Operation Fusion Expert through feature modulation. Keyboard inputs are initially projected into one-hot encodings, and then transformed through an MLP to match the latent feature dimensionality. The resulting keyboard features are processed through an additional MLP to predict affine transformation parameters, which are subsequently applied to modify the latent features. Video prompts are incorporated into InstructNet through additive fusion at the embedding layer.

### C.3 COMPUTATION RESOURCES AND COSTS

Regarding computational resources, our training infrastructure consisted of 24 NVIDIA H800 GPUs distributed across three servers, with each server hosting 8 GPUs equipped with 80GB of memory per unit. We implemented distributed training across both machines and GPUs, leveraging Zero-2 optimization to reduce computational overhead. The training process was structured into two phases: the base model training, which took approximately 25 days, and the InstructNet training phase, completed in 7 days. For storage, we utilized approximately 50TB to accommodate the dataset and model checkpoints.

## D   EXPERIMENT DETAILS AND FURTHER ANALYSIS

### D.1   FAIRNESS STATEMENT AND CONTRIBUTION DECOMPOSITION

In our experiments, we compared four models (OpenSora-Plan, OpenSora, MiraDiT, and CogVideo-X) and five commercial models (Gen-2, Kling 1.5, Tongyi, Pika, and Luma). OpenSora-Plan, OpenSora, and MiraDiT explicitly state that their training datasets (Panda-70M, MiraData) include a significant amount of 3D game/engine-rendered scenes. This makes them suitable baselines for evaluating game content generation. Additionally, while CogVideo-X and commercial models do not disclose training data, their outputs suggest familiarity with similar visual domains. Therefore, the comparisons are fair in the context of assessing game content generation capabilities. To address concerns about potential overlap between training and test data, we ensured that the test set included only content types not explicitly present in the training set.

Additionally, to disentangle the effects of data and framework design, we sampled 10K subsets from both MiraData (which contain high-quality game video data) and OGameData and conducted a set of ablation experiments with OpenSora (a state-of-the-art open-sourced video generation framework). The results are as follows:

Table 7: The decomposition of contributions from OGameData and model design

| Model | FID | FVD | TVA | UP | MS | DD | SC | IQ |
|---|---|---|---|---|---|---|---|---|
| Ours w/ OGameData | 289.5 | 1181.3 | 0.83 | 0.67 | 0.99 | 0.64 | 0.95 | 0.49 |
| OpenSora w/ OGameData | 295.0 | 1186.0 | 0.70 | 0.48 | 0.99 | 0.84 | 0.93 | 0.50 |
| Ours w/ MiraData | 303.7 | 1423.6 | 0.57 | 0.30 | 0.98 | 0.96 | 0.91 | 0.53 |

As shown in the table above, we supplemented a comparison with OpenSora on MiraData. In comparing Domain Alignment Metrics(averaged FID and FVD scores) and Visual Quality Metrics (averaged TVA, UP, MS, DD, SC, and IQ scores), our framework and dataset demonstrate clear advantages. Aligning the dataset (row 1 and row 2), it can be observed that our framework (735.4, 0.76) outperforms the OpenSora framework (740.5, 0.74), indicating the advantage of our architecture design. Additionally, fixing the framework, the model training on the OGameData (735.4, 0.76) surpasses the model training on MiraData (863.65, 0.71), highlighting our dataset's superiority in the gaming domain. These results confirm the efficacy of our framework and the significant advantages of our dataset.

## D.2 Experimental Settings

In this section, we delve into the details of our experiments, covering the calculation of metrics, implementation details, evaluation datasets, and the details of our ablation study.

**Evaluation Benchmark.** To evaluate the performance of our methods and other benchmark methods, we constructed two evaluation datasets: OGameEval-Gen and OGameEval-Ins. The OGameEval-Gen dataset contains 50 text-video pairs sampled from the OGameData-GEN dataset, ensuring that these samples were not used during training. For a fair comparison, the captions were generated using GPT-4o. For the OGameEval-Ins dataset, we sampled the last frame of ten videos from the OGameData-INS eval dataset, which were also unused during training. We generated two types of instructional captions for each video: character control (e.g., move-left, move-right) and environment control (e.g., turn to rainy, turn to sunny, turn to foggy, and create a river in front of the main character). Consequently, we have 60 text-video pairs for evaluating control ability. To ensure a fair comparison, for each instruction, we utilized GPT-4o to generate two types of captions: Structural instructions to evaluate our methods and dense captions to evaluate other methods.

**Metric Details.** To comprehensively evaluate the performance of GameGen-$\mathbb{X}$, we utilize a suite of metrics that capture various aspects of video generation quality and interactive control. This implementation is based on VBench (Huang et al. (2024b)) and CogVideoX (Yang et al. (2024)). By employing this set of metrics, we aim to provide a comprehensive evaluation of GameGen-$\mathbb{X}$'s capabilities in generating high-quality, realistic, and interactively controllable video game content. The details are following:

*FID (Fréchet Inception Distance) (Heusel et al. (2017))*: Measures the visual quality of generated frames by comparing their distribution to real frames. Lower scores indicate better quality.

*FVD (Fréchet Video Distance) (Rakhimov et al. (2020))*: Assesses the temporal coherence and overall quality of generated videos. Lower scores signify more realistic and coherent video sequences.

*UP (User Preference) (Yang et al. (2024))*: In alignment with the methods of CogVideoX, we implemented a single-blind study to evaluate video quality. The final quality score for each video is the average of evaluations from all ten experts. The details are shown in Table 8.

*TVA (Text-Video Alignment) (Yang et al. (2024))*: Following the evaluation criteria established by CogVideoX, we conducted a single-blind study to assess text-video alignment. The final quality score for each video is the average of evaluations from all ten experts. The details are shown in Table 9.

*SR (Success Rate)*: We assess the model's control capability through a collaboration between humans and AI, calculating a success rate. The final score is the average, and higher scores reflect models with greater control precision.

*MS (Motion Smoothness) (Huang et al. (2024b))*: Measures the fluidity of motion in the generated videos. Higher scores reflect smoother transitions between frames.

*DD (Dynamic Degrees) (Huang et al. (2024b))*: Assesses the diversity and complexity of dynamic elements in the video. Higher scores indicate richer and more varied content.

*SC (Subject Consistency) (Huang et al. (2024b))*: Measures the consistency of subjects (e.g., characters, objects) throughout the video. Higher scores indicate better consistency.

*IQ (Imaging Quality) (Huang et al. (2024b))*: Measures the technical quality of the generated frames, including sharpness and resolution. Higher scores indicate clearer and more detailed images.

Table 8: User Preference Evaluation Criteria.

| Score | Evaluation Criteria |
|---|---|
| 1 | High video quality: 1. The appearance and morphological features of objects in the video are completely consistent 2. High picture stability, maintaining high resolution consistently 3. Overall composition/color/boundaries match reality 4. The picture is visually appealing |
| 0.5 | Average video quality: 1. The appearance and morphological features of objects in the video are at least 80% consistent 2. Moderate picture stability, with only 50% of the frames maintaining high resolution 3. Overall composition/color/boundaries match reality by at least 70% 4. The picture has some visual appeal |
| 0 | Poor video quality: large inconsistencies in appearance and morphology, low video resolution, and composition/layout not matching reality |

Table 9: Text-video Alignment Evaluation Criteria.

| Score | Evaluation Criteria |
|---|---|
| 1 | 100% follow the text instruction requirements, including but not limited to: elements completely correct, quantity requirements consistent, elements complete, features accurate, etc. |
| 0.5 | 100% follow the text instruction requirements, but the implementation has minor flaws such as distorted main subjects or inaccurate features. |
| 0 | Does not 100% follow the text instruction requirements, with any of the following issues: 1. Generated elements are inaccurate 2. Quantity is incorrect 3. Elements are incomplete 4. Features are inaccurate |

**Ablation Experiment Design Details.** We evaluate our proposed methods from two perspectives: generation and control ability. Consequently, we design a comprehensive ablation study. Due to the heavy cost of training on our OGameData-GEN dataset, we follow the approach of Pixart-alpha (Chen et al. (2023)) and sample a smaller subset for the ablation study. Specifically, we sample 20k samples from OGameData-GEN to train the generation ability and 10k samples from OGameData-INS to train the control ability. This resulted in two datasets, OGameData-GEN-Abl and OGameData-INS-Abl. The generation process is trained for 3 epochs, while the control process is trained for 2 epochs. All experiments are conducted on 8 H800 GPUs, utilizing the PyTorch framework. Here, we provide a detailed description of our ablation studies:

*Baseline:* The baseline's setting is aligned with our model. Only utilizing a smaller dataset and training 3 epochs for generation and 2 epochs for instruction tuning with InstructNet.

*w/ MiraData:* To demonstrate the quality of our proposed datasets, we sampled the same video hours sample from MiraData. These videos are also from the game domain. Only utilizing this dataset, we train a model for 3 epochs.

*w/ Short Caption:* To demonstrate the effectiveness of our captioning methods, we re-caption the OGameData-Gen-Abl dataset using simple and short captions. We train the model's generation ability for 3 epochs and use the rewritten short OGameEval-Gen captions to evaluate this variant.

*w/ Progressive Training:* To demonstrate the effectiveness of our mixed-scale and mixed-temporal training, we adopt a different training method. Initially, we train the model using 480p resolution and 102 frames for 2 epochs, followed by training with 720p resolution and 102 frames for an additional epoch.

*w/o Instruct Caption:* We recaption the OGameData-INS-Abl dataset on our ablation utilizing a dense caption. Based on this dataset and the baseline model, we train the model with InstructNet for 2 epochs to evaluate the effectiveness of our proposed structural caption methods.

*w/o Decomposition:* The decoupling of generation and control tasks is essential in our approach. In this variant, we combine these two tasks. We trained the model on the merged OGameData-Gen-Abl and OGameData-INS-Abl dataset for 5 epochs, splitting the training equally: 50% for generation and 50% for instruction tuning.

*w/o InstructNet:* To evaluate the effectiveness of our InstructNet, we utilized OGameData-INS-Abl to continue training the baseline model for control tasks for 2 epochs.

### D.3 HUMAN EVALUATION DETAILS

**Overview.** We recruited 10 volunteers through an online application process, specifically selecting individuals with both gaming domain expertise and AIGC community experience. Prior to the evaluation, all participants provided informed consent. The evaluation framework was designed to assess three key metrics: user preference, video-text alignment, and control success rate. We implemented a blind evaluation protocol where videos and corresponding texts were presented without model attribution. Evaluators were not informed about which model generated each video, ensuring unbiased assessment.

**User Preference.** To assess the overall quality of generated videos, we evaluate them across several dimensions, such as motion consistency, aesthetic appeal, and temporal coherence. This evaluation focuses specifically on the visual qualities of the content, independent of textual prompts or control signals. By isolating the visual assessment, we can better measure the model's ability to generate high-quality, visually compelling, and temporally consistent videos. To ensure an unbiased evaluation, volunteers were shown the generated videos without any accompanying textual prompts. This approach allows us to focus solely on visual quality metrics, such as temporal consistency, composition, object coherence, and overall quality. The evaluation criteria in Table 8 consist of three distinct quality tiers, ranging from high-quality outputs that demonstrate full consistency and visual appeal to low-quality outputs that exhibit significant inconsistencies in appearance and composition.

**Text-Video Alignment.** The text-video alignment evaluation aims to assess how well the model can follow textual instructions to generate visual content, with a particular focus on gaming-style aesthetics. This metric looks at both semantic accuracy (how well the text elements are represented) and stylistic consistency (how well the video matches the specific gaming style), providing a measure of the model's ability to faithfully interpret textual descriptions within the context of gaming. Evaluators were shown paired video outputs along with their corresponding textual prompts. The evaluation framework focuses on two main aspects: (1) the accuracy of the implementation of instructional elements, such as object presence, quantity, and feature details, and (2) how well the video incorporates gaming-specific visual aesthetics. The evaluation criteria in Table 9 use a three-tier scoring system: a score of 1 for perfect alignment with complete adherence to instructions, 0.5 for partial success with minor flaws, and 0 for significant deviations from the specified requirements. This approach provides a clear, quantitative way to assess how well the model follows instructions, while also considering the unique demands of generating game-style content.

**Success Rate.** The purpose of the control success rate evaluation is to assess the model's ability to accurately follow control instructions provided in the prompt. This evaluation focuses on how well the generated videos follow the specified control signals while maintaining natural transitions and avoiding any abrupt changes or visual inconsistencies. By combining human judgment with AI-assisted analysis, this evaluation aims to provide a robust measure of the model's performance in responding to user controls. We implemented a hybrid evaluation approach, combining feedback from human evaluators and AI-generated analysis. Volunteers were given questionnaires where they watched the generated videos and assessed whether the control instructions had been successfully followed. For each prompt, we generated three distinct videos using different random seeds to ensure diverse outputs. The evaluators scored each video: a score of 1 was given if the control was successfully implemented, and 0 if it was not. The criteria for successful control included strict adherence to the textual instructions and smooth, natural transitions between scenes without abrupt changes or visual discontinuities. In addition to human evaluations, we used PLLaVA (Xu et al. (2024)) to generate captions for each video, which were provided to the evaluators as a supplemen-

tary tool for assessing control success. Evaluators examined the captions for the presence of key control-related elements from the prompt, such as specific keywords or semantic information (e.g., "turn left," "rainy," or "jump"). This allowed for a secondary validation of control success, ensuring that the model-generated content matched the intended instructions both visually and semantically. For each prompt, we computed the success rate for each model by averaging the scores from the human evaluation and the AI-based caption analysis. This dual-verification process provided a comprehensive assessment of the model's control performance. Higher scores indicate better control precision, reflecting the model's ability to accurately follow the given instructions.

### D.4 Analysis of Generation Speed and Corresponding Performance

In this subsection, we supplement our work with experiments and analyses related to generation speed and performance. Specifically, we conducted 30 open-domain generation inferences on a single A800 and a single H800 GPU, with the CUDA environment set to 12.1. We recorded the time and corresponding FPS, and reported the VBench metrics, including SC, background consistency (BC), DD, aesthetic quality (AQ), IQ, and averaged score of them (overall).

**Generation Speed.** The Table 10 reported the generation speed and corresponding FPS. In terms of generation speed, higher resolutions and more sampling steps result in increased time consumption. Similar to the conclusions found in GameNGen (Valevski et al. (2024)), the model generates videos with acceptable imaging quality and relatively high FPS at lower resolutions and fewer sampling steps (e.g., 320x256, 10 sampling steps).

Table 10: Performance comparison between A800 and H800

| Resolution | Frames | Sampling Steps | Time (A800) | FPS (A800) | Time (H800) | FPS (H800) |
|---|---|---|---|---|---|---|
| $320 \times 256$ | 102 | 10 | ~7.5s/sample | 13.6 | ~5.1s/sample | 20.0 |
| $848 \times 480$ | 102 | 10 | ~60s/sample | 1.7 | ~20.1s/sample | 5.07 |
| $848 \times 480$ | 102 | 30 | ~136s/sample | 0.75 | ~44.1s/sample | 2.31 |
| $848 \times 480$ | 102 | 50 | ~196s/sample | 0.52 | ~69.3s/sample | 1.47 |
| $1280 \times 720$ | 102 | 10 | ~160s/sample | 0.64 | ~38.3s/sample | 2.66 |
| $1280 \times 720$ | 102 | 30 | ~315s/sample | 0.32 | ~57.5s/sample | 1.77 |
| $1280 \times 720$ | 102 | 50 | ~435s/sample | 0.23 | ~160.1s/sample | 0.64 |

**Performance Analysis.** From Table 11, we can observe that increasing the number of sampling steps generally improves visual quality at the same resolution, as reflected in the improvement of the Overall score. For example, at resolutions of 848x480 and 1280x720, increasing the sampling steps from 10 to 50 significantly improved the Overall score, from 0.737 to 0.800 and from 0.655 to 0.812, respectively. This suggests that higher resolutions typically require more sampling steps to achieve optimal visual quality. On the other hand, we qualitatively studied the generated videos. We observed that at a resolution of 320p, our model can produce visually coherent and texture-rich results with only 10 sampling steps. As shown in Fig. 16, details such as road surfaces, cloud textures, and building edges are generated clearly. At this resolution and number of sampling steps, the model can achieve 20 FPS on a single H800 GPU. We also observed the impact of sampling steps on the generation quality at 480p/720p resolutions, as shown in Fig. 17. At 10 sampling steps, we observed a significant enhancement in high-frequency details. Sampling with 30 and 50 steps not only further enriched the textures but also increased the diversity, coherence, and overall richness of the generated content, with more dynamic effects such as cape movements and ion effects. This aligns with the quantitative analysis metrics.

Table 11: Performance metrics for different resolutions and sampling steps

| Resolution | Frames | Sampling Steps | SC | BC | DD | AQ | IQ | Average |
|---|---|---|---|---|---|---|---|---|
| $320 \times 256$ | 102 | 10 | 0.944 | 0.962 | 0.4 | 0.563 | 0.335 | 0.641 |
| $848 \times 480$ | 102 | 10 | 0.947 | 0.954 | 0.8 | 0.598 | 0.389 | 0.737 |
| $848 \times 480$ | 102 | 30 | 0.964 | 0.960 | 0.9 | 0.645 | 0.573 | 0.808 |
| $848 \times 480$ | 102 | 50 | 0.955 | 0.961 | 0.9 | 0.615 | 0.570 | 0.800 |
| $1280 \times 720$ | 102 | 10 | 0.957 | 0.963 | 0.3 | 0.600 | 0.453 | 0.655 |
| $1280 \times 720$ | 102 | 30 | 0.954 | 0.956 | 0.7 | 0.617 | 0.558 | 0.757 |
| $1280 \times 720$ | 102 | 50 | 0.959 | 0.959 | 0.8 | 0.657 | 0.584 | 0.812 |

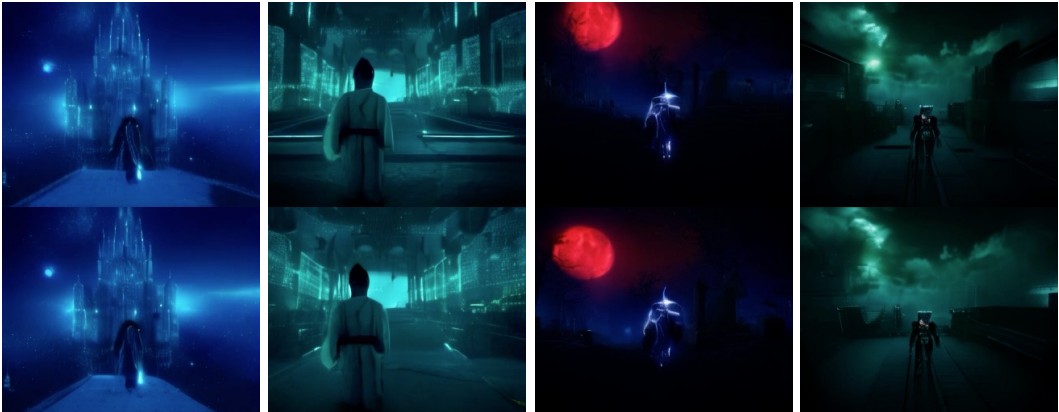

Figure 16: Generated scenes with a resolution of 320x256 and 10 sampling steps. Despite the lower resolution, the model effectively captures key scene elements.

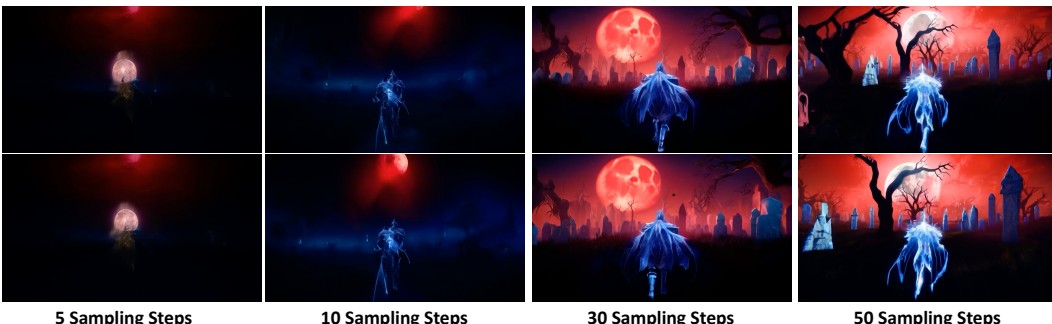

**5 Sampling Steps**     **10 Sampling Steps**     **30 Sampling Steps**     **50 Sampling Steps**

Figure 17: Generated scenes at a resolution of 848x480 with varying sampling steps: 5, 10, 30, and 50. As the number of sampling steps increases, the visual quality of the generated scenes improves significantly.

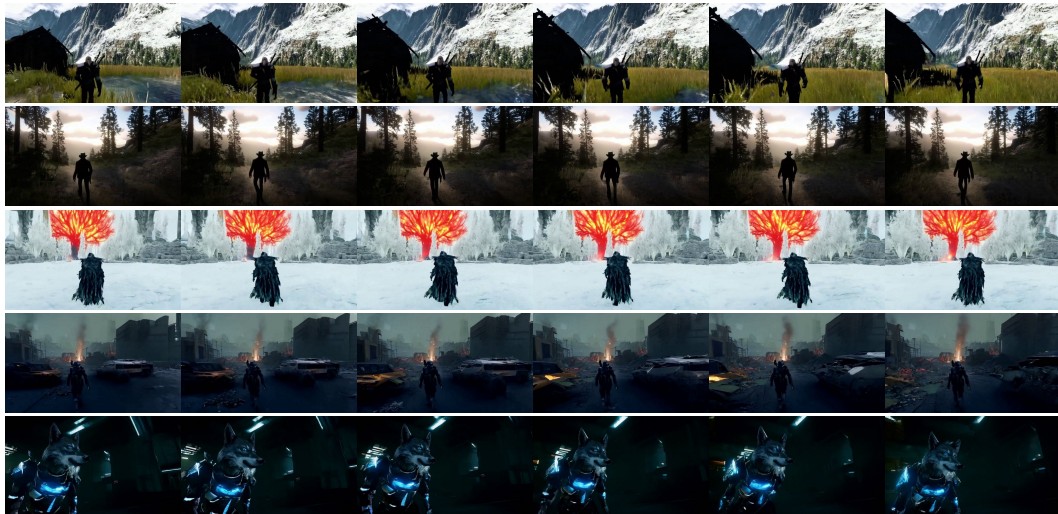

Figure 18: **Character Generation Diversity**. The model demonstrates its capability to generate a wide range of characters. The first three rows depict characters from existing games, showcasing detailed and realistic designs. The last two rows present open-domain character generation, illustrating the model's versatility in creating unique and imaginative characters.

### D.5 FURTHER QUALITATIVE EXPERIMENTS

**Basic Functionality.** Our model is designed to generate high-quality game videos with creative content, as illustrated in Fig. 18, Fig. 19, Fig. 20, and Fig. 21. It demonstrates a strong capability for diverse scene generation, including the creation of main characters from over 150 existing games as well as novel, out-of-domain characters. This versatility extends to simulating a wide array of actions such as flying, driving, and biking, providing a wide variety of gameplay experiences. In addition, our model adeptly constructs environments that transition naturally across different seasons, from spring to winter. It can depict a range of weather conditions, including dense fog, snowfall, heavy rain, and ocean waves, thereby enhancing the ambiance and immersion of the game. By introducing diverse and dynamic scenarios, the model adds depth and variety to generated game content, offering a glimpse into potential engine-like features from generative models.

**Open-domain Generation Comparison.** To evaluate the open-domain content creation capabilities of our method compared to other open-source models, we utilized GPT-4o to randomly generate captions. These captions were used to create open-domain game video demos. We selected three distinct caption types: Structural captions aligned with our dataset, short captions, and dense and general captions that follow human style. The results for Structural captions are illustrated in Fig. 23, Fig. 22, Fig. 24, Fig. 25, Fig. 26, and Fig. 27. The outcomes for short captions are depicted in Fig. 28 and Fig. 29, while the results for dense captions are visualized in Fig. 30. For each caption type, we selected one example for detailed analysis. As illustrated in Fig. 24, we generated a scene depicting a warrior walking through a stormy wasteland. The results show that CogVideoX lacks scene consistency due to dramatic light changes. In contrast, Opensora-Plan fails to accurately follow the user's instructions by missing realistic lighting effects. Additionally, Opensora's output lacks dynamic motion, as the main character appears to glide rather than walk. Our method achieves superior results compared to these approaches, providing a more coherent and accurate depiction. We selected the scene visualized in Fig. 29 as an example of short caption generation. As depicted, the results from CogVideoX fail to fully capture the textual description, particularly missing the ice-crystal hair of the fur-clad wanderer. Additionally, Opensora-Plan lacks the auroras in the sky, and Opensora's output also misses the ice-crystal hair feature. These shortcomings highlight the robustness of our method, which effectively interprets and depicts details even with concise captions. The dense caption results are visualized in Fig. 30. Our method effectively captures the text details, including the golden armor and the character standing atop a cliff. In contrast, other methods fail to accurately depict the golden armor and the cliff, demonstrating the superior capability of our approach in representing detailed information.

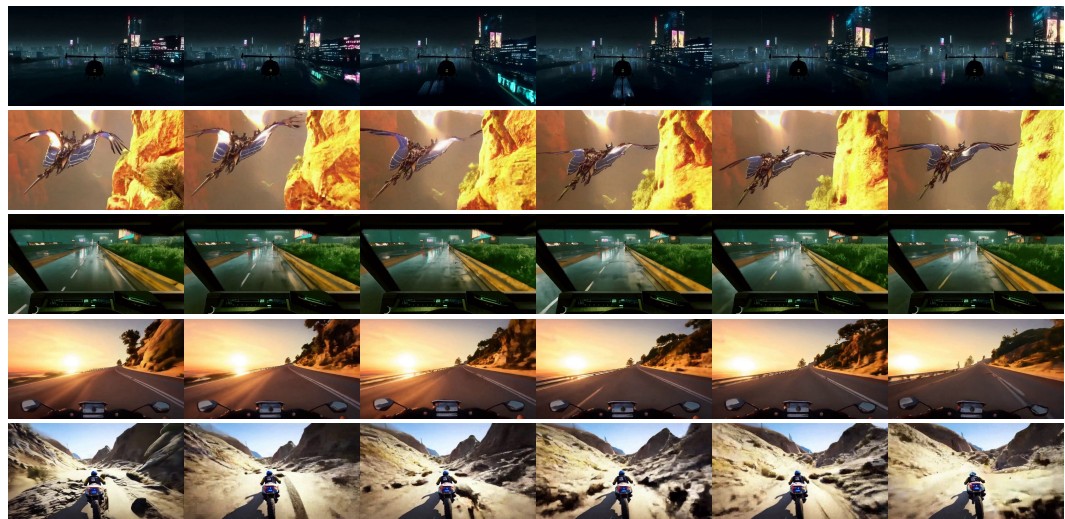

Figure 19: **Action Variety in Scene Generation.** The model effectively demonstrates diverse action scenarios. From top to bottom: piloting a helicopter, flying through a canyon, third-person driving, first-person motorcycle riding, and third-person motorcycle riding. Each row showcases the model's dynamic range in generating realistic and varied action sequences.

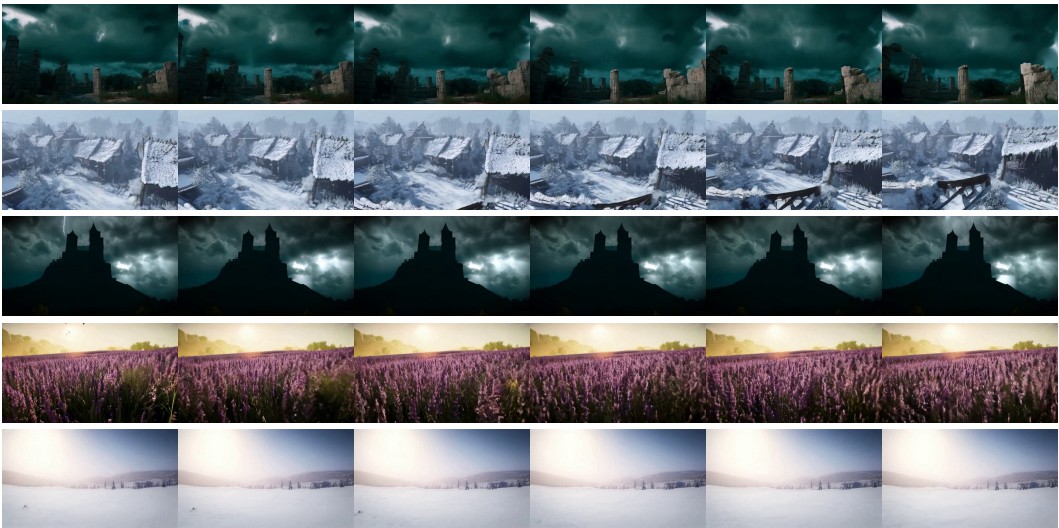

Figure 20: **Environmental Variation in Scene Generation.** The model illustrates its capability to produce diverse environments. From top to bottom: a summer scene with an approaching hurricane, a snow-covered winter village, a summer thunderstorm, lavender fields in summer, and a snow-covered winter landscape. These examples highlight the model's ability to capture different seasonal and weather conditions vividly.

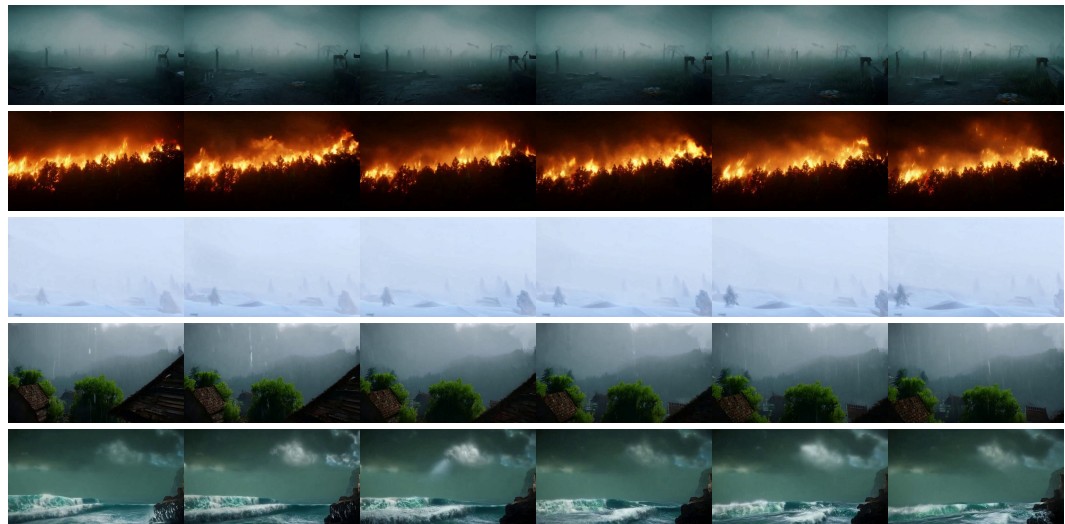

Figure 21: **Event Diversity in Scene Generation.** The model showcases its ability to depict a range of dynamic events. From top to bottom: dense fog, a raging wildfire, heavy rain, and powerful ocean waves. Each scenario highlights the model's capability to generate realistic and intense atmospheric conditions.

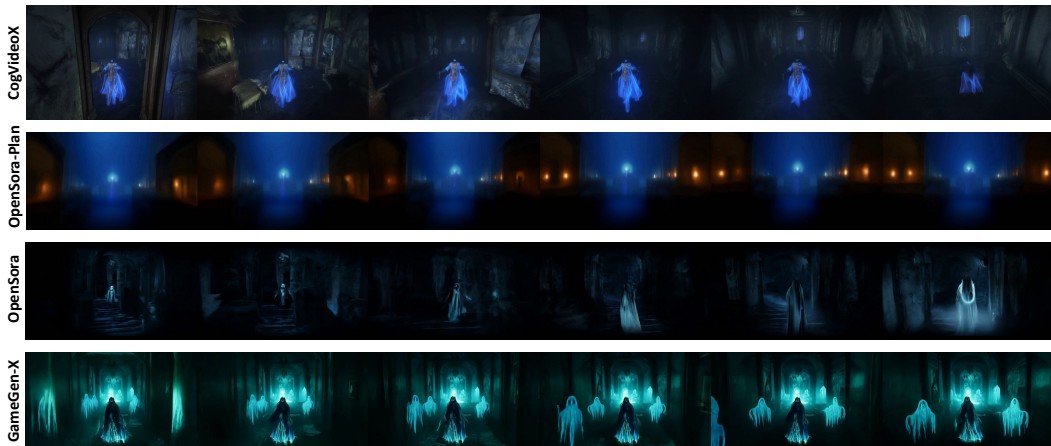

Figure 22: Structural Prompt: A spectral mage explores a haunted mansion filled with ghostly apparitions. In "Phantom Manor," the protagonist, a mysterious figure shrouded in ethereal robes, glides through the dark, decaying halls of an ancient mansion. The walls are lined with faded portraits and cobweb-covered furniture. Ghostly apparitions flicker in and out of existence, their mournful wails echoing through the corridors. The mage's staff glows with a faint, blue light, illuminating the path ahead and revealing hidden secrets. The air is thick with an eerie, supernatural presence, creating a chilling, immersive atmosphere. aesthetic score: 6.55, motion score: 12.69, perspective: Third person.

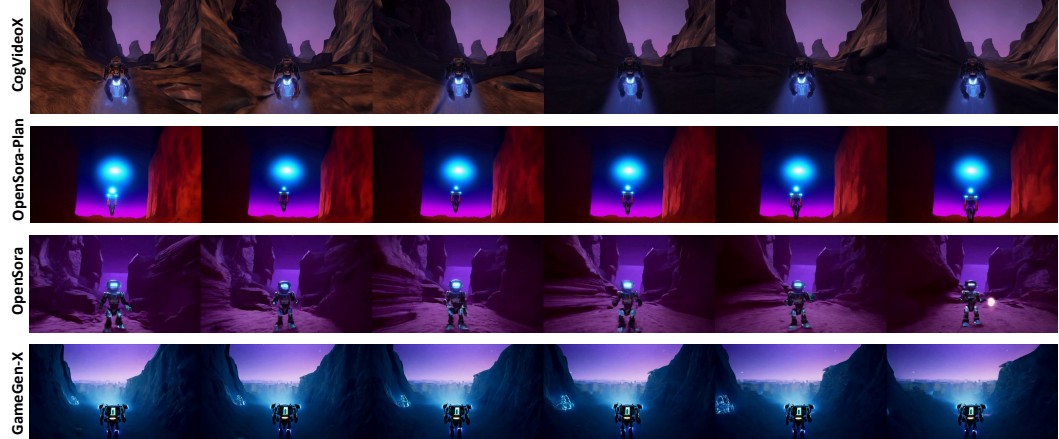

Figure 23: Structural Prompt: A robotic explorer traverses a canyon filled with ancient, alien ruins. In "Mechanized Odyssey," the main character, a sleek, humanoid robot with a glowing core, navigates through a vast, rocky canyon. The canyon walls are adorned with mysterious, ancient carvings and partially buried alien structures. The robot's sensors emit a soft, blue light, illuminating the path ahead and revealing hidden details in the environment. The sky is a deep, twilight purple, with distant stars beginning to appear, adding to the sense of exploration and discovery. aesthetic score: 6.55, motion score: 12.69, perspective: Third person.

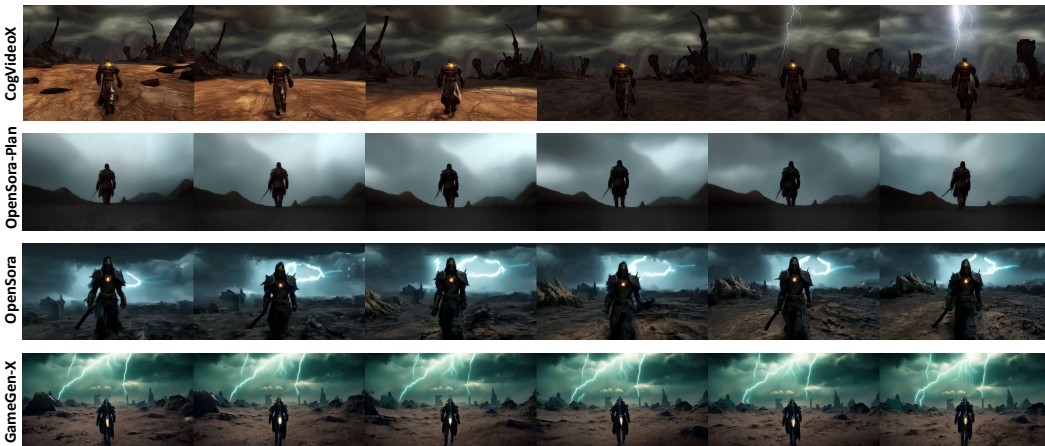

Figure 24: Structural Prompt: A lone warrior walks through a stormy wasteland, the sky filled with lightning and dark clouds. In "Stormbringer", the protagonist, clad in weathered armor with a glowing amulet, strides through a barren, rocky landscape. The ground is cracked and dry, and the air is thick with the smell of ozone. Jagged rocks and twisted metal structures dot the horizon, while bolts of lightning illuminate the scene intermittently. The warrior's path is lit by the occasional flash, creating a dramatic and foreboding atmosphere. aesthetic score: 6.55, motion score: 12.69, perspective: Third person.

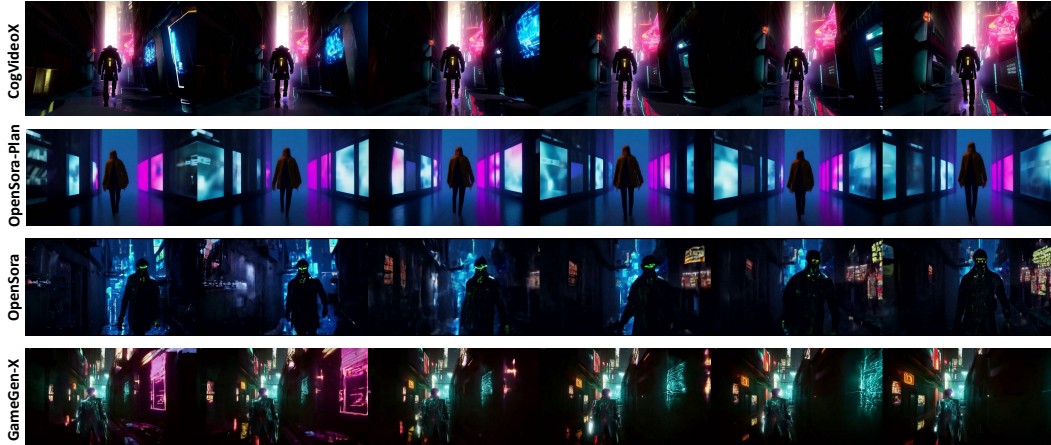

Figure 25: Structural Prompt:A cybernetic detective walks down a neon-lit alley in a bustling city. In "Neon Shadows," the protagonist wears a trench coat with glowing circuitry, navigating through a narrow alley filled with flickering holographic advertisements. Rain pours down, causing puddles on the ground to reflect the vibrant city lights. The buildings loom overhead, casting long shadows that create a sense of depth and intrigue. The detective's steps are steady, their eyes scanning the surroundings for clues in this cyberpunk mystery. aesthetic score: 6.55, motion score: 12.69, perspective: Third person.'

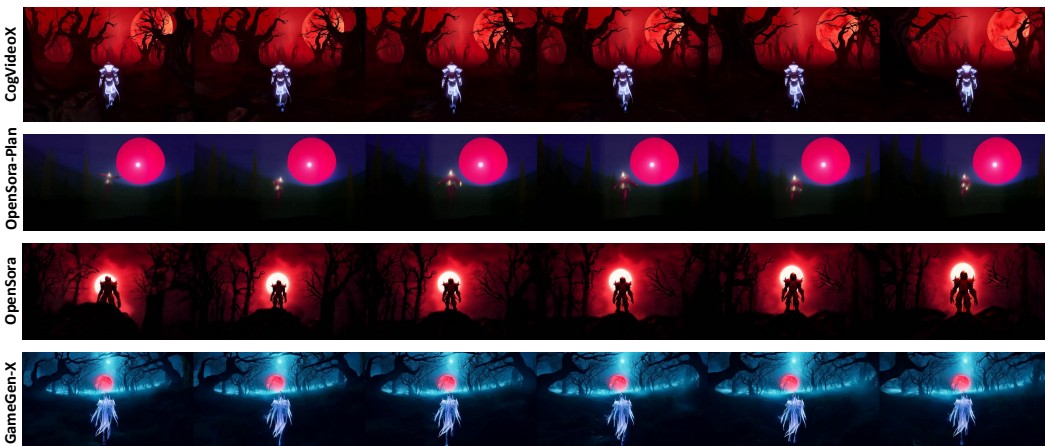

Figure 26: Structural Prompt: A spectral knight walks through a haunted forest under a blood-red moon. In "Phantom Crusade," the protagonist, a translucent, ethereal figure clad in spectral armor, moves silently through a dark, misty forest. The trees are twisted and gnarled, their branches reaching out like skeletal hands. The blood-red moon casts an eerie light, illuminating the path with a sinister glow. Ghostly wisps float through the air, adding to the chilling atmosphere. The knight's armor shimmers faintly, reflecting the moonlight and creating a hauntingly beautiful scene. aesthetic score: 6.55, motion score: 12.69, perspective: Third person.

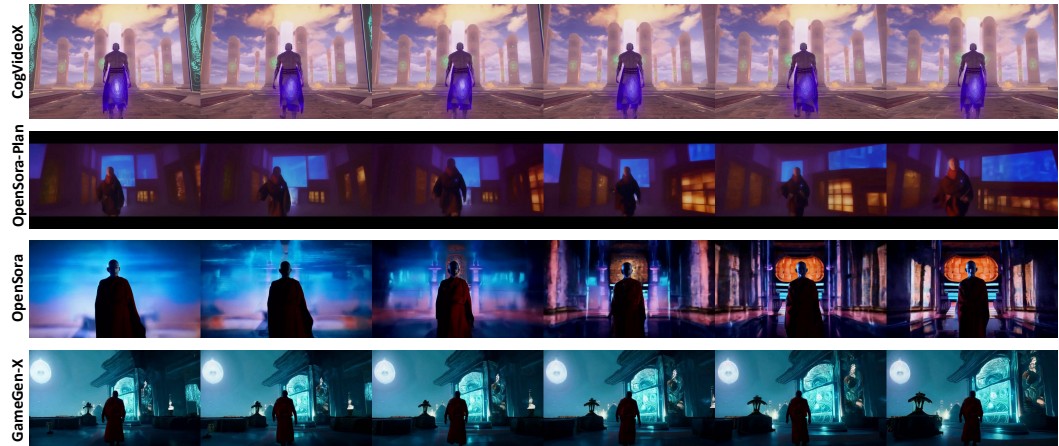

Figure 27: Structural Prompt: A cybernetic monk walks through a high-tech temple under a serene sky. In "Digital Zen," the protagonist, a serene figure with cybernetic enhancements integrated into their traditional monk robes, walks through a temple that blends ancient architecture with advanced technology. Soft, ambient lighting and the gentle hum of technology create a peaceful atmosphere. The temple's walls are adorned with holographic screens displaying calming patterns and mantras. The monk's cybernetic components emit a faint, soothing glow, symbolizing the fusion of spirituality and technology in this tranquil sanctuary. aesthetic score: 6.55, motion score: 12.69, perspective: Third person.

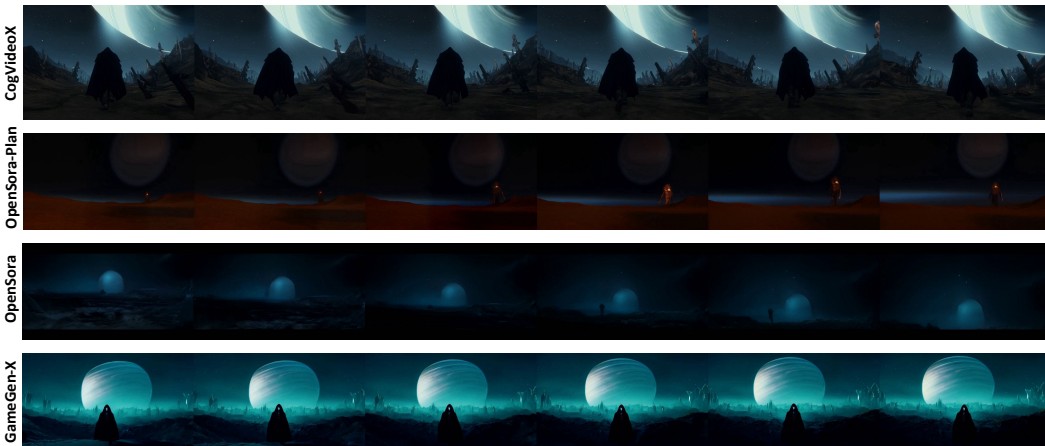

Figure 28: Short Prompt: "Echoes of the Void": A figure cloaked in darkness with eyes like stars walks through a valley where echoes of past battles appear as ghostly figures. The ground is littered with ancient, rusted weapons, and the sky is an endless void with a single, massive planet looming close, its rings casting eerie shadows.

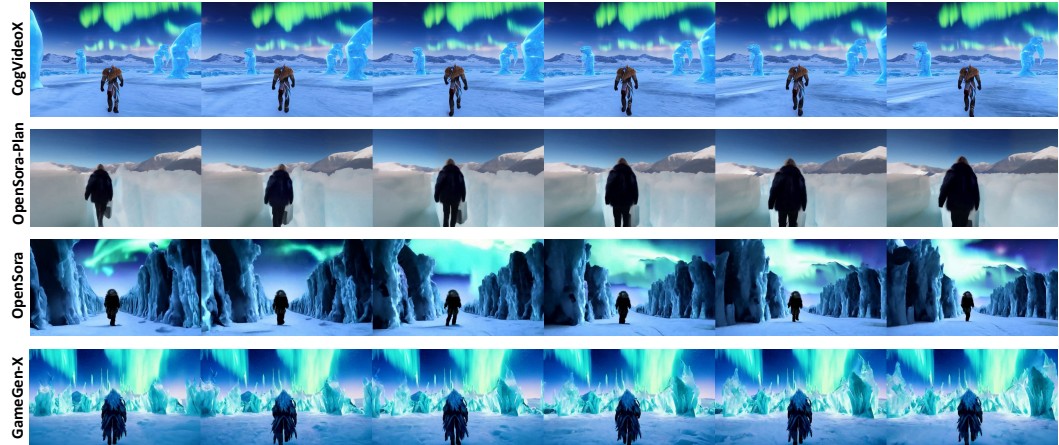

Figure 29: Short Prompt: "Glacier Wanderer": A fur-clad wanderer with ice-crystal hair treks across a glacier under a sky painted with auroras. Giant ice sculptures of mythical creatures line his path, each breathing out cold mist. The horizon shows mountains that pierce the sky, glowing with an inner light.

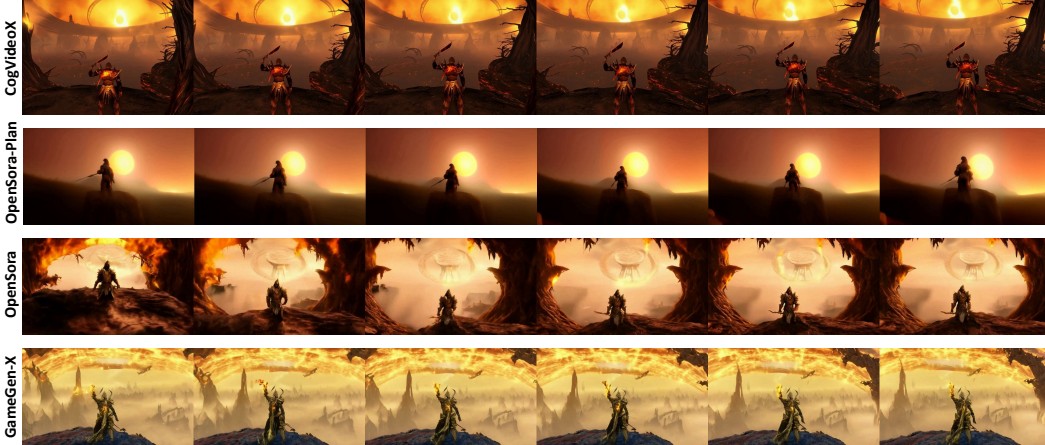

Figure 30: Dense Prompt: A lone Tarnished warrior, clad in tattered golden armor that glows with inner fire, stands atop a cliff overlooking a vast, blighted landscape. The sky burns with an otherworldly amber light, casting long shadows across the desolate terrain. Massive, twisted trees with bark-like blackened iron stretch towards the heavens, their branches intertwining to form grotesque arches. In the distance, a colossal ring structure hovers on the horizon, its edges shimmering with arcane energy. The air is thick with ash and embers, swirling around the warrior in mesmerizing patterns. Below, a sea of mist conceals untold horrors, occasionally parting to reveal glimpses of ancient ruins and fallen titans. The warrior raises a curved sword that pulses with crimson runes, preparing to descend into the nightmarish realm below. The scene exudes a sense of epic scale and foreboding beauty, capturing the essence of a world on the brink of cosmic change.

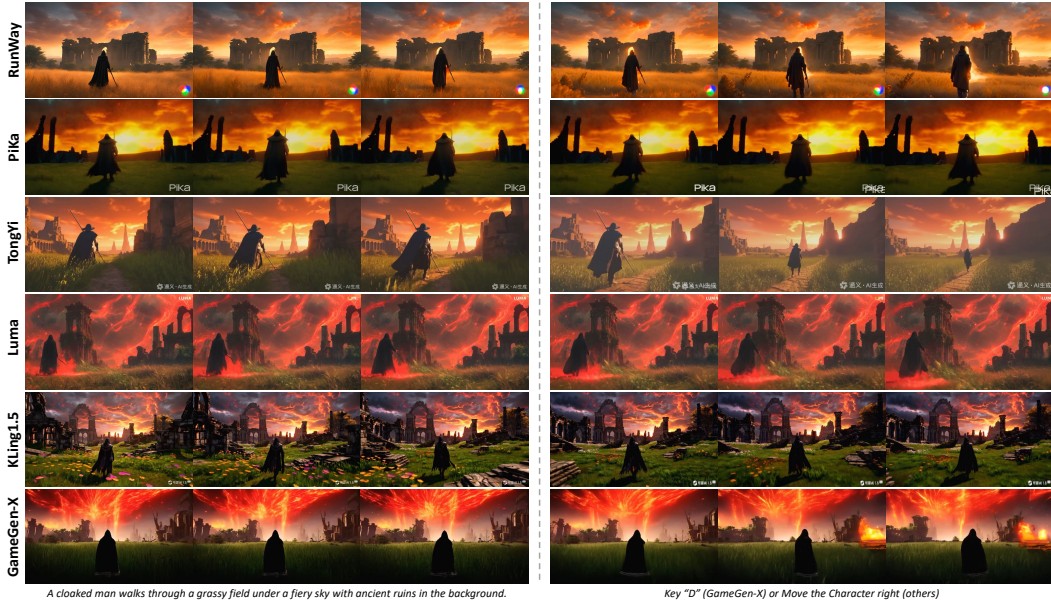

Figure 31: Comparison results of GameGen-𝕏 with commercial models. This figure contrasts our approach with several commercial models. The left side displays results from text-generated videos, while the right side shows text-based continuation of videos. From top to bottom, the models include Runway Gen2, Pika, Tongyi, Luma, Kling1.5, and GameGen-𝕏. Luma, Kling1.5, and GameGen-𝕏 effectively followed the caption in the first part, including capturing the fiery red sky, while Gen2, Pika, and Tongyi did not. In the second part, our method successfully directed the character to turn right, a control other methods struggled to achieve.

**Interactive Control Ability Comparison.** To comprehensively assess the controllability of our model, we compared it with several commercial models, including Runway Gen2, Pika, Luma, Tongyi, and KLing 1.5. Initially, we generated a scene using the same caption across all models. Subsequently, we extended the video by incorporating text instructions related to environmental changes and character direction. The results are presented in Fig. 31, Fig. 32, Fig. 33, Fig. 34, and Fig. 35. Our findings reveal that while commercial models can produce high-quality outputs, Runway, Pika, and Luma fall short of meeting game demo creation needs due to their free camera perspectives, which lack the dynamic style typical of games. Although Tongyi and KLing can generate videos with a game-like style, they lack adequate control capabilities; Tongyi fails to respond to environmental changes and character direction, while KLing struggles with character direction adjustments.

**Video Prompt.** In addition to text and keyboard inputs, our model accepts video prompts, such as edge sequences or motion vectors, as inputs. This capability allows for more customized video generation. The generated results are visualized in Fig. 36 and Fig. 37.

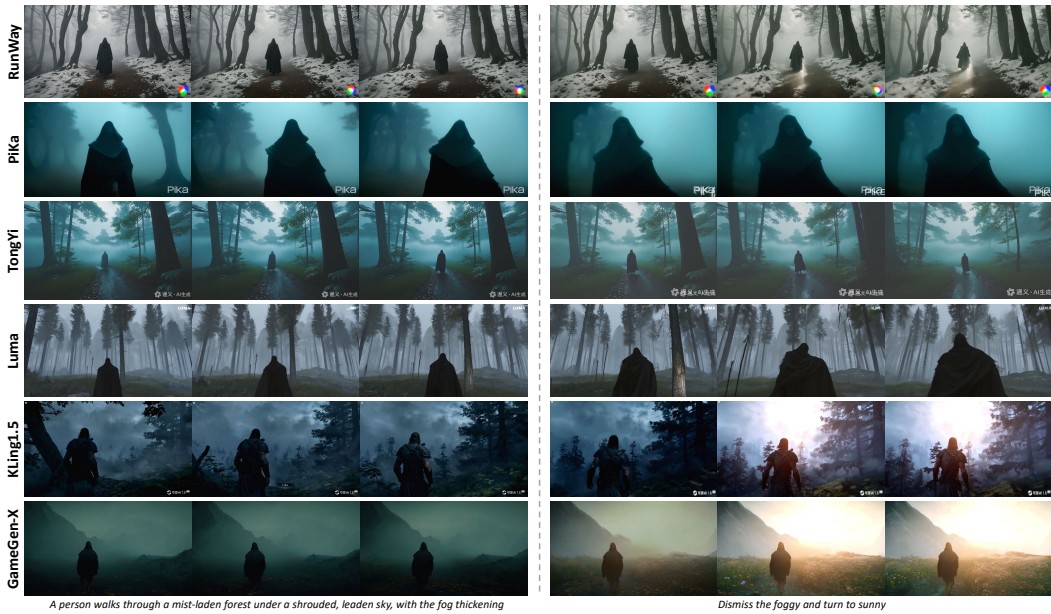

Figure 32: Comparison results of GameGen-𝕏 with commercial models. This figure presents a comparison between our approach and several commercial models. The left side depicts text-generated video results, while the right side shows text-based video continuation. From top to bottom, the models include Runway Gen2, Pika, Tongyi, Luma, Kling1.5, and GameGen-𝕏. In the initial segment, Luma, Kling1.5, and GameGen-𝕏 effectively adhered to the caption by accurately depicting the dense fog and path, while other models lacked these elements. In the continuation, only Kling1.5 and our approach successfully transformed the environment by clearing the fog, whereas other methods failed to follow the text instructions.

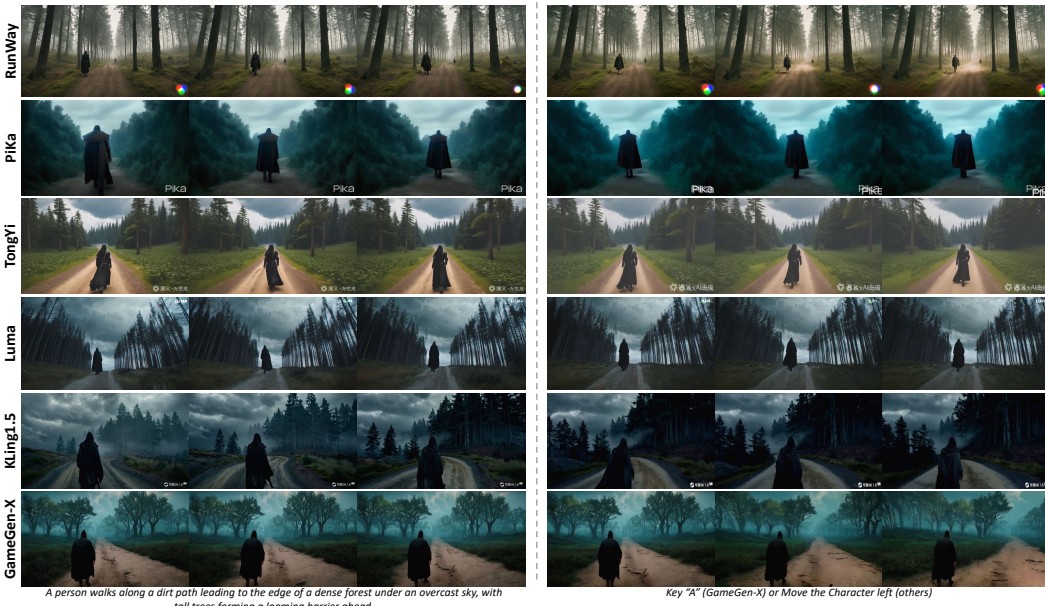

Figure 33: Comparison results of GameGen-𝕏 with commercial models. This figure compares our approach with several commercial models. The left side displays text-generated video results, while the right side shows text-based video continuation. From top to bottom, the models include Runway Gen2, Pika, Tongyi, Luma, Kling1.5, and our method. In the initial segment, all methods effectively followed the caption. However, in the continuation segment, only our model successfully controlled the character to turn left.

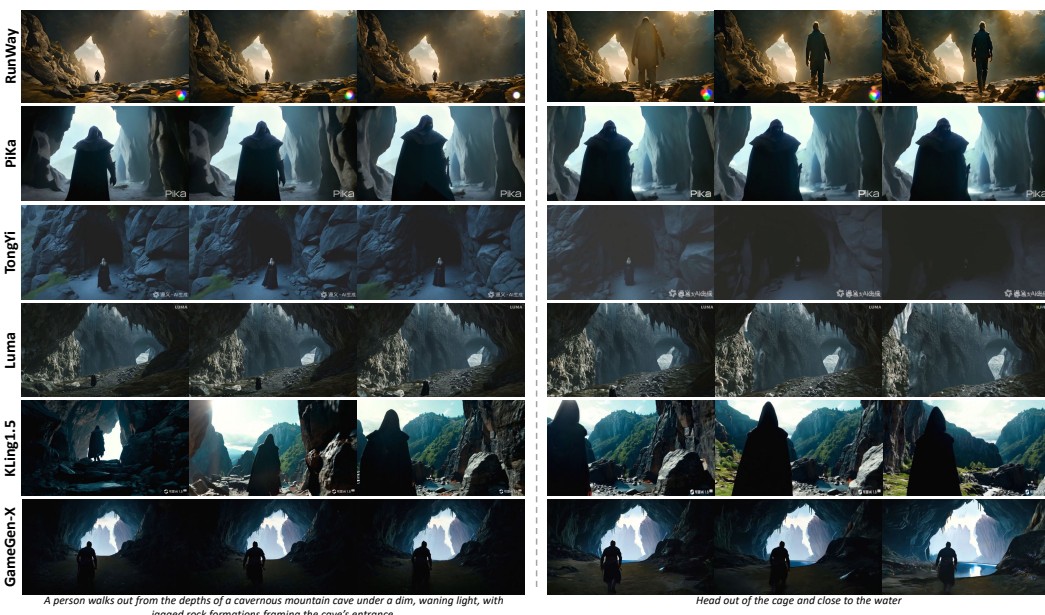

Figure 34: Comparison results of GameGen-𝕏 with commercial models. This figure presents a comparison between our approach and several commercial models. The left side showcases text-generated video results, while the right side illustrates video continuation using text. From top to bottom, the models include Runway Gen2, Pika, Tongyi, Luma, Kling1.5, and our method. In the first segment, only Pika, Kling1.5, and our method correctly followed the text description. Other models either failed to display the character or depicted them entering the cave instead of exiting. In the continuation segment, both our method and Kling1.5 successfully guided the character out of the cave. Our approach maintains a consistent camera perspective, enhancing the game-like experience compared to Kling1.5.

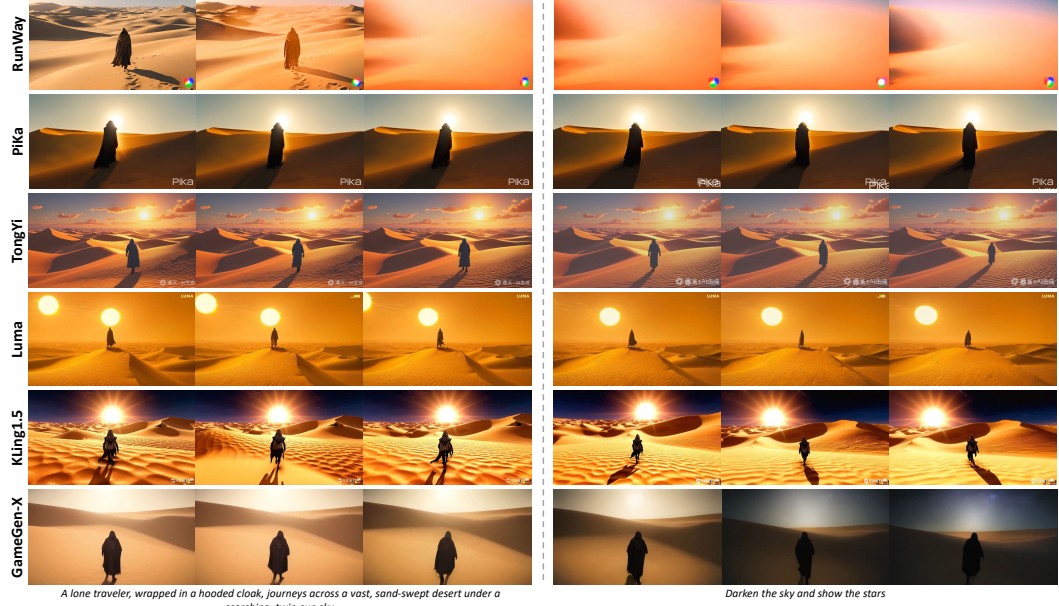

Figure 35: Comparison results of GameGen-𝕏 with commercial models. This figure presents a comparison between our approach and several commercial models. The left side shows text-generated video results, while the right side illustrates video continuation using text. From top to bottom, the models include Runway Gen2, Pika, Tongyi, Luma, Kling1.5, and our method. In the initial segment, all methods successfully followed the text description. However, in the continuation segment, only our method effectively altered the environment by darkening the sky and revealing the stars.

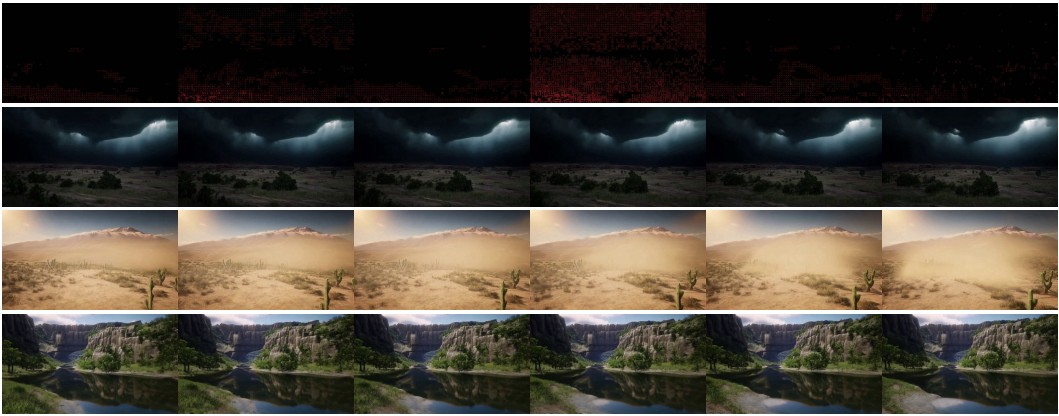

Figure 36: Video Generation with Motion Vector Input. This figure demonstrates how given motion vectors enable the generation of videos that follow specific movements. Different environments were created using various text descriptions, all adhering to the same motion pattern.

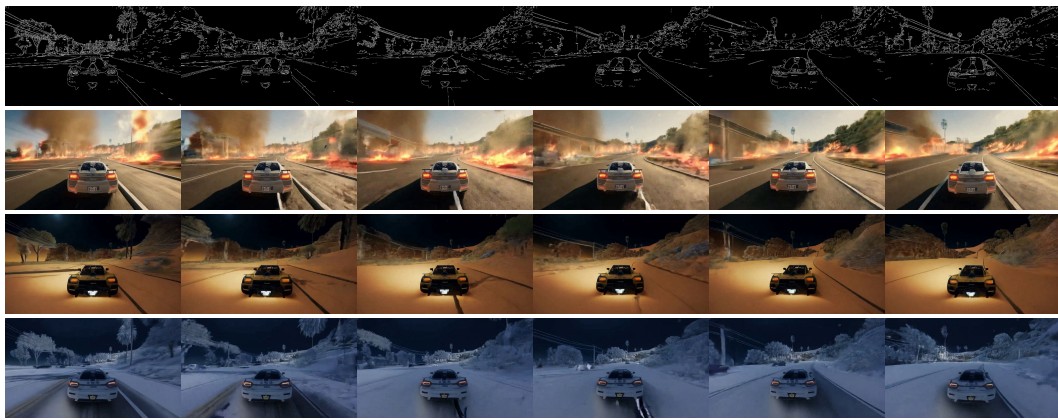

Figure 37: Video Scene Generation with Canny Sequence Input. Using the same canny sequence, different text inputs can generate video scenes that match specific content requirements.

# E DISCUSSION

## E.1 LIMITATIONS

Despite the advancements made by GameGen-$\mathbb{X}$, several key challenges remain:

*Real-Time Generation and Interaction*: In the realm of gameplay, real-time interaction is crucial, and there is a significant appeal in developing a video generation model that enables such interactivity. However, the computational demands of diffusion models, particularly concerning the sampling process and the complexity of spatial and temporal self-attention mechanisms, present formidable challenges.

*Consistency in Auto-Regressive Generation*: Auto-regressive generation often leads to accumulated errors, which can affect both character consistency and scene coherence over long sequences (Valevski et al. (2024)). This issue becomes particularly problematic when revisiting previously generated environments, as the model may struggle to maintain a cohesive and logical progression.

*Complex Action Generation*: The model struggles with fast and complex actions, such as combat sequences, where rapid motion exceeds its current capacity (Huang et al. (2024a)). In these scenarios, video prompts are required to guide the generation, thereby limiting the model's autonomy and its ability to independently generate realistic, high-motion content.

*High-Resolution Generation*: GameGen-$\mathbb{X}$ is not yet capable of generating ultra-high-resolution content (e.g., 2K/4K) due to memory and processing constraints (He et al. (2024a)). The current hardware limitations prevent the model from producing the detailed and high-resolution visuals that are often required for next-gen AAA games, thereby restricting its applicability in high-end game development.

*Long-Term Consistency in Video Generation:* In gameplay, maintaining scene consistency is crucial, especially as players transition between and return to scenes. However, our model currently exhibits a limitation in temporal coherence due to its short-term memory capacity of just 1-108 frames. This constraint results in significant scene alterations upon revisiting, highlighting the need to enhance our model's memory window for better long-term scene retention. Expanding this capability is essential for achieving more stable and immersive video generation experiences.

*Physics Simulation and Realism:* While our methods achieve high visual fidelity, the inherent constraints of generative models limit their ability to consistently adhere to physical laws. This includes realistic light reflections and accurate interactions between characters and their environments. These limitations highlight the challenge of integrating visually compelling content with the physical realism required for experience.

*Multi-Character Generation:* The distribution of our current dataset limits our model's ability to generate and manage interactions among multiple characters. This constraint is particularly evident in scenarios requiring coordinated combat or cooperative tasks.

*Integration with Existing Game Engines:* Presently, the outputs of our model are not directly compatible with existing game engines. Converting video outputs into 3D models may offer a feasible pathway to bridge this gap, enabling more practical applications in game development workflows.

In summary, while GameGen-$\mathbb{X}$ marks a significant step forward in open-world game generation, addressing these limitations is crucial for its future development and practical application in real-time, high-resolution, and complex game scenarios.

## E.2 POTENTIAL FUTURE WORKS

Potential future works may benefit from the following aspects:

*Real-Time Optimization:* One of the primary limitations of current diffusion models, including GameGen-$\mathbb{X}$, is the high computational cost that hinders real-time generation. Future research can focus on optimizing the model for real-time performance (Zhao et al. (2024b;a); Xuanlei Zhao & You (2024), essential for interactive gaming applications. This could involve the design of lightweight diffusion models that retain generative power while reducing the inference time. Addi-

tionally, hybrid approaches that blend autoregressive methods with non-autoregressive mechanisms may strike a balance between generation speed and content quality (Zhou et al. (2024)). Techniques like model distillation or multi-stage refinement might further reduce the computational overhead, allowing for more efficient generation processes (Wang et al. (2023b)). Such advances will be crucial for applications where instantaneous feedback and dynamic responsiveness are required, such as real-time gameplay and interactive simulations.

*Improving Consistency:* Maintaining consistency over long sequences remains a significant challenge, particularly in autoregressive generation, where small errors can accumulate over time and result in noticeable artifacts. To improve both spatial and temporal coherence, future works may incorporate map-based constraints that impose global structural rules on the generated scenes, ensuring the continuity of environments even over extended interactions (Yan et al. (2024)). For character consistency, the introduction of character-conditioned embeddings could help the model maintain the visual and behavioral fidelity of in-game characters across scenes and actions (He et al. (2024b); Wang et al. (2024). This can be achieved by integrating embeddings that track identity, pose, and interaction history, helping the model to better account for long-term dependencies and minimize discrepancies in character actions or appearances over time. These approaches could further enhance the realism and narrative flow in game scenarios by preventing visual drift.

*Handling of Complex Actions:* Currently, GameGen-$\mathbb{X}$ struggles with highly dynamic and complex actions, such as fast combat sequences or large-scale motion changes, due to limitations in capturing rapid transitions. Future research could focus on enhancing the model's ability to generate realistic motion by integrating motion-aware components, such as temporal convolutional networks or recurrent structures, that better capture fast-changing dynamics (Huang et al. (2024a)). Moreover, training on high-frame-rate datasets would provide the model with more granular temporal information, improving its ability to handle quick motion transitions and intricate interactions. Beyond data, incorporating external guidance, such as motion vectors or pose estimation prompts, can serve as additional control signals to enhance the generation of fast-paced scenes. These improvements would reduce the model's dependency on video prompts, enabling it to autonomously generate complex and fast-moving actions in real-time, increasing the depth and variety of in-game interactions.

*Advanced Model Architectures:* Future advancements in model architecture will likely move towards full 3D representations to better capture the spatial complexity of open-world games. The current 2D+1D approach, while effective, limits the model's ability to fully understand and replicate 3D spatial relationships. Transitioning from 2D+1D attention-based video generation to more sophisticated 3D attention architectures offers an exciting direction for improving the coherence and realism of generated game environments (Yang et al. (2024); Lab & etc. (2024)). Such a framework could better grasp the temporal dynamics and spatial structures within video sequences, improving the fidelity of generated environments and actions. On the other dimension, instead of only focusing on the generation task, future models could integrate a more unified framework that simultaneously learns both video generation and video understanding. By unifying generation and understanding, the model could ensure consistent layouts, character movements, and environmental interactions across time, thus producing more cohesive and immersive content (Emu3 Team (2024)). This approach could significantly enhance the ability of generative models to capture complex video dynamics, advancing the state of video-based game simulation technology.

*Scaling with Larger Datasets:* While OGameData provides a comprehensive foundation for training GameGen-$\mathbb{X}$, further improvements in model generalization could be achieved by scaling the dataset to include more diverse examples of game environments, actions, and interactions (Ju et al. (2024); Wang et al. (2023c)). Expanding the dataset with additional games, including those from a wider range of genres, art styles, and gameplay mechanics, would expose the model to a broader set of scenarios. This would enhance the model's ability to generalize across different gaming contexts, allowing it to generate more diverse and adaptable content. Furthermore, incorporating user-generated content, modding tools, or procedurally generated worlds could enrich the dataset, offering a more varied set of training examples. This scalability would also improve robustness, reducing overfitting and enhancing the model's capacity to handle novel game mechanics and environments, thereby improving performance across a wider array of use cases.

*Integration of 3D Techniques:* A key opportunity for future development lies in integrating advanced 3D modeling with 3D Gaussian Splatting (3DGS) techniques (Kerbl et al. (2023)). Moving beyond 2D video-based approaches, incorporating 3DGS allows the model to generate complex spatial inter-

actions with realistic object dynamics. 3DGS facilitates efficient rendering of intricate environments and characters, capturing fine details such as lighting, object manipulation, and collision detection. This integration would result in richer, more immersive gameplay, enabling players to experience highly interactive and dynamic game worlds (Shin et al. (2024)).

*Virtual to Reality:* A compelling avenue for future research is the potential to adapt these generative techniques beyond gaming into real-world applications. If generative models can accurately simulate highly realistic game environments, it opens the possibility of applying similar techniques to real-world simulations in areas such as autonomous vehicle testing, virtual training environments, augmented reality (AR), and scenario planning. The ability to create interactive, realistic, and controllable simulations could have profound implications in fields such as robotics, urban planning, and education, where virtual environments are used to test and train systems under realistic but controlled conditions. Bridging the gap between virtual and real-world simulations would not only extend the utility of generative models but also demonstrate their capacity to model complex, dynamic systems in a wide range of practical applications.

In summary, addressing these key areas of future work has the potential to significantly advance the capabilities of generative models in game development and beyond. Enhancing real-time generation, improving consistency, and incorporating advanced 3D techniques will lead to more immersive and interactive gaming experiences, while the expansion into real-world applications underscores the broader impact these models can have.

