# OpenReview forum: "GameGen-X: Interactive Open-world Game Video Generation"
_ICLR.cc/2025/Conference — ICLR 2025 Poster_

### Official Review · Reviewer_1M8F · 2024-10-29

**Soundness:** 3
**Presentation:** 3
**Contribution:** 3
**Rating:** 6
**Confidence:** 4

**Summary:**

This work focuses on generating high-quality, controllable open-world game videos that feature game engine traits. It emphasizes interactive controllability to simulate gameplay effectively. Notably, the authors collected a large-scale Open-World Video Game Dataset (OGameData), which consists of over one million diverse gameplay video clips from more than 150 games, along with informative captions generated by GPT-4o. Methodologically, they introduce a diffusion transformer model as the foundation model and a specially designed network called InstructNet for interactive control. The model is trained on the large-scale OGameData dataset using a two-stage process involving pre-training of the foundation model and instruction tuning for InstructNet.

**Strengths:**

1. This work collects a substantial number of open-world game videos from over 150 games, ultimately constructing more than 1,000,000 text-video pairs with highly detailed annotations. Its scale and diversity of annotations make it stand out, and the release of this dataset is expected to advance the field of game video generation.
2. It produces high-quality, more general realistic game video content. Previous works on game video generation often focused on specific game types, primarily 2D games or limited early 3D games. This work offers a more diverse and high-definition range of scene types for game video generation.

**Weaknesses:**

1. This work attempts to address the interactive control of open-world game video generation for gameplay simulation. However, to fully tackle the interactive issue, the generation speed needs to be considered,  as interactive experiences demand stringent timing requirements, which poses significant challenges. For instance, Google’s [1] achieves real-time rendering, even making it a viable game engine. While this work focuses on higher-resolution video generation, exploring the relationship between speed and performance would be beneficial, along with providing data on rendering time and speed.

[1] Dani Valevski, Yaniv Leviathan, Moab Arar, and Shlomi Fruchter. Diffusion models are real-time game engines. arXiv preprint arXiv:2408.14837, 2024.

2. The paper claims to simulate game engine features like diverse events, yet the examples provided offer quite limited dynamic event simulation, primarily addressing environmental changes like weather and lighting. There remains a gap to true gameplay simulation, such as incorporating NPC interactions or triggering more game-like special events.

**Questions:**

1.  Please provide data on the time required to generate a video segment at different resolutions or for different types of content. A section to analyze the trade-offs between generation quality and speed would be better.

2.  The training details of InstructNet lack specificity regarding the acquisition of video data corresponding to keyboard bindings. It would be beneficial to include more comprehensive information on the data collection process and the training methodology employed.

---

> ### Author Response · Authors · 2024-11-21
> **Response to Reviewer 1M8F (Part 1/3)**
>
> We sincerely appreciate the reviewer's insightful and constructive comments regarding our paper related to generation speed and complex game event simulation. **Due to the time limitation, we are still running the experiments related to generation performance and speed. We apologize for this issue**. If convenient, we would like to first take this opportunity to clarify the following other points:
>
> **[W2] Complex Game Event Simulation**
>
> In this article, GameGen-X primarily simulates certain game engine features, focusing on environmental characters and their corresponding events and actions. We strongly agree with the reviewers' opinion that simulating realistic gameplay, including storylines and cutscenes, voiceovers, dynamic NPC interactions, and growth systems, remains a challenge in creating an AI game engine. Although this study focuses on creating a game scene video from scratch and interacting with it, GameGen-X also has some extensibility to better support game simulation and creation. Currently, we have successfully achieved basic game scene creation and interaction through the DiTs However, simulating a realistic gaming experience requires more complex systems and higher technical integration.
>
> For example, for **game story design and cutscenes**, future work could consider using large language models (LLMs) to design the overall game storyline and fine-grained cutscene scenarios [1-3]. LLMs have powerful text generation and understanding capabilities, which can help design more complex and coherent game plots, thereby enhancing player immersion.
>
>  Additionally, to further **enhance immersion and engagement**, future work could also embed sound elements into the generation process. Sound plays a crucial role in games, not only enhancing the atmosphere but also guiding players' emotions and actions through sound effects and music. By combining audio generation models, such as AudioGPT [4], more realistic sound effects can be achieved, thereby improving the overall gaming experience.
>
>  Similarly, for **complex game trees, game systems, growth elements, and dynamic NPC interactions**, LLMs might serve as agents to construct a dynamic world and system described in the text[5]. In this case, the game LLM can work in conjunction with diffusion models, with one acting as the core of the game system and the other as the game renderer. The LLM can handle the logic and interactions in the game, while the diffusion model generates high-quality visual content. This collaborative approach can significantly enhance the complexity and playability of the game.
>
> Future work might explore **hybrid generation schemes** based on Transfusion [6] to unify the entire game. Future research could also explore better integration of multimodal data, including text, images, audio, and video, to create richer and more diverse game content. Finally, we believe that with the development of hardware technology, advancements in real-time rendering and interaction technology will also provide more possibilities for the realization of AI game engines.
>
> **Reference**
>
> [1] Videodirectorgpt: Consistent multi-scene video generation via LLM-guided planning, 2023.
>
> [2] LLM-grounded Video Diffusion Models, ICLR, 2024.
>
> [3] VideoStudio: Generating Consistent-Content and Multi-Scene Videos, ECCV, 2024.
>
> [4] Audiogpt: Understanding and generating speech, music, sound, and talking head, AAAI, 2024
>
> [5] Large language models and games: A survey and roadmap, 2024.
>
> [6] Transfusion: Predict the next token and diffuse images with one multi-modal model, 2024.

---

> ### Author Response · Authors · 2024-11-21
> **Response to Reviewer 1M8F (Part 2/3)**
>
> **[Q2] The training details of InstructNet lack specificity regarding the acquisition of video data corresponding to keyboard bindings. It would be beneficial to include more comprehensive information on the data collection process and the training methodology employed.**
>
> Thanks for pointing out this issue, here we provide a detailed description of the keyboard bindings video data collection and the training details of InstructNet.
>
> 1. **Dataset acquisition**:
>
> We purchased games on the Steam platform to conduct our instruction data collection. To accurately simulate the in-game lighting and weather effects, we parsed the game's console functions and configured the weather and lighting change events to occur randomly every 5-10 seconds. To emulate player input, we developed a virtual keyboard that randomly controls the character's movements within the game scenes. Our data collection spanned multiple distinct game areas, resulting in nearly 100 hours of recorded data. The program logged the output signals from the virtual keyboard, and we utilized Game Bar to capture the corresponding gameplay footage. This setup allowed us to synchronize the keyboard signals with frame-level data, ensuring precise alignment between the input actions and the visual output.
>
> 2. **Autoressive Tuning**
>
>   The autoregressive tuning phase combines Mask Mechanism for Video Extension and InstructNet for conditional signal injection to enable controlled and temporally coherent video extensions.
>
>
>  - **Mask Mechanism for Video Extension**: The temporal masking strategy is a core component of our autoregressive tuning process, enabling video extension. First, the latent representation of the initial frame is preserved as a fixed reference, anchoring the temporal context for subsequent frame generation. During the training process, frames designated for prediction are added with noise, ensuring that the model focuses on reconstructing unobserved frames while maintaining coherence with observed frames.
>
>
> - **Conditional Signal Injection via InstructNet**: InstructNet is the backbone for integrating diverse control signals, allowing precise and dynamic adjustments during the video extension process. By injecting conditions—such as textual instructions, keyboard inputs, or video prompts—the model can adapt its predictions according to these external signals, enabling interactive and controlled video generation. Textual instructions are incorporated through the Instruction Fusion Expert, which employs cross-attention mechanisms to align video outputs with semantic guidance. Keyboard operations are handled by the Operation Fusion Expert, which projects input signals into latent features and predicts affine transformation parameters for feature modulation. Additionally, video prompts—such as canny-edge maps, motion vectors, and pose sequences—are integrated through additive fusion at the embedding layer, providing rich auxiliary visual context. To simulate diverse use cases, control signals are applied probabilistically: in some scenarios, no control signals are provided, while in others, combinations of text, keyboard inputs, and video prompts are used to guide the model’s behavior.
>
>
>  - **Training Configuration**: The Autoressive Tuning phase is dedicated to fine-tuning InstructNet for control video extension content, with the base model frozen to preserve previously learned generation and video extension ability. Training is conducted on videos with a fixed resolution of 720p and a duration of 4 seconds, focusing solely on the video extension task for all iterations. Unlike the bucket-based sampling strategy in the first phase, this phase uses fixed parameters to ensure consistency. The masking mechanism ensures that unobserved frames are initialized with noise, while the first frame remains as a temporal reference. The control signal injection probabilities are carefully balanced to include diverse scenarios, ranging from no control signals to combinations of text, keyboard inputs, and video prompts.
>
> We apologize for not responding to Q1 and W1 at this time. We will respond to these questions and concerns as soon as possible after finishing the experiment.

---

> ### Author Response · Authors · 2024-11-23
> **Response to Reviewer 1M8F (Part 3/3)**
>
> **[W1] Generation Speed and Performance**
>
> We greatly appreciate the reviewers' perspectives. The core advantage of GameGen-X lies in its ability to generate high-quality open-domain game scenes with interactive control over character and environment dynamics. This enhances the creativity of the generated content and provides novel experiences. In this part, we will supplement our work with experiments and analyses related to generation speed and performance. Specifically, we conducted 30 open-domain generation inferences on a single A800 and a single H800 GPU, with the CUDA environment set to 12.1. We recorded the time and corresponding FPS, and reported the VBench metrics, including subject_consistency (SC), background_consistency (BC), dynamic_degree (DD), aesthetic_quality (AQ), imaging_quality (IQ), and overall. We have uploaded new videos to demonstrate the visualization results at different resolution sampling steps (https://drive.google.com/file/d/16ibysz0LpdmPvew2elD4OcWu3GLooZok/view?usp=sharing).
>
> 1. **The Inference Time**
> | Resolution | Frames | Sampling Steps | Time (A800) | FPS (A800) | Time (H800) | FPS (H800) |
> |------------|--------|----------------|-------------|------------|-------------|------------|
> | 320 x 256  | 102    | 10             | ~7.5s/sample   | 13.6       | ~5.1s/sample    | 20.0       |
> | 848 x 480  | 102    | 10             | ~60s/sample     | 1.7        | ~20.1s/sample   | 5.07       |
> | 848 x 480  | 102    | 30             | ~136s/sample    | 0.75       | ~44.1s/sample   | 2.31       |
> | 848 x 480  | 102    | 50             | ~196s/sample   | 0.52       | ~69.3s/sample   | 1.47       |
> | 1280 x 720 | 102    | 10             | ~160s/sample    | 0.64       | ~38.3s/sample   | 2.66       |
> | 1280 x 720 | 102    | 30             | ~315s/sample   | 0.32       | ~57.5s/sample   | 1.77       |
> | 1280 x 720 | 102    | 50             | ~435s/sample    | 0.23       | ~160.1s/sample  | 0.64       |
>
> In terms of generation speed, higher resolutions and more sampling steps result in increased time consumption. Although GameGen-X is primarily trained at 848x480 and 1280x720 resolutions, for alignment with GameNGen, we also included inference tests at an untrained resolution of 320x256. Similar to the conclusions found in GameNGen, the model generates videos with acceptable imaging quality and relatively high FPS at lower resolutions and fewer sampling steps (e.g., 320x256, 10 sampling steps). We plan to introduce more optimization algorithms and technical solutions in the future to maintain high FPS even at higher resolutions. Additionally, we plan to explore how to unify single-frame rendering and clip generation to further enhance creativity, generation quality, and real-time operability.
>
> 2. **Performance Analysis**
> | Resolution | Frames | Sampling Steps | SC    | BC    | DD  | AQ    | IQ    | Average |
> |------------|--------|----------------|-------|-------|-----|-------|-------|---------|
> | 320 x 256  | 102    | 10             | 0.944 | 0.962 | 0.4 | 0.563 | 0.335 | 0.641   |
> | 848 x 480  | 102    | 10             | 0.947 | 0.954 | 0.8 | 0.598 | 0.389 | 0.737   |
> | 848 x 480  | 102    | 30             | 0.964 | 0.960 | 0.9 | 0.645 | 0.573 | 0.808   |
> | 848 x 480  | 102    | 50             | 0.955 | 0.961 | 0.9 | 0.615 | 0.570 | 0.800   |
> | 1280 x 720 | 102    | 10             | 0.957 | 0.963 | 0.3 | 0.600 | 0.453 | 0.655   |
> | 1280 x 720 | 102    | 30             | 0.954 | 0.956 | 0.7 | 0.617 | 0.558 | 0.757   |
> | 1280 x 720 | 102    | 50             | 0.959 | 0.959 | 0.8 | 0.657 | 0.584 | 0.812   |
>
> From the table, we can observe that increasing the number of sampling steps generally improves visual quality at the same resolution, as reflected in the improvement of the Overall score. For example, at resolutions of 848x480 and 1280x720, increasing the sampling steps from 10 to 50 significantly improved the Overall score, from 0.591 to 0.800 and from 0.655 to 0.812, respectively. This suggests that higher resolutions typically require more sampling steps to achieve optimal visual quality.
>
> On the other hand, we qualitatively studied the generated videos. We observed that at a resolution of 320p, our model can produce visually coherent and texture-rich results with only 10 sampling steps. As shown in the accompanying video, details such as road surfaces, cloud textures, and building edges are generated clearly. At this resolution and number of sampling steps, the model can achieve 20 FPS on a single H800 GPU.
>
> We also observed the impact of sampling steps on the generation quality at 480p/720p resolutions. At 10 sampling steps, we observed a significant enhancement in high-frequency details. Sampling with 30 and 50 steps not only further enriched the textures but also increased the diversity, coherence, and overall richness of the generated content, with more dynamic effects such as cape movements and ion effects. This aligns with the quantitative analysis metrics.

---

> ### Author Response · Authors · 2024-11-23
> **Response to Reviewer 1M8F (Part 3/3)**
>
> We greatly appreciate the reviewers' valuable feedback, which provided us with the opportunity to explain and demonstrate the relationship between generation speed and model performance as well as the details of instruction tuning. We have added details of instruction tuning in the appendix including the data acquisition, design details of InsturctNet, and training strategies. We will include more detailed analyses and visualization results in the appendix of future versions of the paper.
>
> Reference:
>
> [1] VBench: Comprehensive Benchmark Suite for Video Generative Models, CVPR, 2024.
>
> [2] VBench++: Comprehensive and Versatile Benchmark Suite for Video Generative Models, 2024.
>
> [3] Diffusion Models Are Real-Time Game Engines, 2024.

---

> ### Author Response · Authors · 2024-11-25
>
> Dear Reviewer 1M8F,
>
> Thank you for your valuable time and effort in reviewing our work. With only 2 days remaining, we would greatly appreciate receiving your feedback on our response to facilitate further discussion. If any aspects of our explanation are unclear, please feel free to let us know. We would be happy to provide any additional clarification promptly before the discussion deadline.
>
> Thank you once again for your invaluable comments and consideration, which are greatly beneficial in improving our paper.
>
> Best,
>
> GameGen-X Team

---

> > ### Comment · Reviewer_1M8F · 2024-11-26
> > **Thank you for reply**
> >
> > I appreciate the authors taking the time to provide detailed responses to all the questions raised. The in-depth discussion on Complex Game Event Simulation reflects a thoughtful approach to modeling extended problems, and the supplementary experiments on Generation Speed and Performance are thorough and well-executed. While I believe this paper is above the acceptance threshold, there still remains room for improvement in the proposed methods and the presented experimental results to fully align with the paper's claims of achieving video generation for interactive open-world gameplay simulation.

---

> > ### Author Response · Authors · 2024-11-26
> >
> > Thank you for taking the time to review our paper and give feedback. We appreciate your recognition of our efforts to tackle the challenges in simulating complex game events, as well as your valuable insights into the performance and generation speed of our approach. Your thoughtful review has been instrumental in refining our work. Thank you for your thoughtful review and comments!

---

### Official Review · Reviewer_WQcm · 2024-11-01

**Soundness:** 3
**Presentation:** 1
**Contribution:** 3
**Rating:** 8
**Confidence:** 5

**Summary:**

In this paper, the authors introduce a diffusion transformer model aimed at generating and controlling video game sequences in challenging 3D open-domain game worlds. The authors also present the gameplay dataset they collected to train the model, OGameData. The dataset has 1 million video clips from across 150 videos, annotated with text descriptions using GPT4o. Both the model and the dataset have 2 components. One for text-to-video generation (OGameData-GEN and the pretrained foundation model) and one for instruction tuning (OGameData-INS and InstructNet).

**Strengths:**

- The work is original in the sense that is the first main contribution to the field in terms of interactive video game generation in large scale, complex, open worlds

- It is great to see such examples of tackling complex research environments at scale, with potential direct benefits to the game development process.

- The author(s) introduce a complex system, both in terms of the dataset it required for training (including a resource intensive collection and curation process), as well as in terms of the pretrained foundation model and the interactive control network, allowing users to control the output via either text or mouse and keyboard inputs

**Weaknesses:**

- There is one strong concern I have regarding the data collection process for the OGameData dataset. My score highly depends on evidence that data collection will pass the ethics review and there is evidence provided on the consent given by the humans that produced the data. There should be understanding and agreement for it to be used for research purposes and open sourced. Please elaborate on how the data for OGameData has been collected? In Appendix B.1. you mention selecting online video websites as one of the primary sources. It would be good to know:
   - The exact sources of the video data
   - Any agreements or permissions obtained from video creators and game studios
   - The ethical review process they followed, if any
   - How you plan to address potential copyright or licensing issues

- It is unclear why all UI elements have been removed from the dataset, it would be great to gain further clarity on that from the author(s). In a lot of open-world gameplay , the player relies on UI element understanding, such as health levels, navigation information via mini maps, affordance of actions to take, inventory etc.
   - How does this decision impact the model's ability to generate realistic gameplay experiences?
   - Do you plan to incorporate UI elements in future iterations of the model?

- Please correct me if I missed this, but the main body of the paper does not clearly indicate that all the data and the generation is within the constraints of a single agent. What is the model’s ability to model other dynamic environment elements (NPCs, other players, moving vehicles etc.)? It would be good to:
   - Explicitly state whether the model is limited to single-agent scenarios
   - If so, discuss the implications of this limitation on the model's applicability
   - If not, provide details on how the model handles multiple dynamic elements in the environment

- The paper is dense, so it took a while to disambiguate if the main body of the paper provides sufficient detail for capturing the core contributions of the paper or if a lot of essential details were included in the appendix.

**Questions:**

Clarification Questions:

-	Is it the correct understanding that the OGameData-GEN dataset comprises of data from 150 video games, whilst the OGameData-INS dataset contains only a subset of 5 game titles? Without checking Appendix B for clarification, it is difficult for the reader to grasp these details from the main body of the paper.
-	For video clip compression (Section 3.2) it would be good to add more details about the size of the latent representation z, as well as the resolution of the video clips used in training.
-	How were the spatial and temporal downsampling factors determined (s_t, s_h, s_w)?
-	In section 3.2, under unified video generation and continuation, you mention incorporating bucket training, classifier-free diffusion guidance and rectified flow for better generalization performance – did you run any ablation studies to understand better the impact of introducing these 3 components?
-	What are the values x for context length that you considered for video continuation?
-	In the InstructNet design, what were the considerations for choosing N (the number of InstructNet blocks)? Did you experiment with different values?
-	For the multi-modal experts introduced in Section 3.3, what are the sizes considered for the instruction embeddings and keyboard input embeddings (f_I and f_O)?
-	Under Interactive control, you mention the incorporation of video prompts V_p enhances the model’s ability to generate motion-consistent frames – did you conduct any experiments or ablations to measure the observed improvement?
-	Is there a mention on the computational resources required to store and stream the data for training, as well as for training the foundation model and InstructNet? It would be a useful proxy for people planning to reproduce the work.
-	Similarly, is there any information presented on the inference times of GameGen-X?
-	In evaluating the control ability you mention using both human experts and PLLaVa. What is the ratio between the 2 evaluation modalities?
-	On qualitative results for generation, apart from the discussion on diversity, would it be possible to elaborate on the length and consistency of the videos generated by GameGen-X? From the demo videos included, most are under <8-10 seconds.
-	In the ablation studies (Tables 4 and 5), there seem to be no DD and IQ metrics – what is the reason for that?

Minor comments/Suggestions:

-	In Section 2.2, it would be good to specify the human experts’ level of familiarity with the titles and elaborate on how the GPT-4o text annotations were checked for quality and accuracy.
-	In Section 3.3, you introduce the c condition under the Interactive Control subsection, but it is mentioned beforehand in Multi-modal experts. It would be clearer to the reader to introduce the structure of c under the Multi-modal experts’ subsection, where it appears for the first time.
-	For readability, it would be good to illustrate z, the latent variable in Figure 4.
-	It would be useful to include a more detailed explanation on the choice of baselines in the experiments Section. For example, Mira is not included under the results for control ability, is it because it does not have support for it? It would be good to clarify that.
-	It would be good to link to Appendix D (Discussion) when mentioning remaining challenges in the conclusion.
-	I know this appeared after the submission deadline, but it would be worth adding to the related work section as a referece: https://www.decart.ai/articles/oasis-interactive-ai-video-game-model

**Details Of Ethics Concerns:**

As stated in the weaknesses section, I would like to see the an ethics review approval for all the data included in the OGameData dataset, especially as the author(s) plan to opensource it. [Edit: authors addressed concerns in their response]

---

> ### Author Response · Authors · 2024-11-21
> **Response to Reviewer WQcm (Part 1/4)**
>
> We sincerely feel grateful for the reviewer's insightful and constructive comments regarding our paper related to dataset design, and multi-agent generation, etc. We would like to take this opportunity to clarify the following points and update the Appendix in our revised manuscript:
>
> **[W1] Dataset and Compliance**
>
> We understand the reviewer's concerns regarding our data collection process and appreciate attention to the ethical and legal considerations.  Regarding our dataset, we followed best practices established by previous works [1-6] and ensured compliance with the fair use and non-commercial policies of the platforms.
>
> Specifically, our data sources include both internet-collected and locally gathered data. The Internet data collection methods were inspired by established datasets like Panda-70M [5] and MiraData [6]. We collected gameplay videos from YouTube in compliance with platform regulations, while also performing comprehensive cleaning and integration of game and 3D-rendered videos from Panda-70M [5] and MiraData [6]. Additionally, we recorded local gameplay footage with proper permissions and respect for copyright laws. The local data collection was primarily focused on constructing OGameData-INS, aimed at capturing control signals and corresponding gameplay footage to meet the needs of character control training.
>
> Additionally, in our future data open-sourcing efforts, we will align with the data open-sourcing paradigms of existing works, providing only supplementary textual annotations, URLs, and timestamps of the videos  [1-6] to ensure adherence to regulations. We also have included detailed information regarding data usage and protocols, and a copyright compliance statement in the appendix to ensure compliance. We also followed the ethical review process of our institution. We will handle the open sourcing of the dataset and model with caution, enforcing agreements to ensure that this project is used solely for research purposes and not for commercialization. Through these measures, we aim to ensure the legality and compliance of the data, providing the research community with high-quality annotated game content video data, finally thereby benefiting the research community.
>
> **[W2] UI Elements and Gaming System Simulation**
>
> **The Reason for UI Element Removal**: Our current focus is on generating game scenes and characters, as well as controlling corresponding events and actions. In the early stages of development, we did not specifically filter out UI elements. However, we found that UI elements, which vary across different games, often caused the generated videos to appear cluttered and detract from the core visual aspects of the game scenes (for an example, see the sample video: https://drive.google.com/file/d/1Te95mJf5tdHpmUOqCrwdfDCMmhJD8168/view?usp=sharing). To ensure that the generated content focused on the game environment and character interaction, we made the decision to filter out large UI elements during data cleaning.
>
> **Simuliting Gaming System**: We believe that generating realistic games solely based on visual conditions is very challenging due to the complex interactions, storylines, and progression systems involved in games. In the current version based on DiT, we are more focused on whether the model can achieve a certain degree of interactive rendering functionality, generating new scenes and characters, and allowing interaction with users.  Generating a truly realistic gameplay experience is a complex task, given the intricate systems involved in games, including interactions, storylines, and progression. Therefore, in the current version based on DiT, we are prioritizing the development of interactive rendering functionality and the generation of new scenes and characters, allowing for basic user interaction.
>
> In our future versions, we plan to explore how to simulate a real game system. At that time, we will consider reintroducing UI elements into the screen through hard coding and supporting new game features. For example, we could use large language models (LLMs) to design the overall game storyline and fine-grained cutscene scenarios [7-9]. Sound elements could be embedded in the generation process [10]. For complex game trees, game systems, progression elements, and dynamic NPC interactions, LLMs might serve as agents to construct a dynamic world and system described by text [11]. In this scenario, the game LLM could work in tandem with a diffusion model, with one serving as the core of the game system and the other as the game renderer. The LLM would handle the game's logic and interactions, while the diffusion model would generate high-quality visual content. This collaborative approach could significantly enhance the complexity and playability of the game. Future work might explore a hybrid generation scheme based on Transfusion [12] to unify the entire game.

---

> ### Author Response · Authors · 2024-11-21
> **Response to Reviewer WQcm (Part 2/4)**
>
> **[W3] Multi-agent Ability**
>
> Thank you for your thoughtful feedback. To clarify, our dataset and model generation are not limited to single-subject scenes. GameGen-X is capable of handling multi-subject scenes, such as those involving NPCs, vehicles, and multiple protagonists (please see the sample video,  https://drive.google.com/file/d/1-PLP8sohLyI5Wsn_gnOPbDFjD-_c5Ppq/view?usp=sharing.). However, we have observed that the quality of multi-agent scene generation is not yet as high as that of single-subject scenes. This is primarily due to data distribution, where single-agent scenes are predominant, and NPCs, vehicles, and other dynamic elements appear less frequently and for shorter durations. As a result, the model has had less opportunity to learn and refine its generation of these elements.
> Moreover, the core purpose of GameGen-X is to assist in the game scene and character designs, creating characters in open-domain game content and enabling interaction with them. Therefore, our work has not explicitly focused on multi-agent scene design. Future work could improve this by introducing layout conditions [13,14], which would explicitly control the trajectories and appearances of NPCs, vehicles, and other elements, thereby achieving better dynamic object generation. It may enhance the quality of multi-subject scene generation, enabling GameGen-X to perform more effectively in handling complex scenes.
>
> **[W4] Improving paper structure:**
>
> Thank you for your suggestions. In the current version, we have provided more useful information, such as model design, data collection, and copyright notices. We have also integrated important information, such as data sources, into the main text. We will continue to revise and improve the paper in the future.
>
> **Response to [Q1]**:
> Yes. The OGameData-GEN dataset consists of data from 150 video games, providing a broad range of content to support game content generation. In contrast, the OGameData-INS dataset contains a smaller subset of 5 game titles, specifically chosen to align with our goal of refining the model’s ability to control the environment and tasks. OGameData-INS was constructed to focus on high-quality, task-specific content, inspired by [20]. We will revise the main text to make this distinction clearer, as we agree it may not be immediately evident without checking Appendix B.
>
> **Response to [Q2]**:
>  We introduced bucket training, which can support multiple resolutions and frames, leading to varied latent representation sizes. We have updated the paper to include more details on the compression ratio, latent dimension, downsampling information in the base model, and the resolution of the video clips used in training. Please refer to the revised Implementation and Design Details in the Appendix for this additional information.
>
> **Response to [Q3]**:
> The values for (s_t, s_h, s_w) are set to (4,8,8), following the convention established by previous studies [15-17]. Specifically, the temporal dimension is compressed by a factor of 4, which allows us to handle longer video sequences during training. However, we have found that this temporal compression can result in suboptimal modeling of fast-moving objects, such as NPCs and other dynamic elements. Future work will aim to strike a better balance between compressing the temporal dimension and effectively capturing fast-moving objects. Additionally, we are exploring new techniques to improve the model’s ability to capture and represent these rapidly moving targets. The specific values for (s_t, s_h, s_w) have been updated in the latest appendix.
>
> **Response to [Q4]**:
> In our ablation study section, we present the results of the bucket training ablation, which demonstrates its effectiveness in improving model performance. The introduction of rectified flow (Reflow) significantly enhanced the visual quality, with noticeable improvements compared to earlier versions of the model. However, regarding classifier-free diffusion guidance (CFG), while we did not conduct a separate ablation study specifically for it, previous works [18] suggest that CFG has a strong potential to improve visual quality. Therefore, we decided to incorporate it based on its demonstrated effectiveness in related studies.
>
> **Response to [Q5]**:
> During inference time, we set the context length x to 5 frames.

---

> ### Author Response · Authors · 2024-11-21
> **Response to Reviewer WQcm (Part 3/4)**
>
> **Response to [Q6]:**
> In the design of InstructNet, we drew inspiration from the Pixart-delta architecture [19] and found that inserting all modules led to memory overflow issues. Therefore, when selecting the number of InstructNet blocks (N), we aimed to strike a balance between model performance and memory usage. Specifically, we chose to use half of the blocks used in the foundation model, to avoid memory issues while still achieving reasonable performance. We did experiment with different configurations, but this approach provided the most stable performance under our memory constraints. In future work, we plan to further explore variations in the number of blocks (N) and the insertion patterns (e.g., skip connections) to assess their impact on both performance and memory usage.
>
> **Response to [Q7]:**
> The dimension of these two embeddings is 1152, while the patch number is different based on the input length.
>
> **Response to [Q8]:**
> To clarify, the purpose of video prompts V_p is to map real-world content to virtual game scenes or perform global edits based on previously generated content. Although this is not the core functionality of GameGen-X—namely, generating open-domain game content and interacting with it—we recognize the potential of this feature to expand the model's use cases.
>
> **Response to [Q9]:**
>  The total computational resources required for data and checkpoint storage are approximately 50 terabytes (50T).
>
> **Response to [Q10]:**
> We will provide more detailed information later.
>
> **Response to [Q11]:**
> In our evaluation of control ability, each sample was evaluated by both human experts and PLLaVa at a ratio of 10:1. Specifically, we used 10 human experts and 1 PLLaVa model for each evaluation. The detailed evaluation procedure and results can be found in Appendix D.3 Human Evaluation Details.
>
>
> **Response to [Q12]:**
> GameGen-X is capable of generating longer videos by using a streaming-style approach, thanks to the unified training of both text-to-video and video continuation models. We have demonstrated this capability in a 10x-accelerated streaming generation example, where GameGen-X generates a video of a man riding a horse in a forest using around 100 control signal sequences to guide the clip.
>
> In our experiments, we found that GameGen-X can generate videos up to around 30 minutes in length. However, we did observe some challenges with maintaining consistency over longer durations. Specifically, after around 20 minutes of generation, we noticed that the main character’s clothing may change, likely due to the absence of explicit conditions to ensure character consistency over extended timeframes.
> This suggests that while GameGen-X can generate long videos, additional conditions and controls will be needed in future work to maintain character consistency and improve the overall coherence of long-range video sequences. This is a problem we plan to address in subsequent iterations of the model.
>
> **Response to [Q13]:**
> The DD and IQ metrics were not included in the main tables due to space constraints. However, we will provide the complete results in a supplementary table below for readers' reference.
> Regarding the IQ metric, the results across different ablation experiments are quite similar, suggesting that the various modules have a consistent impact on image quality. On the other hand, for the DD metric, we observed that the baseline score is relatively low, while the corresponding SC metric is notably higher. This discrepancy may indicate that the model performs well in certain aspects of scene coherence (SC) despite having a lower DD score.
>
> | Method                  | Resolution | Frames | FID ↓ | FVD ↓  | TVA ↑ | UP ↑  | MS ↑  | DD ↑  | SC ↑  | IQ ↑  |
> |-------------------------|------------|--------|-------|--------|-------|-------|-------|-------|-------|-------|
> | w/ MiraData             | 720p       | 102    | 303.7 | 1423.6 | 0.70  | 0.48  | 0.99  | 0.84  | 0.94  | 0.51  |
> | w/ Short Caption        | 720p       | 102    | 303.8 | 1167.7 | 0.53  | 0.49  | 0.99  | 0.78  | 0.94  | 0.49  |
> | w/ Progression Training | 720p       | 102    | 294.2 | 1169.8 | 0.68  | 0.53  | 0.99  | 0.68  | 0.93  | 0.51  |
> | Baseline                | 720p       | 102    | 289.5 | 1181.3 | 0.83  | 0.67  | 0.99  | 0.64  | 0.95  | 0.49  |
>
> | Method                  | Resolution | Frames | SR-C ↑ | SR-E ↑ | UP ↑  | MS ↑  | DD ↑  | SC ↑  | IQ ↑  |
> |-------------------------|------------|--------|--------|--------|-------|-------|-------|-------|-------|
> | w/o Instruct Caption    | 720p       | 102    | 31.6%  | 20.0%  | 0.34  | 0.99  | 0.82  | 0.87  | 0.41  |
> | w/o Decomposition       | 720p       | 102    | 32.7%  | 23.3%  | 0.41  | 0.99  | 1.00  | 0.88  | 0.41  |
> | w/o InstructNet         | 720p       | 102    | 12.3%  | 17.5%  | 0.16  | 0.98  | 0.98  | 0.86  | 0.43  |
> | Baseline                | 720p       | 102    | 45.6%  | 45.0%  | 0.50  | 0.99  | 0.78  | 0.90  | 0.42  |

---

> ### Author Response · Authors · 2024-11-21
> **Response to Reviewer WQcm (Part 4/4)**
>
> **Response to [C1]:**
> Thank you for your suggestion. To ensure high-quality data collection, we established a set of stringent selection criteria and provided example videos to help guide the human experts, even if they were not familiar with the specific game titles in the collected videos. The selection criteria included the following aspects:
>
> 1. Game Release Date
> 2. Game Genre (e.g., RPG, Action)
> 3. Perspective (e.g., first-person or third-person view)
> 4. Shot Type (e.g., long, medium, close shots)
> 5. Camera Type (e.g., free camera or gameplay view)
> 6. UI Proportion (e.g., filtering out videos with significant UI elements)
> 7. Controlled Subject (e.g., character, animal, vehicle)
> 8. Action Complexity (e.g., identifying complex action elements such as climbing, jumping, or interactions in action-adventure games)
>
> Regarding the verification of GPT-4’s text annotations, we placed a strong emphasis on prompt design to enhance the quality and accuracy of the annotations. Initially, we used 100 clips to design and refine our prompts. Through multiple iterations, we improved the prompts to ensure they could generate structured and dynamic descriptions of the video content. These prompts were carefully crafted to include both high-level overviews and detailed descriptions, ensuring that the generated annotations were consistent with the videos’ content. We also embedded some meta-information annotated during the video screening process into the prompts to avoid hallucination issues and improve accuracy.
>
> **Response to [C2]:**
> Thank you for your suggestion. We appreciate your feedback. In the revised manuscript, we have updated the structure of the c condition and moved its introduction to the Multi-modal experts subsection, where it first appears, to improve clarity for the reader.
>
> **Response to [C3]:**
> Thank you for your suggestion. We appreciate your feedback. We will update Figure 4 to include an illustration of the latent variable z for improved readability.
>
> **Response to [C4]:**
> Thank you for your suggestion. We appreciate your feedback. Mira is not included under the results for control ability because it does not support this functionality. We clarified this in the revised manuscript.
>
> **Response to [C5]:**
> Thank you for your suggestion. We have updated the conclusion to include a reference to Appendix D (Discussion) when mentioning the remaining challenges.
>
> **Response to [C6]:**
> Thank you for your suggestion. We have cited and discussed this valuable work in the related work section of the revised manuscript.
>
> Reference:
>
> [1] Openvid-1m: A large-scale high-quality dataset for text-to-video generation, 2024.
>
> [2] Swap Attention in Spatiotemporal Diffusions for Text-to-Video Generation, 2023.
>
> [3] Internvid: A large-scale video-text dataset for multimodal understanding and generation, 2023.
>
> [4] Vript: A Video Is Worth Thousands of Words, NeurIPS, 2024.
>
> [5] MiraData: A Large-Scale Video Dataset with Long Durations and Structured Captions, NeurIPS, 2024.
>
> [6] Panda-70m: Captioning 70m videos with multiple cross-modality teacher, CVPR, 2024.
>
> [7] Videodirectorgpt: Consistent multi-scene video generation via LLM-guided planning, 2023.
>
> [8] LLM-grounded Video Diffusion Models, ICLR, 2024.
>
> [9] VideoStudio: Generating Consistent-Content and Multi-Scene Videos, ECCV, 2024.
>
> [10] Audiogpt: Understanding and generating speech, music, sound, and talking head, AAAI, 2024.
>
> [11] Large language models and games: A survey and roadmap. 2024.
>
> [12] Transfusion: Predict the next token and diffuse images with one multi-modal model, 2024.
>
> [13] DrivingDiffusion: Layout-Guided Multi-view Driving Scenarios Video Generation with Latent Diffusion Model, ECCV, 2025.
>
> [14] Boximator: Generating Rich and Controllable Motions for Video Synthesis, ICML, 2024.
>
> [15]  Open-sora-plan, April 2024. URL https://doi.org/10.5281/zenodo.10948109.
>
> [16] Open-sora: Democratizing efficient video production for all, March 2024b. URL https://github.com/hpcaitech/Open-Sora.
>
> [17] Language Model Beats Diffusion-Tokenizer is key to visual generation, ICLR, 2024.
>
> [18] Classifier-free diffusion guidance, 2022.
>
> [19] Pixart-{\delta}: Fast and controllable image generation with latent consistency models, 2024.
>
> [20] Pandora: Towards General World Model with Natural Language Actions and Video States, 2024.

---

> ### Author Response · Authors · 2024-11-22
> **Response to [Q10]**
>
> Thank you for your question and patience. We provide the inference time below, which is calculated by 30 times generation. We tested our model on two kinds of mainstream GPU cards, A800 and H800.
>
> | Resolution | Frames | Sampling Steps | Time (A800) | FPS (A800) | Time (H800) | FPS (H800) |
> |------------|--------|----------------|-------------|------------|-------------|------------|
> | 320 x 256  | 102    | 10             | ~7.5s/sample   | 13.6       | ~5.1s/sample    | 20.0       |
> | 848 x 480  | 102    | 10             | ~60s/sample     | 1.7        | ~20.1s/sample   | 5.07       |
> | 848 x 480  | 102    | 30             | ~136s/sample    | 0.75       | ~44.1s/sample   | 2.31       |
> | 848 x 480  | 102    | 50             | ~196s/sample   | 0.52       | ~69.3s/sample   | 1.47       |
> | 1280 x 720 | 102    | 10             | ~160s/sample    | 0.64       | ~38.3s/sample   | 2.66       |
> | 1280 x 720 | 102    | 30             | ~315s/sample   | 0.32       | ~57.5s/sample   | 1.77       |
> | 1280 x 720 | 102    | 50             | ~435s/sample    | 0.23       | ~160.1s/sample  | 0.64
>
> In terms of generation speed, higher resolutions and more sampling steps result in increased time consumption. Similar to the conclusions found in GameNGen, the model generates videos with acceptable imaging quality and relatively high FPS at lower resolutions and fewer sampling steps (e.g., 320x256, 10 sampling steps). We plan to introduce more optimization algorithms and technical solutions in the future to maintain high FPS even at higher resolutions (https://drive.google.com/file/d/16ibysz0LpdmPvew2elD4OcWu3GLooZok/view?usp=sharing). Additionally, we plan to explore how to unify single-frame rendering and clip generation to further enhance creativity, generation quality, and real-time operability.

---

> ### Author Response · Authors · 2024-11-25
>
> Dear Reviewer WQcm,
>
> Thank you for your valuable time and effort in reviewing our work. With only 2 days remaining, we would greatly appreciate receiving your feedback on our response to facilitate further discussion. If any aspects of our explanation are unclear, please feel free to let us know. We would be happy to provide any additional clarification promptly before the discussion deadline.
>
> Thank you once again for your invaluable comments and consideration, which are greatly beneficial in improving our paper.
>
> Best,
>
> GameGen-X Team

---

> > ### Comment · Reviewer_WQcm · 2024-11-25
> >
> > Thank you very much for taking the time to offer detailed responses to all the questions raised in the review and for adding the suggested changes. The implementation details are incredibly useful for the community for reproducibility. As other reviewers pointed out, my main concern regarded the data collection procedure, as the initial submission did not include enough detail on the collection sources and the copyright compliance. The authors have addressed these concerns in their updates, so I am not increasing my score from 5 to 8.

---

> > > ### Author Response · Authors · 2024-11-26
> > >
> > > Thank you for your detailed review and constructive feedback. We appreciate your recognition of our efforts to address the challenges you mentioned, particularly in the data collection procedure and copyright compliance. We are pleased to hear that our updates have satisfactorily addressed your concerns. Your insightful comments have been invaluable in refining our work, and we look forward to further improvements. Thank you again for your thoughtful review!

---

### Official Review · Reviewer_bSLg · 2024-11-04

**Soundness:** 3
**Presentation:** 3
**Contribution:** 3
**Rating:** 8
**Confidence:** 4

**Summary:**

This paper proposes GameGen-X, a diffusion based model for open-world game generation. Specifically, this paper proposes two detailed crafted datasets: OGameData-Gen and OGameData-Ins. OGameData-Gen is used to pre-train the diffusion model to understand and generate continuous open-world game-style videos, where OGameData-Ins is used to instruct tune the model to understand special inputs (e.g., keyboard inputs) to better control the continuation of the game generation based on some input frames. The dataset is well-curated to have 1M videos, with multiple filtering metrics and human-in-loop filtering to maintain the high quality. Then, this paper trains a video diffusion model with two-stage training on the two datasets for open-world game generation. Specifically, an instruct net is designed to take in different special inputs. Empirically, on their provided evaluation dataset, GameGen-X achieves superior performance than other state-of-the-art video diffusion models (e.g., kling).

**Strengths:**

1. A large well-curated dataset for open-world video games. The curation of the dataset contains filtering on different aspects (e.g., semantic alignment, motion), which results in a high-quality large-scale dataset.
2. The idea to build a video diffusion model for open-world video games is essentially interesting, and the results and demo videos are impressive. Besides, quantitatively, the proposed approach also achieves better performance than other SoTA diffusion models.
3. Detailed ablation studies demonstrate the effectiveness of the proposed component (i.e., two-stage training strategy and the design of the instructnet).
4. This paper is well-written and easy to follow.

**Weaknesses:**

1. One main concern is the proposed GameGen-X is specially fine-tuned/designed for open-world video games, while other diffusion models compared (e.g., kling) are trained for a general text-to-video generation, which makes the comparison somehow unfair to other models.
2. The qualitative examples in the website demo for game generation (e.g., under generation comparison) don't seem to look much better than other models (e.g., cogvideoX).

**Questions:**

Please refer to weakness.

---

> ### Author Response · Authors · 2024-11-21
> **Response to Reviewer bSLg (Part 1/2)**
>
> We sincerely appreciate the reviewer's insightful and constructive comments regarding our paper related to the fairity of comparison and qualitative examples. We have followed your suggestions and updated them in the appendix of our revised manuscript. We would like to take this opportunity to clarify the following points:
>
> **[W1]  Comparison Fairness**
>
> We appreciate the reviewer’s concern regarding the fairness of comparing GameGen-X. In our experiments, we compared GameGen-X with four other models (OpenSora-Plan, OpenSora, MiraDiT, and CogVideo-X), as well as five commercial models (Gen-2, Kling 1.5, Tongyi, Pika, and Luma). Several of these models, such as OpenSora-Plan, OpenSora, and MiraDiT, indicate that their data sources include Panda-70M and MiraData, which contain a significant number of 3D game/engine-rendered scenes.
> Additionally, while CogVideo-X and commercial models do not disclose training data, their outputs suggest familiarity with similar visual domains. We hope that this clarification of model capabilities will address the reviewer's concerns. Although there are no perfectly comparable works in game content generation, we have strived to ensure experiment fairness in terms of the model selection.
>
> To the best of our knowledge, our paper is a pioneering work attempting to systematically address the problem of open-domain game content generation and its interactive control. Therefore, we aim to illustrate our efforts in building this problem from the ground up. Comparisons with other models are intended to illustrate our special capabilities in game video generation, open-domain generation, and interactive control, rather than to claim absolute superiority in all visual generation metrics or model abilities.
>
> Additionally, to disentangle the effects of data and framework design, we sampled 10K subsets from both MiraData (which contain high-quality game video data) and OGameData and conducted a set of ablation experiments with OpenSora (a state-of-the-art open-sourced video generation framework). Due to the time limitation, we quickly verified the decoupled contribution based on these two additional experiments. We could compare more experiments in the future version. The results are as follows:
>
>   | Metric                | FID   | FVD    | TVA | UP | MS | DD | SC  | IQ | Alignment Metrics | Quality Metric |
>   |-----------------------|-------|--------|-----|----|----|----|-----|----|--------------------------|-----------------------|
>   | Ours / OGameData-Subset      | 289.5 | 1181.3 | 0.83 | 0.67 | 0.99 | 0.64 | 0.95 | 0.49 | 735.4                    | 0.76                  |
>   | OpenSora / OGameData-Subset  | 295.0 | 1186.0 | 0.70 | 0.48 | 0.99 | 0.84 | 0.93 | 0.50 | 740.5                    | 0.74                  |
>   | Ours / MiraData-Subset       | 303.7 | 1423.6 | 0.57 | 0.30 | 0.98 | 0.96 | 0.91 | 0.53 | 863.65                   | 0.71                  |
>
> As shown in the table above, we supplemented a comparison with OpenSora on MiraData. In comparing Alignment Metrics(averaged FID and FVD scores) and Quality Metrics (averaged TVA, UP, MS, DD, SC, and IQ scores), our framework and dataset demonstrate clear advantages. Aligning the dataset (row 1 and row 2), it can be observed that our framework (735.4, 0.76) outperforms the OpenSora framework (740.5, 0.74), indicating the advantage of our architecture design. Additionally, fixing the framework, the model training on the OGameData-Subset (735.4, 0.76) surpasses the model training on MiraData-Subset (863.65, 0.71), highlighting our dataset's superiority in the gaming domain. These results confirm the efficacy of our framework and the significant advantages of our dataset.
>
> To further ensure fairness, contribution, and generalization, we have updated multiple sets of in-domain and open-domain generation samples in the Qualitative Comparison section on our project website (3a2077.github.io). These samples highlight: a) The existing open-sourced models can generate game scene videos, owing to Panda-70M and MiraData. 2) These samples show that our model performs better in generating known game scenes and creating new game content. Therefore, combined with the table above and the in-domain, open-domain, and streaming generation results (https://drive.google.com/file/d/1vZE4SKzLDqfErBV0B5MAbVHUizVZysdS/view?usp=sharing) demonstrate our contributions, as well as the generalization capability of our model (i.e., creating new game scenes and content).

---

> ### Author Response · Authors · 2024-11-21
> **Response to Reviewer bSLg (Part 2/2)**
>
> **[W2] The concern about the quality of qualitative examples**
>
> We appreciate the reviewer’s observation regarding the qualitative examples in our website demo. Our primary focus in showcasing these examples is on how well our model adheres to the game logic, particularly in generating open-world game scenes. As mentioned on the project website (3a2077.github.io), we have updated additional visual comparisons to highlight that our model better accommodates the generation of open-world game content. From these examples(https://drive.google.com/file/d/1vZE4SKzLDqfErBV0B5MAbVHUizVZysdS/view?usp=sharing), we demonstrate that our model excels in supporting longer streaming generation and maintaining temporal smoothness, character and scene consistency, visual stability, game style, and camera-following logic.
> Meanwhile, we would like to clarify that we do not claim our model outperforms existing models, such as CogVideo-X, in every visual quality metric. In fact, our evaluation showed that some metrics, like the Dynamic Degree, may favor videos with pixel instability or mutations, while the Subject Consistency metric often assigns higher scores to more static clips. Despite this, our model demonstrates clear strengths in generating smoother, more stable scenes with consistent character details, and in producing more coherent game content videos.
>
> We sincerely appreciate the reviewer's insightful and constructive comments regarding our paper. In response to the main concerns, we have provided detailed clarifications above. Firstly, we compared multiple models and emphasized that our research is the first to systematically address the problem of open-domain game content generation and its interactive control. Through supplementary ablation experiments, we demonstrated and disentangled the significant contributions of both our method and dataset. Secondly, we updated the visual comparisons on our website, highlighting our model's advantages in generating open-world game content scenes, particularly in terms of temporal smoothness, character and scene consistency, visual stability, game style, and camera-following logic. Once again, we thank the reviewer for the valuable feedback to improve our paper.

---

> > ### Comment · Reviewer_bSLg · 2024-11-22
> >
> > Thanks for the new comparison experiment and qualitative examples. My concerns are addressed and I've raised my score to accept (8).

---

> > > ### Author Response · Authors · 2024-11-23
> > >
> > > Thank you for your positive feedback. We appreciate your recognition of our efforts to address your concerns regarding comparison fairness and the quality of qualitative examples. Your insights have been invaluable in guiding our revisions, and we are pleased to hear that the new comparison experiment and qualitative examples have met your expectations. Thank you for your thoughtful review and comments!

---

### Official Review · Reviewer_c3SK · 2024-11-04

**Soundness:** 3
**Presentation:** 3
**Contribution:** 3
**Rating:** 6
**Confidence:** 4

**Summary:**

In this paper the authors present a new dataset of modern AAA games for the purpose of world model training, which they call the Open-World Video Game Dataset (OGameData). Then then present their model, GameGen-X, a diffusion transformer for generating and controlling game video. GameGen-X is similar to other video generation models with the addition of InstructNet, which modifies the latents of GameGen-X for controllability. The authors present comparisons with a number of open-source video models. Finding that they generally produce more game-like video and may be better at control, according to some metrics.

**Strengths:**

The primary strength of the paper without question is the authors' new dataset. There is no dataset even close to this in terms of quality or size, it's a really exciting potential addition to this research area. This is primarily a strength in terms of originality, quality, and significance. I say primarily since the authors do not include access to the dataset at the review stage, though they do not have some metrics.

The authors' GameGen-X and InstructNet are also strengths, but I have concerns with them limiting them as strengths, as I'll get to below.

**Weaknesses:**

The paper is relatively free of weaknesses in terms of originality, thanks in large part due to OGameData. However, the authors' work has some weaknesses in terms of the quality, clarity, and significance. This primarily comes down to (1) the authors' stated motivations and how this aligns with their work, (2) the way the authors overview their system, and (3) the experiments

### Motivations and the Dataset

The authors motivate in two primary ways: (1) imagining this as a prototyping or early development tool for open world game developers and (2) imagining this as leading to future interactive experiences with greater user control. These are fine as motivations, but the authors' choice of processing the dataset runs counter to them, somewhat. Specifically, the authors have broken apart their video clips into distinct scenes, meaning their model or other models trained on this dataset will not observe scene transitions. This is somewhat of an oddity for either of the authors' stated motivations and there's no justification for this choice given in the paper. Similarly, the authors do not actually have control input for any of the collected game data. This is again a bit of an oddity given the authors' stated purposes. I would guess that the authors collected this data from some sort of web scrape of gameplay video rather than collecting the video themselves through playing games such that they could capture actual control inputs. However, this isn't specified in the paper. This is a potential concern, especially if the authors did scrape an online repository of videos when such scraping went against the terms and service of the site in question. This should be clarified.

### System Overview

Simply, the authors do not describe any of their system implementation in sufficient detail for replication. The authors state that code will be made available but do not make such code available for review. As such, there's no detail on the system architecture in terms of parameters or hyper parameters. The authors also not disclose the computation required to train their model or the training split used from their dataset. All of this would be required (potentially in appendices or in an external code repo) to ensure that the work is replicable.

### Experiments

I have a number of concerns with the current setup of the experiments. The authors only compare against open video models, which are not attempting the same task and are not trained on the same dataset. As such, it's unclear the extent to which this is just that the training dataset for GameGen-X is more similar to the test dataset. While the authors do specify that the experiments are over test data, given the distribution of games in the dataset, its highly likely that GameGen-X had already trained on the same game that the test data used in the experiments came from. As such, this seems much closer to testing on the training set.

The authors also have several metrics that require human expert raters, but who these experts were or what information they had is not specified. Further, the authors say they only use a single-blind setup, which may suggest the experts knew who they were. As such, there's a clear risk of bias here in terms of the experts feeling social pressure to more positively rate the more game-like videos if they knew that was the goal of this research. Clarity around the methodology and whether the authors have ethics approval would be necessary for readers to trust any of the human participant-based results. Relatedly, the authors state that the SR metric is "evaluated by both human experts and PLLaVA". However, the authors only present a single number. As such, it's not clear how the human expert and PLLaVA evaluations were combined. This throws doubt upon the SR metrics.

The authors repeatedly bold values from their own work, indicating it is the best, when there is equivalent work from prior models. This may mislead readers in terms of understanding when the performance.

The ablation study is helpful, as it demonstrates the value of OGameData, and several of the authors' components. However, since the authors do not train any other models on their dataset outside of these ablations, it's difficult to determine the exact value of the different components of their work.

Overall, I'd say that the experiments are currently the largest weakness of this paper.

**Questions:**

1. Where did the dataset come from?
2. Why splice up the data by scene?
3. What was the methodology with the human expert evaluators?
4. Did the authors receive ethics approval?
5. The authors indicate the dataset is split nearly evening between first and third person video, but primarily show results for third person video, why is this?

**Details Of Ethics Concerns:**

I no longer feel there is an ethical concern

---

> ### Author Response · Authors · 2024-11-21
> **Response to Reviewer c3SK (Part 1/4)**
>
> We sincerely appreciate the reviewer's insightful and constructive comments regarding our paper related to motivations, datasets, experiments, and system designs. We would like to take this opportunity to clarify the following points and update the Appendix in our revised manuscript:
>
> **[W1] Motivations and the Dataset**
>
> 1. **The Design for Video Scene Segmentation (Q2)**:
>
> Thank you for pointing out this issue.  It is worth clarifying that the purpose of segmenting scenes in our dataset is not to divide different in-game areas but to identify and handle artificial discontinuities in gameplay videos. We observed that some gameplay videos contain spliced segments. For example, a continuous gameplay segment might be followed by a transition animation, and then another continuous game·play segment (potentially featuring different characters and scenes without explicit gameplay transitions). Such artificial discontinuity will influence the model training.
>
> Instead, like what the reviewer mentioned, to ensure our dataset supports the long-duration continuation and natural scene transitions, we carefully segment and annotate single continuous gameplay video segments. This approach avoids the influence of artificial discontinuities, such as spliced transitions or animations, which are common in gameplay videos. Since our model inherently observes and learns to simulate long continuous gameplay data distributions, it can generate long-duration streaming video sequences (e.g., a streaming-generated demo with 10 times acceleration, https://drive.google.com/file/d/1vZE4SKzLDqfErBV0B5MAbVHUizVZysdS/view?usp=sharing, where it creates a man horsing in a forest and we use around 100 sets of control signal sequences to control it).
>
>
>  2. **The Design for Data Collection and Control Inputs**:
>
> Thank you for raising this important question.  As described in Section 2.1 of our paper, our dataset combines video data sourced from both the Internet and local collection.
>
> **Data Source**: Internet-collected videos, as the reviewer correctly noted, do not include corresponding frame-level control signals. To address this, we conducted additional local data collection to construct OGameData-INS, which includes approximately 100 hours of gameplay footage with paired control signals. This dataset is designed to support training for both dynamic environment control and character action control. By combining internet-collected data and locally collected data, we aim to leverage the broad generation capabilities learned from diverse internet data while enhancing precise control skills through detailed frame-level control signals in local data.
>
> **Data Compliance**:
> For Internet-sourced videos, we followed established practices from prior works ([1-6]) and adhered to platform fair use terms. The internet data collection method was inspired by Panda-70M [5] and MiraData [6], and we performed comprehensive cleaning and integration of the games and 3D rendering videos included in these datasets and also collected extra data from YouTube. To ensure compliance, we follow a data release paradigm similar to existing works [1-6], providing only textual annotations and corresponding timestamps and video URLs. Referencing previous works [1-6], details regarding data usage and agreements are included in Appendix B.1 Data Availability Statement to ensure transparency and alignment with platform rules. Our project remains strictly non-commercial and solely for research purposes.
>
> **Usage in Training**:
> Our dataset is used in two distinct stages of model training: 1. Pretraining with OGameData-Gen: This stage leverages internet-sourced data to generate diverse game scene videos. 2. Instruction Tuning with OGameData-INS: This stage uses locally collected data with control signals to support dynamic control of environments and character actions.
>
> In summary, we appreciate the reviewer's thoughtful comments on our dataset and are grateful for the opportunity to clarify these aspects. The points raised align with considerations we carefully addressed in our design, and we are encouraged by the recognition of our data contribution. The reviewer's insights have been invaluable in helping us refine our work, and we will ensure these clarifications are explicitly stated in the revised manuscript to improve its quality further.

---

> ### Author Response · Authors · 2024-11-21
> **Response to Reviewer c3SK (Part 2/4)**
>
> We acknowledge the importance of providing sufficient details to ensure replicability and appreciate the reviewer’s constructive comments in this regard. All implementation details, including model architecture, training strategies, and hardware specifications, are now included in the revised manuscript's appendix and will also be available in our future code repository upon publication.
>
> Below, we provide a comprehensive description, covering the training strategy, model architecture and hardware resources to address your concerns:
>
> **[W2] System Overview**
>
> 1. **Training Strategy**
>
> We employed a two-phase training strategy to optimize our model for both video generation and extension tasks. In the first phase, the base model was trained on a combination of text-to-video generation tasks (75% training probability) and video extension tasks (25% training probability). To expose the model to diverse scenarios, we utilized a bucket-based sampling strategy, which included videos of varying resolutions (480p to 1024×1024) and durations (1 to 480 frames at 24 fps). For example, 1024×1024 videos with 102 frames were sampled with an 8.00% probability, while 480p videos with 408 frames had an 18.00% sampling probability. Longer videos were processed by extracting random segments, and all samples were resized, center-cropped, and encoded using a 3D VAE, which compressed spatial dimensions by 8× and temporal dimensions by 4×. Training was optimized with the Adam optimizer (fixed learning rate of 5e-4) over 20 epochs, leveraging techniques like rectified flow to accelerate convergence and random text dropout (25% probability) to enhance generative robustness.
>
>  In the second phase, the focus shifted exclusively to video extension tasks, with videos fixed at a resolution of 720p and a duration of 4 seconds. Diverse control conditions were applied, including combinations of text, keyboard signals, and video prompts (e.g., canny-edge, motion vectors, and pose sequences). For all video extension tasks, the first frame latent was retained as a reference to improve consistency and performance.
>
>
> 2. **Model Architecture**
>
> Our model architecture consists of four key components: a 3D VAE, a T5 text encoder, a Masked Spatial-Temporal Diffusion Transformer (MSDiT) as the base model, and InstructNet for enhanced video extension control.
>
> The 3D VAE compresses videos in both spatial (8×) and temporal (4×) dimensions, extending the Stable Diffusion VAE with temporal layers and Causal 3D CNNs for inter-frame compression. This reduces computational costs while preserving video fidelity.
> The T5 text encoder supports inputs of up to 300 tokens, translating textual descriptions into embeddings for seamless integration with the video generation pipeline.
>
> The MSDiT comprises 28 layers of alternating Spatial and Temporal Transformer Blocks. An initial embedding layer compresses spatial features into tokens by performing an additional 2x downsampling along the height and width dimensions, reducing the spatial resolution further. The resulting latent representations are enriched with metadata (e.g., aspect ratio, frame count, timesteps, and frames per second) through MLPs, aligning them to the model’s latent channel dimension of 1152.  Advanced attention techniques like query-key normalization (QK norm) and rotary position embeddings (RoPE) are employed to enhance performance and stability. Masking mechanisms enable flexible support for both text-to-video generation and video extension tasks by conditioning selectively on unmasked frames.
>
> InstructNet extends the base model with 28 blocks, alternating between spatial and temporal attention, to integrate additional control inputs. Cross-attention fuses textual instructions, while keyboard signals are projected through MLPs to modify latent features. Video prompts, including canny-edge, motion vectors, and pose sequences, are fused at the embedding layer, enabling precise control over extended video outputs.
>
> 3. **Computation Resources and Costs**
>
> Our training infrastructure utilized 24 NVIDIA H100 GPUs (80GB each) across three servers, with 8 GPUs per server. Distributed training was implemented with Zero-2 optimization to minimize computational overhead. The training process consisted of two phases: base model training (25 days) and InstructNet training (7 days). Approximately 50TB of storage was required for datasets and model checkpoints.

---

> ### Author Response · Authors · 2024-11-21
> **Response to Reviewer c3SK (Part 3/4)**
>
> We sincerely thank the reviewer for this detailed feedback on the experimental setup, fairness of comparisons, and evaluation methodology. These insights have been invaluable in helping us identify areas for clarification and improvement.
>
> **[W3]. Experiment**
>
> 1. **Comparison Fairity and Ablation Study for Decoulped Contribution**:
>
> - Fairness of Comparisons
>
> We understand the concern regarding the fairness of comparing GameGen-X, which is specifically fine-tuned for open-world video games, with other general text-to-video diffusion models. In our experiments, we compared four models (OpenSora-Plan, OpenSora, MiraDiT, and CogVideo-X) and five commercial models (Gen-2, Kling 1.5, Tongyi, Pika, and Luma). OpenSora-Plan, OpenSora, and MiraDiT explicitly state that their training datasets (Panda-70M, MiraData) include a significant amount of 3D game/engine-rendered scenes. This makes them suitable baselines for evaluating game content generation.
>
> Additionally, while CogVideo-X and commercial models do not disclose training data, their outputs suggest familiarity with similar visual domains. We hope that this clarification of model capabilities will address the reviewer's concerns. Although there are no perfectly comparable works in game content generation, we have strived to ensure experiment fairness in terms of the model selection.
> To address concerns about potential overlap between training and test data, we ensured that the test set included only content types not explicitly present in the training set.
>
> - Ablation Study for Data and Model Contributions
>
> Additionally, to disentangle the effects of data and framework design, we sampled 10K subsets from both MiraData (which contain high-quality game video data) and OGameData and conducted a set of ablation experiments with OpenSora (a state-of-the-art open-sourced video generation framework). Due to the time limitation, we quickly verified the decoupled contribution based on these two additional experiments. We could compare more experiments in the future version.
> The results are as follows:
>
>   | Metric                | FID   | FVD    | TVA | UP | MS | DD | SC  | IQ |Alignment Metrics | Quality Metric |
>   |-----------------------|-------|--------|-----|----|----|----|-----|----|--------------------------|-----------------------|
>   | Ours / OGameData-Subset      | 289.5 | 1181.3 | 0.83 | 0.67 | 0.99 | 0.64 | 0.95 | 0.49 | 735.4                    | 0.76                  |
>   | OpenSora / OGameData-Subset  | 295.0 | 1186.0 | 0.70 | 0.48 | 0.99 | 0.84 | 0.93 | 0.50 | 740.5                    | 0.74                  |
>   | Ours / MiraData-Subset       | 303.7 | 1423.6 | 0.57 | 0.30 | 0.98 | 0.96 | 0.91 | 0.53 | 863.65                   | 0.71                  |
>
> As shown in the table above, we supplemented a comparison with OpenSora on MiraData. In comparing Alignment Metrics(averaged FID and FVD scores) and Quality Metrics (averaged TVA, UP, MS, DD, SC, and IQ scores), our framework and dataset demonstrate clear advantages. Aligning the dataset (row 1 and row 2), it can be observed that our framework (735.4, 0.76) outperforms the OpenSora framework (740.5, 0.74), indicating the advantage of our architecture design. Additionally, fixing the framework, the model training on the OGameData-Subset (735.4, 0.76) surpasses the model training on MiraData-Subset (863.65, 0.71), highlighting our dataset's superiority in the gaming domain. These results confirm the efficacy of our framework and the significant advantages of our dataset.
>
> - Further Clarification
>
> To further ensure fairness, contribution, and generalization, we have updated multiple sets of in-domain and open-domain generation samples in the Qualitative Comparison section on our project website (3a2077.github.io). These samples highlight: a) The existing open-sourced models can generate game scene videos, owing to Panda-70M and MiraData. 2) These samples show that our model performs better in generating known game scenes and creating new game content. Therefore, combined with the table above and the in-domain, open-domain, and streaming generation results demonstrate our contributions, as well as the generalization capability of our model (i.e., creating new game scenes and content).
>
> Overall, this work focuses on constructing a large-scale game content video generation model that can achieve open-domain generation, create new game scenes, and interact with them. It is the pioneering work attempting to systemically solve this problem from data construction and model design. In addition to quantitative analysis, this paper also emphasizes qualitative comparisons with existing open and closed models. These comparisons with other models are intended to illustrate our special capabilities in game video generation, open-domain generation, and interactive control, rather than to claim absolute superiority in all visual generation metrics or model abilities.

---

> ### Author Response · Authors · 2024-11-21
> **Response to Reviewer c3SK (Part 4/4)**
>
> 2. **Human Expert Raters and Evaluation Methodology**
>
> We conducted a human evaluation to assess the performance of our model across three critical metrics: user preference, text-video alignment, and control success rate. A total of ten expert evaluators with experience in gaming and the AIGC domain were recruited through an online application process. Before evaluation, all participants provided informed consent, acknowledging their understanding of the evaluation procedure and agreeing to participate in a research setting. To minimize bias in the evaluations, we implemented a blind evaluation protocol, where both the videos and the corresponding texts were presented without any attribution to the specific model that generated them.
>
> User Preference: To evaluate the overall quality of the generated videos, we focused on aspects such as motion consistency, aesthetic appeal, and temporal coherence. Videos were shown to evaluators without any textual prompts, isolating the assessment to purely visual characteristics to prevent any potential influence from the provided descriptions.
>
> Text-Video Alignment: This evaluation aimed to measure the semantic and stylistic fidelity of the videos relative to the textual prompts. We also evaluate how well the video represented the game type and style described in the prompts, as gaming aesthetics are crucial to the task.
>
> Control Success Rate: To assess the effectiveness of control signals in our model, we evaluated how accurately the model followed specific instructions in the prompts. For each prompt, three videos were generated using different random seeds to ensure diversity. Evaluators then scored each video on whether it successfully implemented the control instructions, using a binary scale (1 for success, 0 for failure).
> To complement the human evaluation, we also employed PLLaVA to generate captions for each video. These captions were then compared with the original prompts to ensure that key control elements—such as directional actions (e.g., "turn left") or contextual features (e.g., "rainy scene")—were accurately captured in the video. The final control success rate was calculated as the average of the human evaluation and AI-based caption analysis.
>
> 3. **Presentation of Bold Results**:
>
> We apologize for the negligence on the bold results and have fixed them in the experiments.
>
> **[Q1]. Where did the dataset come from?**
>
> We greatly appreciate the reviewer’s attention to the dataset source. As mentioned in the "Motivations and Dataset" section, the video sources are publicly available game content videos from YouTube, all of which were legally sourced and comply with platform policies. Additionally, we recorded local gameplay footage with proper permissions and respect for copyright laws. For the future dataset release, we only provide textual annotations and corresponding timestamps and video URLs following the style of previous works [1,6].
>
> **[Q2] Why splice up the data by scene?**
>
> We appreciate the reviewer’s question regarding the design of the scene cut. As mentioned in the "The Design for Video Scene Segmentation" part of our official response above (part 1/4), we have provided detailed clarification. The scene cut here is to find and split the artificial video transition, instead of cutting the in-game scene.
>
> **[Q4] Did the authors receive ethics approval?**
>
> During contributing to this work, we followed the ethical review process of our institution. We feel sorry that due to the double-blind policy, the relevant ethical review material cannot be provided here, and we will release it upon the paper's acceptance.
>
> **[Q5] The authors indicate the dataset is split nearly evenly between first- and third-person videos, but primarily show results for third-person videos**
>
> We appreciate the reviewer’s insightful comment. In our quantitative experiments, we did not differentiate between first-person and third-person game content video generation, as our focus was on overall model performance across both types. However, in our qualitative experiments, we have also included a substantial number of first-person video generation results, which are available on our anonymous website for further review.
>
> Reference:
>
> [1] Openvid-1m: A large-scale high-quality dataset for text-to-video generation, 2024.
>
> [2] Swap Attention in Spatiotemporal Diffusions for Text-to-Video Generation, 2023.
>
> [3] Internvid: A large-scale video-text dataset for multimodal understanding and generation, 2023.
>
> [4] Vript: A Video Is Worth Thousands of Words, NeurIPS, 2024.
>
> [5] MiraData: A Large-Scale Video Dataset with Long Durations and Structured Captions, NeurIPS, 2024.
>
> [6] Panda-70m: Captioning 70m videos with multiple cross-modality teacher, CVPR, 2024.

---

> > ### Comment · Reviewer_c3SK · 2024-11-23
> > **Re: Response to Reviewer c3SK**
> >
> > Thanks to the authors for all the work they've put into this rebuttal and answering my questions, it's definitely appreciated. As a reminder, I had concerns around the dataset, system overview, and experiments. I can now say that the authors have addressed the majority of my concerns, as such, I have greatly increased my overall rating of the paper from 3 to 6. The authors should be commended for the amount of work put in to improving the document!
> >
> > I would ask though what "we ensured that the test set included only content types not explicitly present in the training set" means in the authors' response? Does "content types" here mean that there are different games in the training and test sets? Clarity on this point would be helpful in addressing my last concerns around the experiments.

---

> ### Author Response · Authors · 2024-11-23
>
> Thank you very much for your thoughtful feedback and for taking the time to review our rebuttal. We are delighted to hear that we were able to address the majority of your concerns and that you found our paper revisions helpful. We are grateful for your constructive comments, which have been invaluable in enhancing the quality of our paper.
>
> For the question regarding content type, it refers to video content that features highly customizable game content elements not present in the training set, such as game scenes, protagonist outfits, environment dynamic changes, and camera angles and paths. Additionally, we have validated the model's emergence capabilities for generating creative game content including scenes and characters, etc. (Please refer to the Open-domain Generation Comparison Part at https://3a2077.github.io/; Figure 7 and Figure 8 in the manuscript; and the Further Qualitative Experiment section in the Appendix.)
>
> Thank you once again for your insightful comments and suggestions!

---

> > ### Comment · Reviewer_c3SK · 2024-11-27
> > **Re: Official Comment by Authors**
> >
> > Thanks for the clarification! I think that the paper likely would be improved by clarifying this, and particularly going into even more detail around the process for selecting clips for the test set. I also think it might have been interesting to test the approach on unseen games, particularly given the potential applications around generating rollouts for unseen/novel games.
> >
> > I'll keep my score as-is for now. I appreciate the answer but it was within my expectations during my last response.

---

> ### Author Response · Authors · 2024-11-28
>
> We are grateful to hear that our feedback has addressed your major concerns. We appreciate your insights and agree that further clarification and experiments on unseen games could enhance the paper, which would be further explored in the future version.  It would be greatly appreciated if you would consider recommending accepting our paper during the reviewers-PC discussion. We thank you once again for your valuable feedback and your great efforts in reviewing our paper.

---

### Author Response · Authors · 2024-11-21
**Update on Acknowledgment for Comments, Anonymous Website, Paper Revision, and Project Release**

The author team of GameGen-X sincerely appreciates your contributions in handling this submission and assisting us in refining the quality of this work. We are pleased to see that the reviewers acknowledge the following aspects of our work.
 1. **The unique contribution of OGameData, which focuses on the game video domain and has a well-curated filter, design, and annotation** (Reviewer c3SK, 1M8F, bSLg).
 2. **The first major contribution to large-scale, complex, open-world interactive video game generation** (Reviewer WQcm, 1M8F).
 3. **Technical contributions of GameGen-X, including better performance, complex system design, and interactive control** (Reviewer bSLg, WQcm).
 4. **Detailed experimental design and results** (Reviewer 1M8F, bSLg)

&nbsp;
As this is a pioneering exploration into **generating open-domain game video content and interacting with them**, we have made efforts to address the reviewers' concerns in the relevant sections.

1. **Updates on the Anonymous Website and Streaming Demo Video**

We updated several groups of qualitative comparison videos on our website (https://3a2077.github.io/).

Additionally, to answer specific problems from reviewers supplementarily, we updated videos to support:
  - **Long video generation and consistency**: a 400-second streaming-generated video with 10 times acceleration, where it creates a man horsing in a forest and we use around 100 sets of control signal sequences to control it. https://drive.google.com/file/d/1vZE4SKzLDqfErBV0B5MAbVHUizVZysdS/view?usp=sharing.
  - **The reason for minimizing UI elements**: sample videos in an early model without minimizing the UI elements in data collection, https://drive.google.com/file/d/1Te95mJf5tdHpmUOqCrwdfDCMmhJD8168/view?usp=sharing.
  - **Ability for generating multiple objects or first perspective videos**: sample videos with multiple objects or the first perspective, https://drive.google.com/file/d/1-PLP8sohLyI5Wsn_gnOPbDFjD-_c5Ppq/view?usp=sharing.
-  **Visualization of results across various resolutions and sampling steps**: sample videos with various resolutions and sampling steps, our model can achieve 20 FPS under 320p/10 sampling steps with acceptable visual quality (https://drive.google.com/file/d/16ibysz0LpdmPvew2elD4OcWu3GLooZok/view?usp=sharing).

2. **Paper Revision**
  Based on the reviewers' valuable comments, we revised our paper from the following perspectives:
  - Added Appendix B.1 Data Availability Statement and Clarification, and revised the data collection part in Appendix B.2 Construction Details: discuss and ensure the data compliance and supplement more details for the data collection pipeline including data source and keyboard data collection method.
  - Added Appendix C Implementation and Design Details: to improve the reproduction ability, we supplement more information related to the training strategy, the core design of the model architecture, and computation resources.
  - Added Appendix D.1 Fairness Statement and Contribution Decomposition: discuss the fairness of comparison with open-sourced models from the dataset and ability perspective and the decoupling of the contribution of OGameData and GameGen-X.
  - Added Appendix D.3 Human Evaluation Details: provide more information related to the evaluation pipeline and details.
  - Added Appendix D.4 Analysis of Generation Speed and Corresponding Performance: provide more information related to the generation speed and visual quality.
  - Beyond the significant parts mentioned above, we also included other valuable suggestions from reviews in our paper. Feel free to ask for any further advice and comments.

3. **Dataset & Code Release**

Reviewer c3SK raises a question that we believe will be a concern for all reviewers.
Below is our response:

**Yes**. We do have the plan to release our dataset and our code to support the research community once the paper is accepted, or possibly even earlier if the review scores are favorable. To demonstrate our commitment, we have provided a subset of small datasets on our anonymous website (around 10K annotations). Additionally, all reviewers are welcome to request extra video samples with specific prompts or functionality, and we will generate those videos and make them available on the website.

---

### Public Comment · ~Haoxuan_Che1 · 2025-02-08
**Acknowledge and Revision Illustration**

We appreciate the area chairs’ and reviewers' constructive feedback in reviewing our paper. Based on these comments, we have made improvements to the manuscript. We particularly appreciate the reviewers' recognition of our work, including motivations, experimental design, system architecture and the construction of the dataset.

In response to the review comments, we have made the following improvements in the camera-ready version:

- Added Appendices B.1 and B.2, providing detailed data availability statements and data collection process details.
- Added Appendix C, offering implementation details including training strategies, model architecture design, and computational resources.
- Added Appendices D.1, D.3, and D.4, supplementing fairness statements, contribution breakdown, human evaluation procedures, and generation speed and performance analysis.
- Fixed typos including the collected data modality, used computational resources, and the tense in the paper.
- Updated the comparison videos on the project website and added demonstration videos for long video generation, UI element minimization, multi-object generation, and effects at different resolutions.
- Updated acknowledgment, references, and citations to include previously omitted sources.
- Revised several key sections to clarify the methodology and contributions, ensuring a more objective and rigorous presentation of our findings.

We once again thank the area chairs and reviewers for their valuable suggestions.

---

### Meta-Review · Area_Chair_KTw3 · 2024-12-21

**Metareview:**

This is a nice paper, introducing an excellent dataset for generating game videos that will likely be of much use, and a diffusion-based method for controllable game video generation. The dataset is perhaps the biggest contribution, but the method, while not super novel, is also a contribution. Potential concerns includes that the paper is a little short on technical details (but that has improved, and the code will anyway be open-sourced) and that the authors seem to overselling their contribution ("unique", "pioneering"... just stick to the facts and people will take you more seriously). The concerns are not serious enough to prevent acceptance of the paper.

**Additional Comments On Reviewer Discussion:**

The authors engaged diligently and constructively with the reviewers.

---

### Decision · Program_Chairs · 2025-01-22

Accept (Poster)